# Parenchymal cues define Vegfa-driven venous angiogenesis by activating a sprouting competent venous endothelial subtype

**Laetitia Préau** [1,2,6], **Anna Lischke** [1,6], **Melanie Merkel**[1], **Neslihan Oegel** [1], **Maria Weissenbruch** [1], **Andria Michael**[1], **Hongryeol Park**[3], **Dietmar Gradl**[1], **Christian Kupatt** [4] & **Ferdinand le Noble** [1,2,5] ✉

Formation of organo-typical vascular networks requires cross-talk between differentiating parenchymal cells and developing blood vessels. Here we identify a Vegfa driven venous sprouting process involving parenchymal to vein cross-talk regulating venous endothelial Vegfa signaling strength and subsequent formation of a specialized angiogenic cell, prefabricated with an intact lumen and pericyte coverage, termed L-Tip cell. L-Tip cell selection in the venous domain requires genetic interaction between vascular Aplnra and Kdrl in a subset of venous endothelial cells and exposure to parenchymal derived Vegfa and Apelin. Parenchymal Esm1 controls the spatial positioning of venous sprouting by fine-tuning local Vegfa availability. These findings may provide a conceptual framework for understanding how Vegfa generates organo-typical vascular networks based on the selection of competent endothelial cells, induced via spatio-temporal control of endothelial Kdrl signaling strength involving multiple parenchymal derived cues generated in a tissue dependent metabolic context.

During evolution, vertebrates developed intricately branched vascular networks that perfuse organs according to a form follows function principle. In this optimization model, the spatial vessel organization is governed by cross-talk between vascular structures and the surrounding tissues, and involves integration of local metabolic and hemodynamic cues[1–3]. However, the molecular and cellular substrates that add organ specificity to the vascular growth and patterning process are largely unknown[2,3]. Vascular endothelial growth factor (Vegf) and its receptors Vegf-receptor-2/Kdr and Vegf-receptor-1/Flt1 are key-regulators of developmental angiogenesis and branching remodeling in most organs including pancreas, liver, skeletal muscle, heart, brain and peripheral nervous system[4,5]. How this relatively limited set of molecules can account for the great diversity in organ vessel patterning is unknown but considered clinically relevant for designing organo-typical pro- and anti-angiogenic therapies[1,2,6]. Here we addressed the molecular basis accounting for heterogeneity in

[1]Department of Cell and Developmental Biology, Institute of Zoology (ZOO), Karlsruhe Institute of Technology (KIT), Fritz Haber Weg 4, 76131 Karlsruhe, Germany. [2]Institute for Biological and Chemical Systems—Biological Information Processing, Karlsruhe Institute of Technology (KIT), PO Box 3640, 76021 Karlsruhe, Germany. [3]Dept. Tissue Morphogenesis, Max Planck Institute for Molecular Biomedicine, Roentgen Strasse 20, 48149 Muenster, Germany. [4]Klinik und Poliklinik für Innere Medizin I, Klinikum rechts der Isar, Technical University Munich, and DZHK (German Center for Cardiovascular Research), partner site Munich, Munich, Germany. [5]Institute of Experimental Cardiology, University of Heidelberg, Im Neuenheimer Feld 669, 69120 Heidelberg, Germany and DZHK (German Center for Cardiovascular Research), partner site Heidelberg/Mannheim, Heidelberg, Germany. [6]These authors contributed equally: Laetitia Préau, Anna Lischke. ✉e-mail: ferdinand.noble@kit.edu

angiogenic remodeling processes and the contribution of parenchymal cues herein by systematic screening of sprout remodeling in zebrafish with a Vegf gain of function scenario.

Vegf is best known for promoting sprouting angiogenesis[7,8]. Sprouting angiogenesis relies on a migratory endothelial tip cell guiding the vessel sprout at the forefront, and trailing proliferative stalk cells that elongate the sprout and form the sprout lumen[8,9]. The reference model explaining the specification of tip versus stalk cell involves a Vegfa-Kdr-Dll4-Notch-mediated lateral inhibition mechanism which describes sprouting in a number of systems most notably the neonatal mouse retina and zebrafish embryo trunk vasculature[10–14]. Vegfa and activation of endothelial Notch are furthermore implied in arterial specification of endothelial cells[15–17] and in directing tip-derived endothelial cells into developing arteries involving downstream activation of Cxcr4, implicating that Vegfa-Dll4-Notch signaling couples sprouting angiogenesis with artery formation[18–20]. Notch function in sprouting angiogenesis however, shows regional organo-typical differences: e.g. while in the neonatal retina Notch acts as a negative regulator of tip cell selection, in developing bone Notch acts as a positive regulator of angiogenesis to promote bone growth[21].

Single cell sequencing studies suggest a thus far underappreciated role for venous derived endothelial cells in the formation of organo-typical vascular networks and vascular remodeling in (pathological) conditions characterized by excessive Vegf[22–27]. However, the specific functional and molecular mechanisms of how venous endothelial cells integrate specific local tissue cues to achieve organotypical sprouting angiogenesis remain to be elucidated. Sprouting angiogenesis involves complex endothelial cell behaviors that can be studied at single cell resolution in the vasculature of the developing zebrafish embryo[28]. In vivo imaging of zebrafish embryos with a Vegfa gain of function scenario shows that upon Vegfa, sprouts preferentially emanate from trunk intersegmental veins, not from intersegmental arteries[29,30]. These observations are intriguing as loss of function studies have previously shown that Vegfa is essential for primary artery sprouting, and Vegfc-Flt4 signaling for venous sprouting events[31]. The open questions are what determines Vegfa sprouting responsiveness in venous endothelial cells, and are there differences between the molecular and cellular processes that govern Vegfa-driven sprouting in the arterial versus the venous domain. Moreover, venous sprouting upon Vegfa is spatially restricted to the dorsal aspect of intersegmental veins, a domain that is in close contact with the developing spinal cord neural system[29,30]. Ablation of spinal cord radial glia cells[30] or neural cell specific inactivation of flt1[29] phenocopies venous hypersprouting as observed upon Vegfa suggesting that neural cells of the juxta-posed parenchyma may provide local instructive cues determining the spatial positioning of Vegfa-driven venous sprouting events. Still unresolved is how such parenchyma derived cues determine local Vegf bioavailability and venous Kdrl signaling strength to promote venous sprouting specifically at the neuro-vascular interface[29,30,32].

Here, by analyzing zebrafish mutant embryos with a Vegfa gain of function scenario we find that Vegf induced venous sprouting requires genetic interaction between Apelin-receptor-a (aplnra) and Kdrl to promote Vegfa signaling strength in a specific subset of venous endothelial cells. These venous cells migrate, in a blood flow dependent manner from the ventral to the dorsal aspect of venous ISVs, where they become juxta-posed to Vegfa and Apelin producing spinal cord neural cells of the surrounding parenchyma and subsequently start to form a lumenized sprout. Loss of neural apln abrogates this process. We show that the venous endothelial cell selected for sprouting, here termed lumenized tip cell (L-Tip cell), increases in size involving activation of the GEF Trio, and remodels into a unicellular, lumenized and pericyte covered long-range anastomotic connection with an arterial ISV. After connecting with the arterial ISV, a perfused anastomotic vessel network forms at the level of the spinal cord. Unlike Vegfa-driven arterial sprouting, Vegfa induced sprout selection in the

venous domain occurs independent of Dll4-Notch. We furthermore show that Vegfa induced venous sprouting requires expression of endothelial cell specific molecule-1 (esm1) by spinal cord neural cells. Esm1 is upregulated in flt1[-/-] and Esm1 is regarded as a competitive antagonist for Vegfa binding to the extracellular matrix (ECM)[33]. Consequently, expressing esm1 and thereby masking potential Vegfa binding sites in the parenchymal ECM, effectively increases Vegfa availability, whereas loss of esm1 and increasing ECM binding, decreases Vegfa. In line with this we show that loss of (neural) esm1 specifically inhibits ectopic venous sprouting in embryos with a Vegfa gain of function scenario including flt1[-/-] and vhl[-/-] mutants, whereas neural specific overexpression of esm1 augments the extent of venous sprouting and anastomosis formation.

We conclude that Vegfa-driven venous sprouting involves a specific vein−parenchymal cross-talk mechanism regulating local Vegfa availability and angiopotential of venous aplnra-kdrl co-expressing endothelial cells. Such findings may provide a conceptual framework for understanding how Vegfa may generate organo-typical vascular networks based on the selection of specific venous angiogenic endothelial cells that are activated via spatio-temporal control of endothelial Kdrl signaling strength involving multiple parenchymal cues generated in a tissue dependent metabolic context. These observations may open new therapeutic avenues for selective vascular targeting approaches based on tissue specific modulation of Vegf levels.

## Results

### Vegf gain of function induces venous sprouting involving lumenized tip cells

We addressed tissue specific angiogenic remodeling events by systematic analysis of mutants with a Vegfa gain of function scenario (Fig. 1a–c; Supplemental Fig. 1p, q)[29]. In vivo confocal imaging of trunk vascular remodeling in flt1[-/-] and vhl[-/-] mutants showed ectopic angiogenic sprouts emanating from the dorsal aspect of intersegmental veins (vISV) at the level of the spinal cord (Fig. 1b, c). Based on the spatio-temporal differences with previously reported Vegfa-driven primary sprouting (which gives rise to aISVs) and Vegfc driven secondary sprouting (which gives rise to vISVs and lymphatics) events in the trunk[32], we termed this process tertiary sprouting. Tertiary sprouts emanated from the dorsal aspect of venous ISVs, extended along the spinal cord, and connected with an arterial ISV to form a perfused anastomotic vessel segment (Fig. 1d, e); such anastomotic connections were not observed in WT embryos (Fig. 1b, c).

Vegf-induced tertiary sprouts typically consisted of one single endothelial cell (Fig. 1d; Supplemental Fig. 1b–e), as opposed to the 2–3 cells generally observed in the Vegf driven primary sprouts (Supplemental Fig. 1a, c). This single endothelial cell remodeled into a lumenized and blood perfused anastomotic vessel segment connecting the vISV it emanated from, with the lateral aISV (Fig. 1d; Supplemental Fig. 1h), without additional contribution of stalk or phalanx cells. Upon anastomosis formation, endothelial cells migrated from the ISVs resulting in increased EC numbers in the anastomotic segment (Supplemental Fig. 1f, g). When comparing the tip cell of primary artery sprouts with the tip cell of tertiary sprouts we furthermore noticed differences in the morphology, in particular with respect to the distribution of filopodia extensions. Filopodia in tertiary sprouts were more evenly distributed along the tip cell surface when compared to the filopodia extensions observed in primary sprouts which were more localized at the tip of the cell (Supplemental Fig. 1i–k).

Injection of dextran-FITC to visualize blood plasma distribution showed that tertiary sprouts already carried blood plasma during early sprouting stages (Fig. 1e, left panel, blood plasma was observed in 54/61 sprouts) and continued to do so during the subsequent stages of anastomosis formation (Fig. 1e, right panel; Supplemental Fig. 1m, n). This is consistent with the tertiary sprout being lumenized and functionally connected to the perfused venous circulation (Fig. 1e;

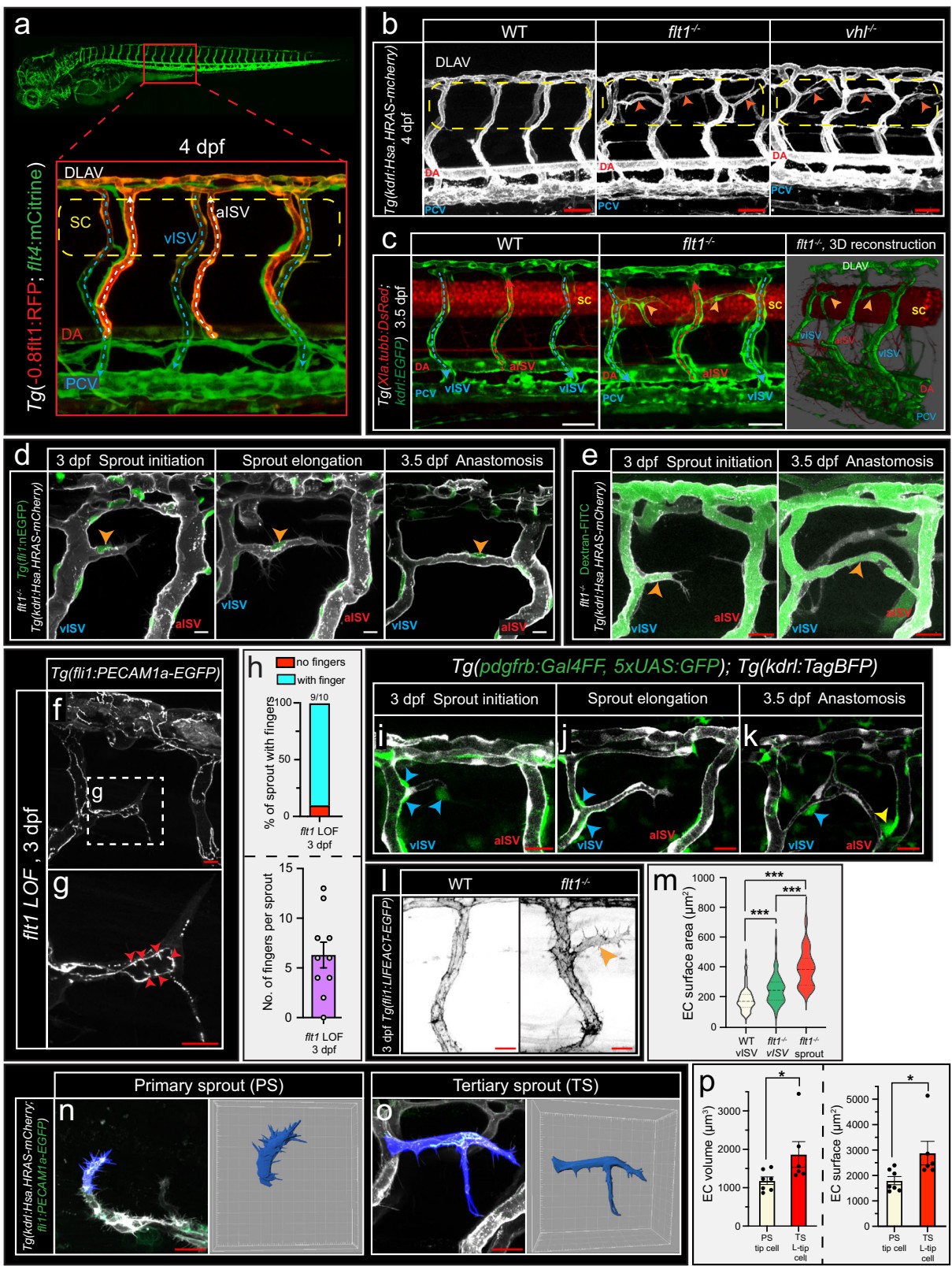

Supplemental Fig. 1m–o). In further support of this, we observed the formation of *pecam1* positive, finger like endothelial junctions at the tip of the developing sprout (Fig. 1f–h). Such endothelial junctional structures are typically induced once a developing vessel is exposed to intraluminal pressure[34]. This indicates that the tip cells of tertiary sprouts were lumenized, carried blood plasma and were exposed to the venous intraluminal blood pressure during their expansion (Supplemental Fig. 1h). Tip cell lumenization is a fundamental difference with the Dll4-Notch sprouting model. In the reference model, lumenization is considered a specific feature of the stalk cell, not of the tip cell[9,35]. Consequently, in Vegf driven primary artery sprouting, lumenization occurs during later stages of the sprouting remodeling process when compared to tertiary sprouting (Supplemental Fig. 1l, n, o).

**Fig. 1 | Tertiary sprouts emanate from veins, and consist of a single lumenized endothelial cell that remodels into an anastomotic connection. a** Confocal images of *Tg(−0.8flt1:RFP;flt4:mCitrine)* showing overview and high magnification of trunk vasculature. Dotted arrows indicate flow direction; Dotted box shows ROI. **b** Trunk vasculature in WT, *flt1⁻/⁻* and *vhl⁻/⁻* mutants. Arrowheads indicate ectopic sprouts. Dotted line indicates the spinal cord. **c** Confocal images showing blood vessels (green) and spinal cord neuronal cells (red) in WT (left) and *flt1⁻/⁻* mutant (middle). 3D reconstruction of a confocal stack in *flt1⁻/⁻* mutant (right). Arrowheads indicate tertiary sprouts. **d** Tertiary sprout branching from vISV during initiation (left), elongation (middle) and upon anastomosis formation (right). Arrowheads indicate nuclei. **e** FITC-Dextran labelled blood plasma shows lumenized tertiary sprout at initiation stage (left) and upon anastomosis formation (right). **f** Confocal imaging of *Tg(fli1:PECAM1a-EGFP)* upon *flt1* loss of function (LOF). **g** Endothelial junctional morphology in the lumenized endothelial tip cell. Arrowheads indicate finger-like structures. **h** Finger-like structure number per sprout upon *flt1* LOF. Percentage (top) and Mean±s.e.m. (bottom); *n* = 10 sprouts/embryos. **i–k** Confocal images of pericyte (green) distribution surrounding blood vessels (blue

arrowheads) during tertiary sprout initiation (**i**), elongation (**j**) and anastomosis formation (**k**) in *flt1⁻/⁻* mutant. Yellow arrowhead indicates pericytes at the anastomotic-aISV connection. **l** Confocal imaging of actin reporter in WT and *flt1⁻/⁻* mutant, arrowhead indicates tertiary sprout. **m** Endothelial surface area for indicated scenario. Violin plot shows median and interquartiles; two-sided Mann−Whitney *U* test, WT: *n* = 74 cells (13 embryos), *flt1⁻/⁻* vISV: *n* = 116 cells (13 embryos), *flt1⁻/⁻* sprout: *n* = 28 cells (13 embryos). *p* < 0.0001 for all comparison. **n**, **o** Confocal imaging of *Tg(kdrl:Hsa.HRAS-mcherry;fli1:PECAM1a-EGFP)*, segmentation and 3D rendering of primary sprout tip cell (**n**) and tertiary sprout volume (**o**). **p** Cell volume and surface area for indicated scenario. Mean ± s.e.m., two-sided Mann−Whitney *U* test, *n* = 7 tip cells/embryos and 6 L-Tip cells/embryos. *p*(volume) = 0.0221, *p*(surface) = 0.0350. Scale bars indicate 50 μm in **b**, **c**; 20 μm in **d**, **f**, **g**, **i–l**; 25 μm in **e**, **n**, **o**. DA dorsal aorta, SC spinal cord, DLAV dorsal lateral anastomotic vessel, aISV arterial intersegmental vessel, vISV venous intersegmental vessel, PCV posterior cardinal vein, dpf days post fertilization, LOF loss of function, EC endothelial cell, PS primary sprouting, TS tertiary sprouting. Source data are provided as a Source Data file.

The presence of blood plasma and the exposure to blood pressure already during early sprout remodeling stages raised the question to what extent hemodynamics are involved. To test this, we lowered heart rate and trunk perfusion in *flt1⁻/⁻* mutants with tricaine[36,37] and found that reducing perfusion inhibited the tertiary sprouting process (Supplemental Fig. 2a–c). Increasing heart rate and trunk perfusion using epinephrine had no significant impact on tertiary sprouting propensity in *flt1⁻/⁻* mutants (Supplemental Fig. 2d–f). Pericytes may aid to stabilize the pressurized expanding tertiary sprout. We observed that the nascent tertiary sprout recruited pericytes in particular at the base of the sprout—the domain where it branched from the vISV (Fig. 1i–k), and around the arterial entry site, the domain where the angiogenic segment made a connection with the perfused aISV (Fig. 1k; Supplemental Fig. 2h). Overall, in *flt1⁻/⁻* mutant embryos, pericyte numbers in both vISVs and aISVs were increased when compared to WT (Supplemental Fig. 2g–i). Since the developing tertiary sprout is exposed to hemodynamic forces, such strategically positioned pericytes, in particular at the branch points, may provide physical stability, and prevent plasma leakage.

Tertiary sprout morphogenesis furthermore involved an increase in endothelial tip cell size (surface area, volume) when compared to venous endothelial cells in other domains, or primary artery tip cells (Fig. 1l–p). This endothelial enlargement may enable the single endothelial cell in a tertiary sprout to both bridge the distance between the vISV and the lateral aISV, and at the same time establish a vessel lumen. Consistent with the tertiary sprout being unicellular, shuffling between the leading cell and trailing cells or competition for the lead position, two typical characteristics of the Vegf-Dll4-Notch sprouting model[38], were not observed during tertiary sprouting. We next reasoned that the differences in sprout morphology may affect metabolic sensitivity, and found that tertiary sprouting was significantly more glycolysis-dependent than primary artery sprouting (Supplemental Fig. 2j–m). Furthermore, while primary artery sprouting involves specific Ca²⁺ signals in the differentiation of artery tip cells[39], such specific Ca²⁺ signals were not observed in the developing tertiary sprout (Supplemental Fig. 2n–q).

Consistent with tertiary sprouts emanating from veins, we found that tertiary sprouts expressed the venous marker *flt4*, and continued to do so until they made an anastomotic connection with the lateral aISV (Supplemental Fig. 3; Supplemental Movie 1). To substantiate that tertiary sprouting requires endothelial cells with a venous identity, we created a trunk vasculature in which almost all trunk ISVs remained arterial by inhibiting *flt4* and suppressing secondary sprouting[29,31]. When the trunk vasculature consists of arterial ISVs only, we find that tertiary venous sprouting is absent (Supplemental Fig. 3c–e). If tertiary sprouting indeed requires venous endothelium, we next reasoned that promoting venous ISV formation in *flt1⁻/⁻* mutants should augment

branching. In zebrafish, it is established that loss of the Notch ligand *dll4* promotes venous cell fate and *dll4* loss-of-function embryos display a trunk vasculature consisting almost exclusively of venous ISVs, at the expense of aISVs[12,29]. Accordingly, loss of *dll4* in *flt1⁻/⁻* mutants, and generating a trunk consisting of veins only, significantly augmented ectopic venous branching when compared with control (Supplemental Fig. 3f–k). The spatial positioning of the venous sprout at the level of the spinal cord (Supplemental Fig. 3i–k), and lumenization of the anastomosis connection were not impaired upon loss of *dll4* (Supplemental Fig. 3i–m). Since upon loss of *dll4* almost all aISVs transformed into a vISV, the anastomotic connections now connected vISV-vISV instead of a vISV-aISV. In contrast to our observations in the spinal cord, in other areas of the nervous system, loss of *dll4* has been shown to abrogate sprout lumenization, and anastomosis formation[19,40]. Taken together, our data suggest that in the tertiary sprouting scenario *dll4* is not required for lumenized tip cell selection in the venous domain, neither for sprout expansion along the spinal cord nor anastomosis formation. *Dll4* merely acts as regulator of venous identity and loss of *dll4* increases venous sprouting propensity by increasing the number of venous ISVs.

Collectively, tertiary sprouts emanated from veins, were pre-fabricated with an intact endothelial lumen and accompanied by pericytes to allow immediate perfusion upon connection to arterial intersegmental vessels. The leading cell in tertiary sprouts displayed both "tip cell" and "stalk cell" features including the ability to form a lumen. To discriminate between this special venous endothelial cell, and the classical tip-stalk cells, we termed the lumenized venous endothelial cell forming the tertiary sprout, "L-Tip cell" (Supplemental Fig. 2r, s; Supplemental Fig. 14).

## Tertiary sprouting and interactions with the parenchyma

Since tertiary sprouts developed specifically at the level of the spinal cord, we next focused on the potential contribution of neural−endothelial cell interactions in promoting L-Tip cell formation and tertiary sprout remodeling. Single cell sequencing of trunk cells and unsupervised hierarchical clustering established 33 clusters (Fig. 2a; Supplemental Fig. 4a, b); sub-clustering of the neural cell population identified 17 distinct clusters (Fig. 2b; Supplemental Fig. 4c). In order to predict cell signaling pathways involved in endothelial−neural cell cross-talk, we used the CellChat tool, which computationally analyzes scRNA-seq expression data with a database of ligand-receptor interactions to predict cell-to-cell interactions[41]. CellChat predicted that neural cells are capable of communicating with endothelial cells, and that this communication is bidirectional (Fig. 2c, d). We then identified the main pathways targeting endothelial cells and found 8 major groups including the Vegf, Cxcl12, and Apelin pathways (Fig. 2e). Analysis of ligand-receptor pairs in both WT and *flt1⁻/⁻* datasets

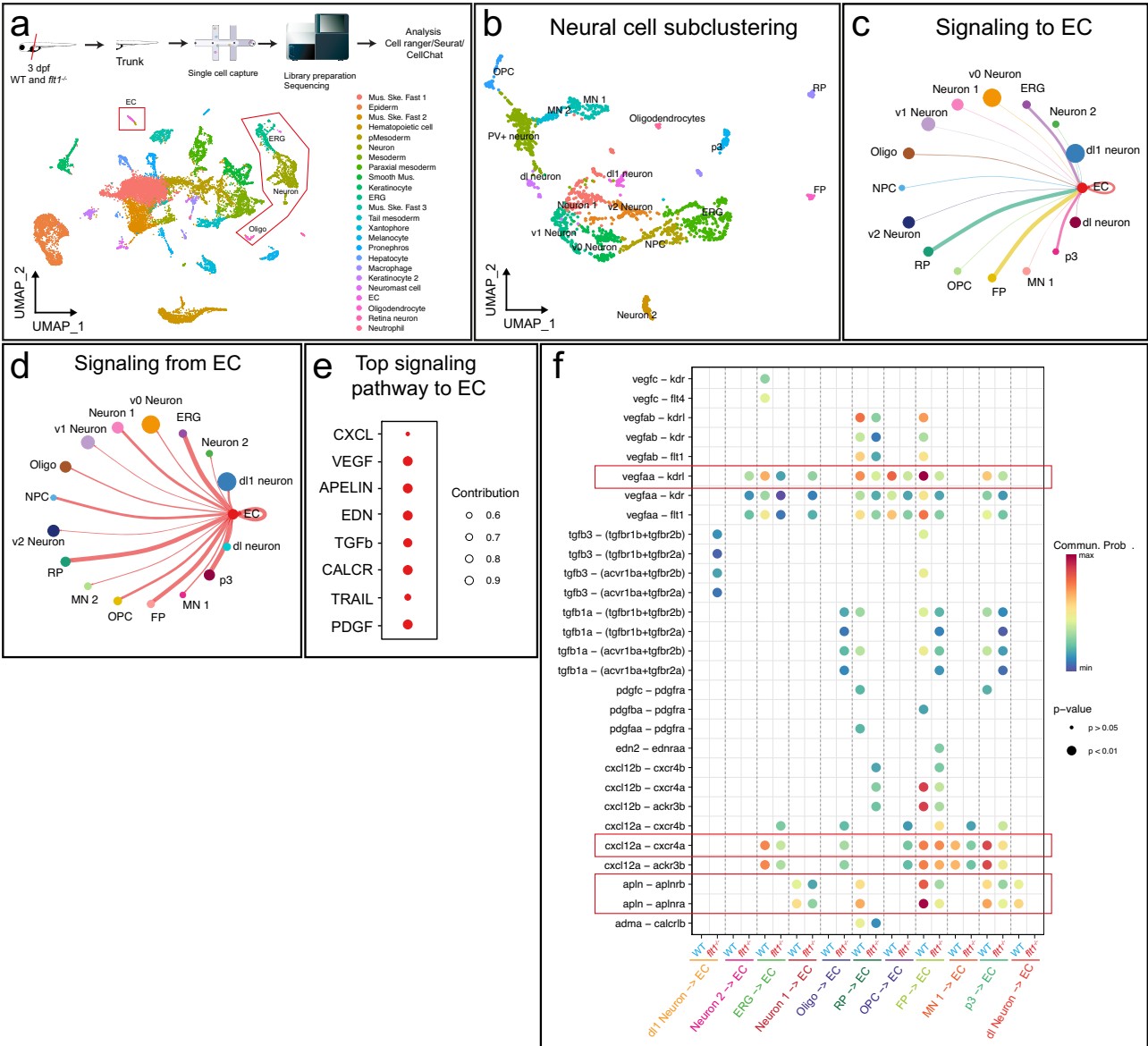

**Fig. 2 | In silico predictions of neural to endothelial cell communication.**
**a** Experimental pipeline and UMAP plot showing the different cell clusters identified by single cell sequencing of 3 dpf WT and *flt1*⁻/⁻ embryos. **b** UMAP plot of the neural cell sub-clusters. Each color represents an individual sub-cluster. **c** Predicted interactions from neural cells to ECs. **d** Predicted interactions from ECs to neural cells. **e** Main pathways predicted to signal from neural cells to ECs. **f** Overview of potential ligand-receptor interactions from neural cells to ECs in WT and *flt1*⁻/⁻.

Circle size indicates p-values. The color scale indicates the communication probability. EC endothelial cell, ERG ependymal radial glia, Oligo oligodendrocytes, Mus. muscle, Ske. skeletal, OPC oligodendrocyte progenitor cell, MN motoneuron, PV parvalbumin, dl dorsal, v ventral, p3 progenitor 3, NPC neuronal progenitor cell, RP roof plate, FP floorplate, EDN endothelin, CALCR calcitonin receptor, TRAIL tumor necrosis factor-related apoptosis-inducing ligand, EDN Endothelin, TGFb Transforming growth factor beta, PDGF Platelet-derived growth factor.

suggests that the most likely candidates for neural-vascular cross-talk are neural derived Vegfaa, Cxcl12a, and Apelin interacting with endothelial Kdrl, Cxcr4a, and Aplnra/b respectively (Fig. 2f; feature plots, violin plots and chord diagrams in Supplemental Fig. 5a–c).

CellChat indicates the likelihood that certain cell-cell interactions may occur, and these computational predictions have to be validated in vivo. It was previously shown that upon neural derived Vegfaa, signaling through vascular Kdrl receptors contributes to trunk spinal cord vascularization in zebrafish[29,30]. In zebrafish hindbrain, Cxcr4a acts downstream of Dll4-Notch in tip cells and controls sprout migration in response to Cxcl12[19]. Morpholino mediated knockdown of *cxcr4a* inhibited angiogenesis in the hindbrain, consistent with the phenotype reported in *cxcr4a* mutants[40], but had no significant impact on tertiary sprouting or lumenization of the anastomotic segments in the trunk of *flt1*⁻/⁻ mutants (Supplemental Fig. 6a–h). In addition, pharmacological

inhibition of Cxcr4 in *flt1*⁻/⁻ mutants had no impact on tertiary sprouting (Supplemental Fig. 6i–l). Similar results were obtained upon morpholino mediated knockdown of *cxcl12a* (Supplemental Fig. 6m–r). Although CellChat predicted a role for Cxcr4a-Cxcl12a, our validation experiments suggest that this signaling cascade is not responsible for tertiary sprouting at the level of the trunk spinal cord in *flt1*⁻/⁻ mutants. We next focused on the neural Apelin to vascular Apelin-receptor-a signaling axis and analyzed a series of (tissue specific) apelin ligand/receptor loss of function scenarios.

## Tertiary sprouting requires genetic interaction between vascular Kdrl and Apelin-receptor-a
Tertiary sprouting is a Vegfa-driven process involving Kdrl signaling[29]. However, *kdrl* is expressed by all endothelial cells and hence Kdrl alone cannot account for venous specificity. We therefore argued that Kdrl

must genetically interact with another receptor and we considered *aplnra*. In zebrafish, two paralogues of the apelin receptor exist, *apelin-receptor-a* (*aplnra*) and *apelin-receptor-b* (*aplnrb*). *Aplnrb* has been implied in arterial sprouting and remodeling[42], and thus far no functional role for *aplnra* has been established[42,43]. *Aplnra* was significantly upregulated in *flt1*[−/−] (Supplemental Fig. 7a), and loss of *aplnra* reduced tertiary sprouting events both in *flt1*[−/−] and in *vhl*[−/−] mutants (Fig. 3a, b, d, e; Supplemental Fig. 7k, l), without affecting the aISV/vISV ratio (Fig. 3c; Supplemental Fig. 7m). The trunk vasculature of *aplnra*[−/−] mutants and morphants looked phenotypically normal (Supplemental Fig. 7b–e, g–j), and *aplnra* mutants showed normal heart rates (Supplemental Fig. 7f). Single cell sequencing of GFP+ cells isolated from *Tg(kdrl:EGFP)* reporter embryos (Fig. 3f, g; Supplemental Fig. 8a) and sub-clustering of the endothelial population (Fig. 3h) identified 12 clusters (Fig. 3i, j). Expression of *aplnra* showed signs of zonation with relatively low expression in the arterial-EC domain (aEC), but steadily increasing expression levels when moving from the artery-capillary domain toward the capillary and the venous capillary domain, to finally obtain the highest expression levels in the venous endothelium domain (vEC) (Fig. 3k). This *aplnra* expression profile would favor *aplnra* preferentially exerting its function in the venous domain, upon cross-talk with neural cells, as indicated by the CellChat analyses (Fig. 2f). We thus hypothesized that at the neuro-vascular interface neural Apelin may signal to venous endothelial Aplnra to enhance Kdrl signaling strength and initiate sprouting specifically at this position (Supplemental Fig. 15).

To enhance Kdrl signaling strength, Apelin receptors may couple to Kdrl receptors through the pertussis toxin (PTX) sensitive Gαi protein[44,45]. We next switched off the coupling between the two receptors by overexpressing PTX[46] specifically in the venous domain. Venous specific inhibition of Gαi signaling in *flt1*[−/−] mutants significantly reduced tertiary sprouting (Fig. 3l, m), without affecting trunk aISV / vISV ratio (Fig. 3n). In human cells, the non-receptor tyrosine kinase c-Abl has been suggested to link G-protein function with Vegf-receptor-2 signaling and loss of Gαi has been shown to reduce c-Abl-phosphorylation subsequently resulting in reduced Kdr-phosphorylation[47]. In line with a functional role for c-Abl in the coupling between Aplnra and Kdrl, we found that inhibiting c-Abl in *flt1*[−/−] or *vhl*[−/−] mutants using the ATP-competitive tyrosine kinase inhibitor Dasatinib (Fig. 3o, p; Supplemental Fig. 8b, c) or Olverembatinib (Supplemental Fig. 8d, e) reduced tertiary sprouting. Taken together these data suggest that in venous endothelial cells Aplnra couples to Kdrl to promote tertiary sprouting.

*Aplnrb* was significantly upregulated in *flt1*[−/−] (Supplemental Fig. 7n), and preferentially expressed in the arterial domain, most notably arterial tip cells (Fig. 3k). Accordingly, *aplnrb*[−/−] mutants and *aplnrb*[−/−];*flt1*[−/−] double mutants showed severe trunk arterial ISV and DLAV remodeling defects (Supplemental Fig. 7o–w), which consequently resulted in trunk vISV perfusion deficits, hence rendering analysis of hemodynamics driven tertiary sprouting in the venous domain impossible. As *aplnrb* was predominantly expressed in arterial tip cells and since *aplnrb* acted as regulator of aISV remodeling, we ruled *aplnrb* out as specific regulator of tertiary sprouting and therefore focused on Apelin-receptor-a.

## Loss of *aplnra* impairs endothelial cell polarization and cell migration in veins

Since venous endothelial cells are enriched in *aplnra*, and sprouts arise in the dorsal part of venous ISVs—a domain that during early stages of trunk vascular remodeling is occupied by remodeled artery derived endothelial cells[48], the question arises when and how *aplnra* expressing venous cells colonize the neuro-vascular interface domain of the vISV. It is established that with time the artery derived cells in the dorsal domain of vISVs will be displaced by venous endothelial cells originating from the posterior cardinal vein (PCV)[48]. To this end PCV derived venous endothelial cells, migrate from ventral to dorsal, against the direction of the blood flow in the vISV, to finally occupy the dorsal domain and displace artery derived cells[48]. Loss of *flt1* resulted in more endothelial cells per venous ISV but had no significant impact on ventral-dorsal movements when compared to WT (Supplemental Fig. 9a–e; Supplemental Movie 2 and 3). Loss of *aplnra* in *flt1*[−/−] significantly reduced ventral-dorsal migration distance of venous endothelial cells in vISVs (Fig. 4a–d). Reduced ventral-dorsal migration upon loss of *aplnra* resulted in accumulation of endothelial cells in the ventral part of the vISV (Fig. 4e, f). Consequently, displacement of artery derived endothelial cells in the dorsal domain by PCV derived venous cells was significantly reduced as evidenced by analysis of *flt4* (venous marker)−*flt1* (arterial marker) expression domains in the vISV upon loss of *aplnra* in *flt1*[−/−] mutants (Fig. 4g, h). Comparable ventral-dorsal migration defects were observed when reducing trunk blood flow perfusion with tricaine (Supplemental Fig. 9f–j). Time-lapse imaging in *flt1*[−/−] mutants injected with *aplnra* targeting morpholino confirmed reduced ventral-dorsal venous cell movements and impaired displacement of remodeled artery derived cells toward the DLAV (Supplemental Fig. 9m–o; Supplemental Movie 4 and 5). Arterial-venous cell fate changes upon loss of *aplnra* inducing arteriolization of vISVs were not observed (Supplemental Fig. 7e, j; Supplemental Fig. 9n, o). Migration against the direction of blood flow has been suggested to involve endothelial cell polarization (Fig. 4i)[48]. WT and *flt1*[−/−] mutants displayed comparable polarization patterns in vISVs (Supplemental Fig. 9k, l). In contrast, loss of *aplnra* expression in *flt1*[−/−] significantly impaired the ability of endothelial cells to polarize toward the flow direction in vISVs (Fig. 4j, k). These data suggests that adequate endothelial cell polarization to flow and subsequent ventral-dorsal endothelial cell migration are required for tertiary sprouting as it results in populating the dorsal−neurovascular interface−domain of the vISV with sufficient *aplnra* expressing cells.

## Tertiary sprouting requires neural Apelin

CellChat predicted interaction between neural Apelin (*apln*) and vascular Apelin-receptor-a (*aplnra*) (Fig. 2e, f) and *apln* expression was significantly elevated in *flt1*[−/−] mutants (Fig. 5a). Analysis of single cell data and reporter embryos showed that *apln* was expressed in spinal cord neural cells (Fig. 5c–e; Supplemental Fig. 10a) and vascular endothelium (Fig. 5b, d, e; Supplemental Fig. 10a). Low dose morpholino mediated knockdown of *apln* in *flt1*[−/−] mutants significantly reduced tertiary sprouting (Fig. 5f, g), and vISV EC number (Fig. 5i) without affecting the aISV/vISV ratio (Fig. 5h). Furthermore, neural specific loss of *apln* in *flt1*[−/−] significantly reduced ectopic vessel formation at the level of the spinal cord (Fig. 5l, m; Supplemental Fig. 10b, c). Conversely, neural specific overexpression of *apln* in *flt1*[−/−] mutants significantly augmented ectopic sprouting (Fig. 5j, k). These data suggest that neural *apln* acts as a positive regulator of tertiary sprouting propensity.

These data furthermore predict that ectopic co-expression of *apelin* and *vegfa* may affect vascular branching. Indeed, overexpression of *apln* in the developing somite, an area well known to express *vegfa*, altered the patterning of trunk ISVs (Supplemental Fig. 10d–f). Apelin receptors can bind Apelin and Apela (*apela*, also known as elabela, or toddler) and *apela* is implied in axial vessel morphogenesis[49]. *Apela* was hardly detectable in neural cells or endothelial cells of 3 dpf WT and *flt1*[−/−] embryos (Supplemental Fig. 5a). Neural specific overexpression of *apela* in *flt1*[−/−] had no impact on tertiary sprouting but instead induced tiny sprout like structures emanating from aISVs in a limited number of embryos (Supplemental Fig. 10g–j). Neural specific overexpression of *apela* in WT had no obvious impact on vascular remodeling (Supplemental Fig. 10k–m).

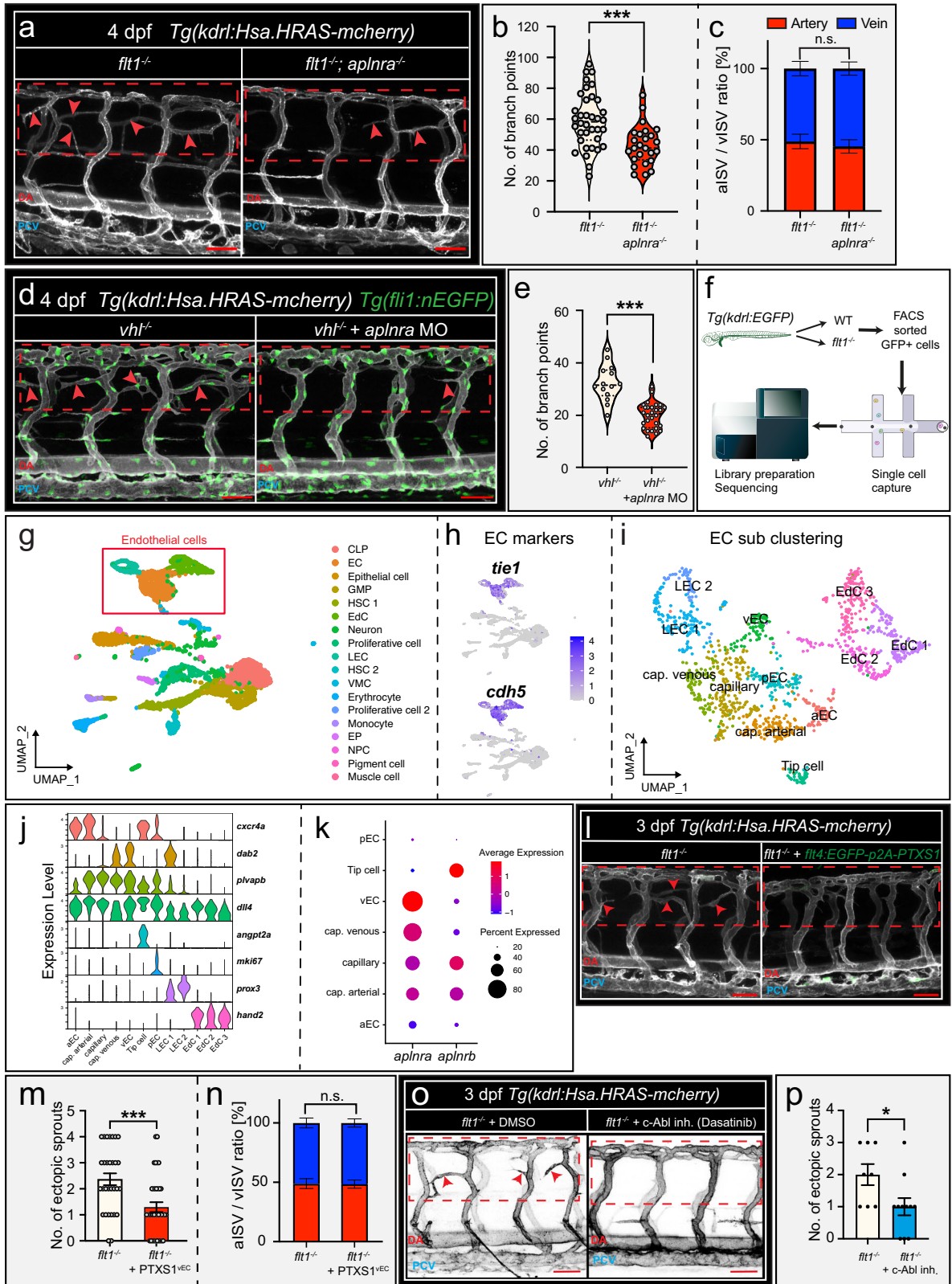

Taken together, we conclude that activation of Apelin−Aplnra signaling, via genetic interaction with Kdrl signaling, aids to augment Vegfa signaling strength and induce tertiary sprouting (Fig. 5n). Neural cells produce Apelin and Vegfa, and the juxtaposed venous endothelial cells express Apelin-receptor-a and Kdrl. Hence, it is logical that sprouts develop in proximity of the anatomical position where ligand−

receptor interaction takes place, which is the neurovascular interface in the dorsal domain of the vISV (Fig. 5n, Supplemental Fig. 15).

**Trio mediates L-Tip cell morphogenesis in tertiary sprouts**

Cell enlargement is a hallmark of L-Tip cell morphogenesis. In zebrafish, an increase in endothelial cell size has been linked to the activity of

**Fig. 3 | Tertiary sprouting upon Vegf gain of function requires *aplnra*.**
**a** Confocal images showing trunk vasculature in *flt1*⁻/⁻ and *flt1*⁻/⁻; *aplnra*⁻/⁻ double mutant at 4 dpf. Dotted box indicates ROI; arrowheads indicate ectopic vessels.
**b** Quantification of branching complexity for indicated scenario. Violin plot shows median and interquartiles; two-sided Unpaired *t* test with Welch's correction, *flt1*⁻/⁻: *n* = 36 and *flt1*⁻/⁻; *aplnra*⁻/⁻: *n* = 26; *p* = 0.0001. **c** Arterial to venous ISV ratio for indicated genotype. Mean±s.e.m.; two-sided Fisher's exact test. flt1: *n* = 36, flt1; aplnra⁻/⁻: *n* = 26. **d** Confocal images showing trunk vasculature in *vhl*⁻/⁻ mutant injected with control or *aplnra* targeting morpholino at 4 dpf. Dotted box indicates ROI, Arrowheads indicate ectopic vessels. **e** Branching complexity for indicated scenario. Violin plot showing median and interquartiles; two-sided Unpaired *t* test, control MO: *n* = 14; *aplnra* MO: *n* = 28; *p* < 0.0001. **f** Single cell sequencing pipeline. GFP+ cells were sorted from *Tg(kdrl:EGFP)* embryos, captured, bar-coded and sequenced. **g** UMAP plot of GFP+ sorted cells. Square indicates endothelial cell clusters. **h** UMAP plot of GFP+ sorted cells, color coded by expression level for *tie1* and *cdh5*. **i** UMAP plot of EC subclusters. **j** Violin plot showing the expression of

marker genes used to identify the individual EC subclusters. **k** Dot-plot showing *aplnra* and *aplnrb* expression for indicated endothelial subclusters. **l** Imaging of trunk vasculature in *flt1*⁻/⁻ and *flt1*⁻/⁻ injected with *flt4-GFP-p2A-PTXS1*. Dotted box indicates ROI; arrowheads indicate ectopic vessels. **m** Quantification of ectopic sprouts for indicated scenario. Mean±s.e.m., two-sided Mann-Whitney U test, *flt1*⁻/⁻: *n* = 30; *flt1*⁻/⁻+*flt4-GFP-p2A-PTXS1*: *n* = 37; *p* = 0.0006. **n** Arterial to venous ISV ratio for indicated scenario. Mean±s.e.m. Two-sided fisher's exact test. *flt1*⁻/⁻: *n* = 30; *flt1*⁻/⁻+*flt4-GFP-p2A-PTXS1*: *n* = 37. **o** Confocal images showing trunk vasculature in *flt1*⁻/⁻ mutant treated with DMSO or with Dasatinib at 3 dpf. **p** Quantification of ectopic sprouts for the indicated scenario. Mean±s.e.m.; two-sided Mann-Whitney *U* test, *flt1*⁻/⁻ + DMSO: *n* = 8, *flt1*⁻/⁻ + Dasatinib: *n* = 11; *p* = 0.0393. Scale bars indicate 50 μm in **a**, **d**, **l**, **o**. MO morpholino, EC endothelial cells, LEC lymphatic endothelial cell, vEC venous endothelial cell, cap. venous capillaries venous, cap. arterial capillaries arterial, aEC arterial endothelial cell, pEC proliferating EC, EdC endocardial cell, PTXS1 Pertussis-Toxin subunit 1, DA dorsal aorta, PCV posterior cardinal vein. Source data are provided as a Source Data file.

the guanine nucleotide exchange factor Trio acting on the small RhoGTPase Rac1[28]. *Trioa* was significantly upregulated in *flt1*⁻/⁻ mutant embryos (Fig. 6a), and *trioa* was expressed prominently in the venous domain (Supplemental Fig. 11a). Single cell sequencing analysis confirmed co-expression of *trioa* with *kdrl* and *rac1* in vascular endothelial cells (Supplemental Fig. 11b). Low dose morpholino mediated knockdown of *trioa* impaired tertiary sprout remodeling by significantly reducing sprout volume and surface area (Fig. 6b–e, Supplemental Fig. 11c). Selective pharmacological inhibition of the GEF1 domain of Trio, the domain relevant for activating Rac1, using ITX3—an established Trio inhibitor in zebrafish[28]—caused similar vascular phenotypes as observed upon loss of *trioa* (Fig. 6f–i; Supplemental Fig. 11d). PI3K acts downstream of Kdrl signaling and links Kdrl with activation of Trio[28]. Accordingly, pharmacological inhibition of PI3K using LY294002 significantly reduced tertiary sprout remodeling (Supplemental Fig. 11e, f). Conversely, venous specific overexpression of GEF1 domain containing *TrioN*[28] (a human Trio deletion construct that lacks the GEF2 domain[28]) in *flt1*⁻/⁻ increased the surface area of the ectopic vascular networks (Fig. 6j–l). Overexpression of *TrioN* in WT vISVs increased lumen diameter without inducing ectopic venous sprouting (Supplemental Fig. 11g, h). The GEF1 domain of Trio is required for activating Rac1[28] in zebrafish and accordingly pharmacological inactivation of Rac1 between 2.5–3.5 dpf in *flt1*⁻/⁻ embryos significantly reduced tertiary sprout remodeling and anastomosis formation at the level of the spinal cord (Fig. 6m, n). These data suggest that Trio is required for L-Tip cell morphogenesis and remodeling downstream of Vegf-PI3kinase in venous endothelium (Supplemental Fig. 11i).

### Tertiary sprouting requires parenchymal Esm1
Analysis of zebrafish mutants with a Vegfa gain of function scenario furthermore showed a significant upregulation of *endothelial cell specific molecule-1* (*esm1*) (Fig. 7a). Esm1 mechanism of action is poorly understood but in vitro data show that Esm1 competes with Vegfa for binding to fibronectin in the extracellular matrix (ECM). Increases in *esm1* consequently displace Vegfa from binding to the ECM, thereby augmenting Vegfa bio-availability and Kdrl signaling (Fig. 7r)[33]. Single cell sequencing (Supplemental Fig. 12a) and analyses of *esm1 BAC*-reporter construct injected embryos revealed *esm1* expression in the developing trunk vasculature, somites, and spinal cord neural cells (Fig. 7b, c). CrisprCas9 *esm1* mutants and *esm1* morphants looked phenotypically normal, and primary artery, secondary venous sprouting and aISV/vISV remodeling were not affected (Supplemental Fig. 12b–j). In contrast, genetic ablation of *esm1* or morpholino mediated loss of *esm1* significantly reduced tertiary sprouting events in all Vegfa gain of function scenarios investigated (Fig. 7d–k; Supplemental Fig. 12k–n). In *flt1*⁻/⁻; *esm1*⁻/⁻ double mutants (Fig. 7g, h), in *vhl*⁻/⁻ mutants injected with *esm1* targeting morpholino (Fig. 7j, k), in *flt1*⁻/⁻ mutants injected with *esm1* ATG targeting morpholino (Supplemental

Fig. 12k, l), or in *esm1*⁻/⁻ mutants injected with a *flt1* ATG targeting morpholino (Supplemental Fig. 12m, n) tertiary sprouting was reduced when compared to *flt1*⁻/⁻ mutant or *flt1* morphant control respectively. Loss of *esm1* had no impact on the aISV to vISV ratios in any of these scenarios (Fig. 7i; Supplemental Fig. 12l). In *flt1*⁻/⁻;*esm1*⁻/⁻ double mutants, remaining sprouts were significantly smaller when compared to *flt1*⁻/⁻ single mutant (Supplemental Fig. 12r, s). Taken together *esm1* acts specifically on tertiary sprouting at the level of the spinal cord and is, during this developmental stage, dispensable for other trunk vascular remodeling events including Vegfa-driven primary artery sprouting.

We next examined if overexpressing *esm1* could augment venous sprouting (Fig. 7l, m; Supplemental Fig. 12p, q). Neural specific overexpression of *esm1* (*esm1^NC*) in *flt1*⁻/⁻ mutants significantly augmented tertiary sprouting (Fig. 7l–right panel, quantification in 7m). Vascular specific overexpression of *esm1* (*esm1^EC*) had no significant effect on sprouting propensity (Fig. 7l–middle panel, quantification in 7m). Neural specific overexpression of *esm1* in WT failed to induce ectopic sprouting or affect trunk aISV/vISV remodeling (Supplemental Fig. 12p, q). Since *flt1*⁻/⁻;*esm1*⁻/⁻ double mutants lacked tertiary sprouting, we next assessed whether re-expressing *esm1* in the neural or vascular domain could restore/rescue tertiary sprouting. We found that neural specific overexpression of *esm1* (*esm1^NC*) in *flt1*⁻/⁻; *esm1*⁻/⁻ double mutants restored tertiary sprouting and anastomosis formation at the level of the spinal cord (Fig. 7p, q). Vascular specific overexpression of *esm1* (*esm1^EC*) did not rescue sprouting defects (Fig. 7n, o) but slightly augmented ISV diameter (Supplemental Fig. 12o). Taken together, these data are consistent with parenchymal *esm1* acting as an essential regulator of Vegfa-driven tertiary sprouting most likely by augmenting Vegf availability at the neuro-vascular interface (Fig. 7r).

### *Flt1, esm1* and *aplnra* contribute to spinal cord vascularization in WT
In WT, spinal cord vascularization starts in the period day 12–14, and similar to tertiary sprouting, is believed to be triggered by a local Vegfa gain of function scenario induced by neural cells[29,30]. This raises the question to what extent the cellular and molecular cues that govern tertiary sprouting also contribute to spinal cord vascularization in WT, day 12–14. We examined the cellular properties of the ISV derived sprouts in the day 12–14 WT scenario and found that the majority of ISV sprouts that vascularized the WT spinal cord, emanated from ISVs that expressed the venous marker *flt4* or *CoupTFII*, in line with venous sprouting events contributing to spinal cord vascularization during this developmental stage (Fig. 8a–e). We furthermore observed sprouts expressing the arterial marker *flt1*, emanating either from arterial ISVs or the vertebral artery (VTA) (Fig. 8b, d). This indicates that the WT spinal cord is vascularized by both arterial and venous

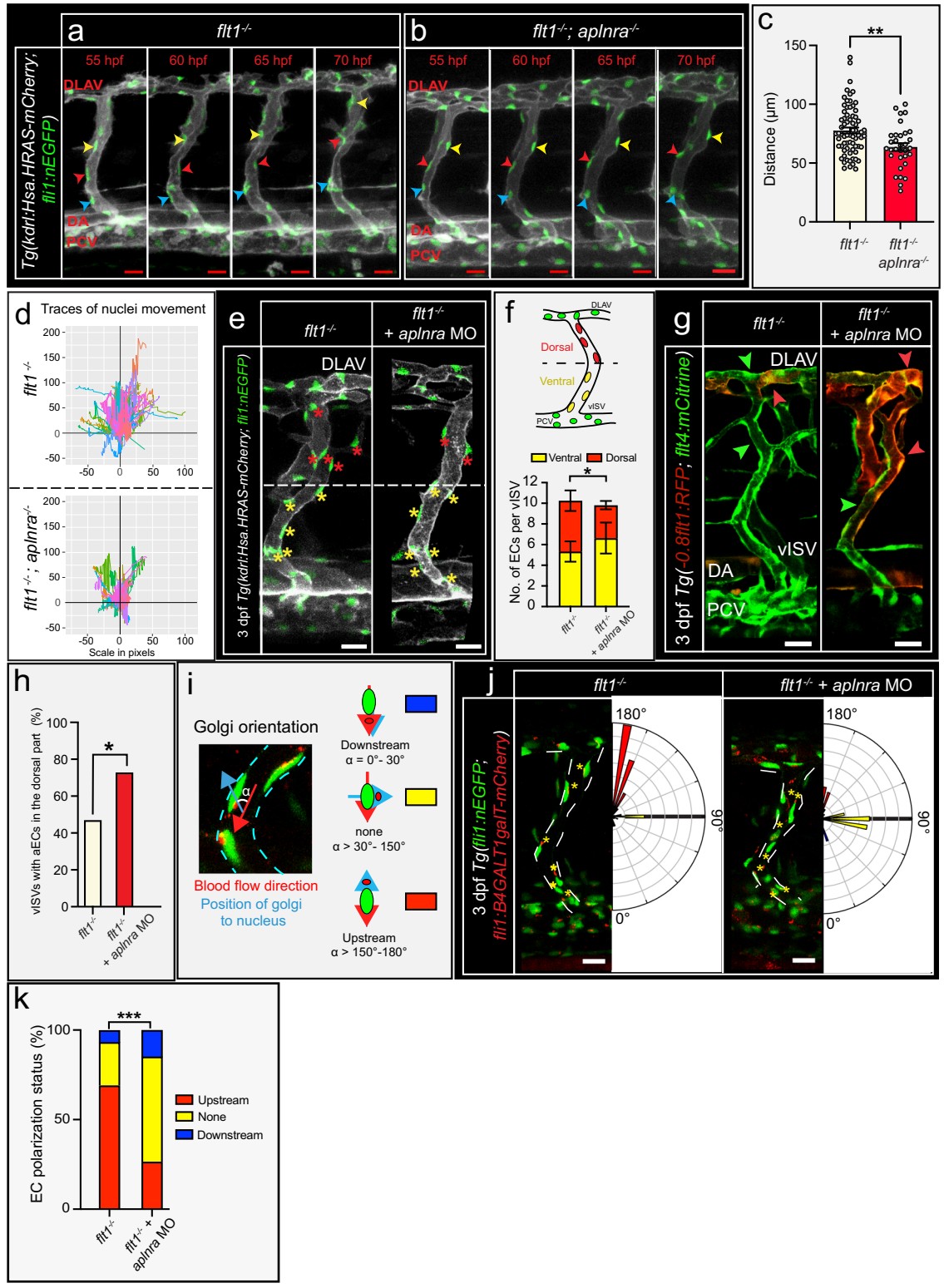

sprouting events. The VTA is an arterial vessel that is anatomically lacking at day 3 of development. Both ISV and VTA derived sprouts consisted on average of between 1 and 2 endothelial cells (Fig. 8f, g). Labelling blood plasma with FITC-dextran to detect lumenized sprouts revealed that in the WT scenario 67% of the ISV derived sprouts were lumenized, whereas VTA derived sprouts were lumenized in about 40% of cases (Fig. 8h, i). Detailed morphological analyses of VTA derived sprouts revealed a population that was short and predominantly lacking a lumen, and a second population that was elongated and

predominantly lumenized (Supplemental Fig. 13a, b). While the VTA is considered arterial, some parts of the VTA expressed both arterial and venous markers, and these regions with a mixed AV identity also gave rise to sprouts (Supplemental Fig. 13c–e). This leaves open the possibility that sprouts derived from this VTA region tended to behave more like a vein, favoring lumenization.

We next addressed whether the signaling molecules involved in tertiary sprouting also contribute to spinal cord vascularization in day 12–14 WT. We found that loss of *flt1* augmented sprouting propensity

**Fig. 4 | Aplnra is required for ventral-dorsal migration of EC in venous ISV.**
**a, b** Confocal images showing endothelial cell movements in *flt1*⁻/⁻ (**a**) and *flt1*⁻/⁻;
*aplnra*⁻/⁻ double mutant (**b**) at indicated time points. Arrowheads indicate EC nuclei
and migration trajectory in the vISV. **c** Quantification of endothelial migration
distance in vISV for indicated scenario. Mean±s.e.m.; two-sided unpaired *t* test with
Welch's correction, *flt1*⁻/⁻: *n* = 70 ECs (2–3 per vISVs), *flt1*⁻/⁻; *aplnra*⁻/⁻: *n* = 32 ECs (2–3
per vISV); *p* = 0.0030. **d** Traces of individual endothelial cell movements in *flt1*⁻/⁻
(top) and *flt1*⁻/⁻; *aplnra*⁻/⁻ double mutant (bottom). *flt1*⁻/⁻: *n* = 70 ECs, *flt1*⁻/⁻; *aplnra*⁻/⁻:
*n* = 32 ECs. **e** Confocal images showing EC distribution in vISVs of *flt1*⁻/⁻ mutant (left)
and *flt1*⁻/⁻ upon loss of *aplnra* (right) at 3 dpf. Asterisks mark ECs in the dorsal (red)
and ventral (yellow) domain. **f** Quantification of dorsoventral distribution of ECs for
indicated scenario. Mean±s.e.m.; two-sided Fischer's exact test; *flt1*⁻/⁻: *n* = 123 ECs
from 13 vISVs, *flt1*⁻/⁻+*aplnra* MO: *n* = 108 ECs from 12 vISVs; *p* = 0.0223. **g** Confocal
images of *Tg(−0.8flt1:RFP;flt4:mCitrine)* to visualize ECs with arterial (red arrow-
head) or venous (green arrowhead) identity in vISVs for *flt1*⁻/⁻ (left) and *flt1*⁻/⁻;

*aplnra*⁻/⁻ double mutant (right) at 3 dpf. **h** Percentage ECs with arterial identity in
the dorsal domain of vISVs. Mean; two-sided Mann–Whitney *U* test; *flt1*⁻/⁻: *n* = 32
vISVs from 16 embryos; *flt1*⁻/⁻; *aplnra*⁻/⁻: *n* = 66 vISVs from 33 embryos; *p* = 0.0148.
**i** Confocal images of *Tg(fli1:nEGFP;fli1:B4GALT1galT-mCherry)* showing EC nuclei
(green), and position of golgi (red) regarding blood flow direction (red arrow)
indicated as angle α. For α = 150-180, polarized against flow-direction; for α = 0–30,
polarized in flow-direction. **j** Confocal images of *Tg(fli1:nEGFP;fli1:B4GALT1galT-
mCherry)* and diagrams showing polarization status of individual ECs in vISV for
*flt1*⁻/⁻ mutant (left) and *flt1*⁻/⁻ upon loss of *aplnra* (right). **k** EC polarization for
indicated scenario. Mean; Chi-square test, *flt1*⁻/⁻: *n* = 165 vECs from 21 embryos, *flt1*⁻/⁻
+ *aplnra* MO: *n* = 196 vECs from 19 embryos, *p* < 0.0001. Scale bars indicate 20 µm
in **a, b, e, g, j**. DA dorsal aorta, PCV posterior cardinal vein, DLAV dorsal lateral
anastomotic vessel, MO morpholino, vISV venous intersegmental vessel. Source
data are provided as a Source Data file.

in the period day 12–14 when compared to age-matched controls
(Fig. 9a, b). In contrast, *esm1*⁻/⁻ and *aplnra*⁻/⁻ mutants showed a sig-
nificantly reduced sprouting propensity during vascularization of the
day 12–14 spinal cord when compared to age-matched controls
(Fig. 9c, d, i, j). *Flt1*⁻/⁻ mutants at day 14 displayed significantly increased
ISV derived sprout numbers and larger sprout lumen diameters when
compared to WT (Fig. 9a, b, e, f); VTA sprout numbers were unchanged
(Supplemental Fig. 13f). Conversely, *esm1*⁻/⁻ mutants showed sig-
nificantly reduced ISV derived sprout numbers and sprout lumen
diameter (Fig. 9c, d, g, h); and slightly reduced VTA numbers (Sup-
plemental Fig. 13g). Collectively our data suggest that several mole-
cular and cellular elements of tertiary sprouting mediated spinal cord
vascularization are recapitulated during spinal cord vascularization as
occurs in day 12–14 WT, including sprout lumenization and the con-
tribution of *aplnra* and *esm1* in mediating sprouting propensity.
However, the spinal cord vascularization processes are not completely
identical, in particular with respect to the contribution of arterial
sprouting events.

## Discussion

Heterogeneity in the vascular branching pattern has been demon-
strated across organs and developing tissues and understanding the
molecular mechanism accounting for this variability is considered
relevant for designing tailored pro- and anti-angiogenic therapies to
combat ischemic cardiovascular and neurodegenerative diseases, and
cancer[2,3]. Vegfa is the key-regulator of angiogenesis in almost all
organs, yet unclear is how a single growth factor can give rise to a
multitude of highly diverse vascular branching architectures. Based on
analysis of vascular remodeling in a series of mutants with a Vegfa gain
function scenario we show that parenchymal cues instructing specific
sprouting competent endothelial cells may account for variability in
Vegfa-driven sprouting angiogenesis and thereby contribute to
building organo-typically ordered vascular architectures. In support of
parenchymal cues controlling competent endothelial cells, we show
that in mutants with a Vegfa gain of function scenario, neural Vegfa and
neural Apelin activate vascular Kdrl–Apelin-receptor-a signaling in a
subset of venous endothelial cells to drive a novel cellular and mole-
cular distinct venous sprouting form termed tertiary sprouting. We
furthermore provide evidence showing that genes controlling tertiary
sprouting are required for vascularization of the spinal cord in WT
larvae.

Tertiary sprouting involves the selection of a lumenized tip cell
(L-Tip cell) in the dorsal aspect of venous ISVs, at the interface with the
spinal cord neural network. The L-Tip cell is prefabricated with an
intact lumen, stabilized by pericyte coverage and forms a long-range
unicellular anastomotic connection with the lateral arterial ISV. To this
end the L-Tip cell increases in size which is achieved by activating the
guanine exchange factor Trio. Trio acts downstream of endothelial
Vegfa signaling and activates the small RhoGTPase Rac1 specifically in

endothelial junctional regions to augment acto-myosin tension pro-
moting endothelial cell enlargement[28]. Loss of Trio function inhibits
L-Tip cell enlargement indicating that Trio driven changes in endo-
thelial cell shape are important for the early stages of venous sprout
remodeling.

Tertiary sprouting requires endothelial cells with a venous iden-
tity, consequently, genetically creating a trunk consisting of pre-
dominantly arterial endothelial cells reduces tertiary sprouting
propensity. In venous ISVs, PCV derived endothelial cells polarize to
the flow direction and subsequently migrate from ventral to dorsal
toward the neuro-vascular interface to replace the remodeled artery
derived EC that have initially populated this area during early stages of
trunk ISV remodeling[48]. We show that restricting the migratory beha-
vior of venous endothelial cells either by interfering with vISV EC
polarization or reducing blood flow, prevents colonization of the
neuro-vascular interface with PCV derived venous endothelial cells and
impairs the tertiary sprouting process. *Kdrl* is expressed in all vascular
endothelial cells, not specifically in veins, and therefore to obtain
venous specificity of the Vegfa-driven venous sprouting process,
endothelial Kdrl has to genetically interact with Apelin-receptor-a, a
receptor that is particularly enriched in venous endothelial cells. The
ventral-dorsal movements as observed in venous ISVs may thus be
required for replacing the remodeled artery derived EC with *aplnra*
expressing venous endothelial cells derived from the posterior cardi-
nal vein. At the spinal cord neuro-vascular interface level, exposure of
these *aplnra* expressing venous endothelial cells to specific par-
enchymal derived cues, in this scenario neural Apelin and Vegfa, may
subsequently induce the venous sprouting process.

The Vegfa induced venous sprouting process involves genetic
interaction between Apelin-receptor-a and Kdrl signaling in venous
endothelial cells, and loss of *aplnra* reduces tertiary sprouting in all
Vegfa gain of function scenarios investigated. *Aplnra* appears specific
for mediating tertiary sprouting during this developmental stage, and
loss of *aplnra* does not interfere with primary artery or secondary
venous sprouting in the trunk. We provide evidence showing that the
genetic coupling between Apelin-receptor-a and Kdrl may be achieved
via Gαi and c-Abl, analogue to a mechanism previously shown to
couple S1PR1 with KDR signaling output in human cells[47]. From a
physiological point of view, this genetic interaction may serve to
potentiate endothelial Vegfa signaling strength and sprouting in a
spatial highly specific manner, namely only in the domain where both
parenchymal cues, Apelin and Vegfa, are co-expressed. Furthermore,
the migrating endothelial cells that later will contribute to tertiary
sprouting are initially polarized to flow. The sprout however expands
along the spinal cord, a structure that is oriented perpendicular to the
flow direction. Hence, for endothelial cells to contribute to the sprout,
they have to adapt their polarization from flow driven toward par-
enchymal derived cues. Parenchymal cues and inducing a strong local
−spatially confined -endothelial Vegfa signal may help these

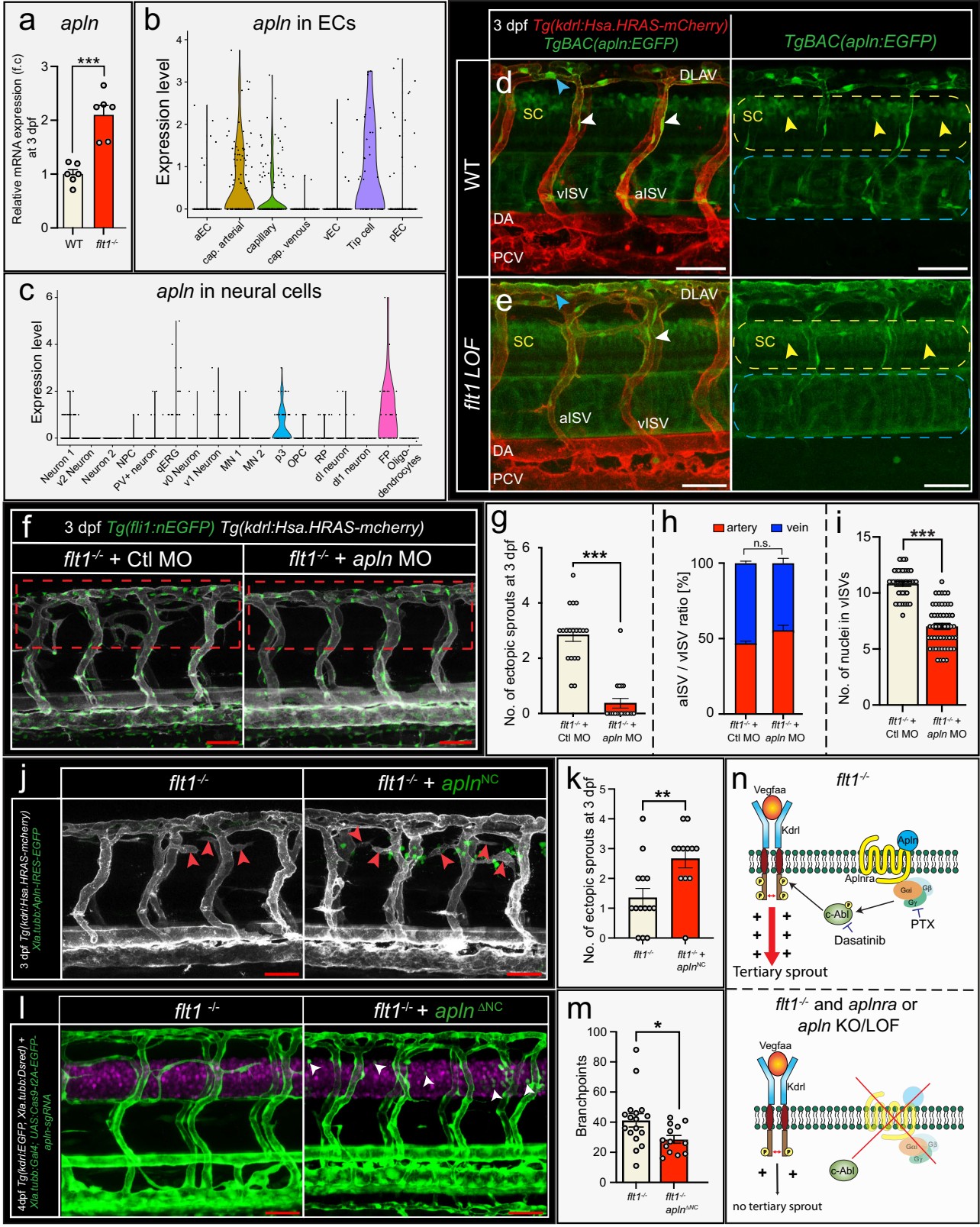

endothelial cells to escape from polarization dictated by flow, and allow extension toward the parenchyma.

Our data furthermore suggest that tertiary sprouting requires a functional contribution of neural *esm1*. In mouse, *Esm1* marks endothelial tip cells and endothelial specific *Esm1* mouse mutants show delayed outgrowth of the neonatal retinal plexus without defects in tip-stalk cell differentiation[33]. The precise mechanism of action of Esm1 is not well established but in vitro studies using human endothelial cells suggest that ESM1 competes with VEGFA for binding with the ECM component fibronectin[33]. This predicts that increases in ESM1 result in displacing VEGFA from the ECM, and thereby augmenting VEGFA availability. Conversely, loss of *ESM1*, allows for more VEGFA binding to the ECM, effectively reducing VEGFA availability[33]. In zebrafish embryos in vivo, *esm1* is expressed in vascular cells and neural cells and

**Fig. 5 | Tertiary sprouting requires neural Apelin and vascular Apelin-receptor-a signaling in venous EC. a** Relative mRNA expression of *apln* for indicated scenario at 3 dpf. Mean±s.e.m.; two-sided Unpaired *t* test with Welch's correction; *n* = 6 samples of 20 pooled embryos per condition; *p* = 0.0007. **b, c** Violin Plot showing *apln* expression for indicated EC (**b**) and neural cell (**c**) cluster. **d, e** Imaging of *Tg^BAC^(apln:EGFP)* reporter in the trunk of WT (**d**) and *flt1* LOF embryos (**e**) at 3 dpf. Dotted line indicates spinal cord (SC; yellow) and notochord (blue) domain. *Apelin* expression detected in ISVs (white arrowheads), DLAV (blue arrowheads), and spinal cord neural cells (yellow arrowheads). **f** Confocal imaging of trunk vasculature and endothelial nuclei (green) in *flt1^-/-^* mutant injected with control (left) or *apln* (right) targeting morpholino at 3 dpf. Dotted box indicates ROI. **g** Quantification of ectopic sprouts for indicated scenario. Mean±s.e.m.; two-sided Mann–Whitney *U* test; *n* = 19 embryos/condition; *p* < 0.0001. **h** Arterial to venous ISV ratio for indicated scenario. Mean±s.e.m.; Two-sided Fisher's exact test; *n* = 5 embryos/condition. **i** Number of nuclei in vISVs for indicated scenario. Mean ± s.e.m.; two-sided Mann–Whitney *U* test; Control MO: *n* = 41 vISVs; *apln* MO: *n* = 53 vISVs; *p* < 0.0001. **j** Confocal images showing trunk vasculature in *flt1^-/-^* embryo (left) and *flt1^-/-^* combined with neuronal specific overexpression of *apln* (right, indicated as *apln^NC^*; *apln* expressing cells in green) at 3 dpf. Arrowheads indicate ectopic sprouts. **k** Quantification of ectopic sprouting for indicated scenario. Mean ±s.e.m.; two-sided Mann–Whitney *U* test; *flt1^-/-^*: *n* = 14, *flt1^-/-^* + *apln^NC^*: *n* = 12; *p* = 0.0046. **l** Confocal images showing trunk vasculature in *flt1^-/-^* mutant (left) and *flt1^-/-^* upon neuronal specific silencing of *apln* (right, indicated as *apln^ΔNC^*). **m** Quantification of branching complexity for indicated scenario. Mean±s.e.m.; two-sided Unpaired t-test with Welch's correction; *flt1^-/-^*: *n* = 17, *flt1^-/-^* + *apln^ΔNC^*: *n* = 13; *p* = 0.0213. **n** Schematic illustration of proposed interaction between Apelin-receptor-a and Kdrl to enhance Kdrl signaling output via Gαi and c-Abl to induce tertiary sprouting. Scale bar indicates 50 μm in **d–f**, **j**, **l**. SC spinal cord, aISV arterial intersegmental vessel, vISV venous intersegmental vessel, DA dorsal aorta, PCV posterior cardinal vein, DLAV dorsal lateral anastomotic vessel, NC neuronal cells, MO morpholino. Source data are provided as a Source Data file.

*esm1* is upregulated in day 3 *flt1^-/-^* mutants. Loss of *esm1* in *flt1* and *vhl* mutants or morphants reduces tertiary sprouting propensity in line with Esm1 acting as a positive regulator of sprouting by increasing Vegf availability. To determine the functional contribution of endothelial *versus* neural *esm1* in mediating tertiary sprouting we introduced *esm1* expression either under control of a vascular or a neural domain specific promoter, into tertiary sprout lacking *flt1^-/-^;esm1^-/-^* double mutants. Re-expressing *esm1* in the neural but not in the vascular domain, rescued tertiary sprouting in *flt1^-/-^;esm1^-/-^* double mutants indicating that neural *esm1* is physiologically relevant for mediating tertiary sprouting. In further support we find that neural specific *esm1* gain of function augments tertiary sprouting to even higher levels in *flt1^-/-^* single mutants. Interestingly, *esm1* is also expressed in developing arterial sprouts but the trunk vascular network of *esm1^-/-^* mutants looks phenotypically normal and shows no defects in primary artery sprouting suggesting that at this stage of development, *esm1* is a marker not a maker of arterial sprouts.

Esm1 acting as a parenchymal cue to determine Vegfa availability and the spatial positioning of venous sprouting upon Vegfa may be another factor contributing to tissue dependent control of vascular architecture. It is established that Vegf levels can be regulated at the level of mRNA transcription through the HiF1α-Vhl pathway[50], mRNA stability[47] or through trapping of Vegf protein by the Vegf scavenging receptor soluble Flt1[29]. In vitro data[33] and our in vivo findings support an additional–hierarchically overarching level of control involving trapping of Vegfa by the ECM, and unlocking of ECM-trapped Vegfa by Esm1 involving an ECM ligand-binding competition model. In the absence of *esm1*, all ECM-Vegfa binding sites will be available to bind Vegfa. The loss of tertiary sprouting observed upon genetically ablating *esm1* suggests that this ECM-Vegfa trapping capacity is sufficiently large to bind enough Vegfa to buffer physiological elevations in Vegfa as occur in *flt1^-/-^* and *vhl^-/-^* mutants, and thereby preclude sprouting. Conversely, to obtain sufficient bio-available Vegfa for venous sprouting to occur in these Vegf gain of function scenarios, ECM binding sites have to be first occupied by Esm1 and this can be achieved by simultaneously expressing *esm1*. Accordingly, *esm1* expression is indeed elevated in *flt1^-/-^* mutants, thus suggesting the existence of a positive feed forward loop to amplify Vegfa levels and promote sprouting. The notion that "excessive" Vegfa resulting for example from the activation of the metabolic HiF1α-Vegf pathway can be "scavenged" by the parenchymal ECM, may have physiological and therapeutic implications especially in settings involving extensive ECM remodeling, as is typically the case during tissue regeneration processes upon ischemic insults[51].

Tertiary sprouting in the venous domain shows considerable molecular, cellular and metabolic differences with Vegfa-driven sprouting in the arterial domain. Primary artery sprouting involves the hierarchical ordering of tip and stalk cells, mediated by Dll4-Notch[9,13]. In this model, tip cells are responsible for vessel guidance, whereas the stalk cells account for lumen formation in the expanding sprout, and loss of *dll4* results in more endothelial cells occupying the tip cell position[9,13,38]. In contrast, in the tertiary sprout, the tip cell is both lumenized and exerts a guidance function and is therefore termed L-Tip cell. Since the expanding tertiary sprout consists of a single cell, there is no need for hierarchical ordering by Dll4-Notch, and accordingly loss of *dll4* has no impact on L-tip cell selection or the ability to form a lumenized anastomotic vessel segment. We furthermore show that loss of Cxcr4-Cxcl12 signaling, a pathway considered to act downstream of Dll4-Notch and responsible for sprout guidance in certain areas of the nervous system[19,40], has no impact on tertiary sprouting. In the trunk, loss of *dll4* promotes venous identity leading to an expansion of the venous ISV domain, hence there are quantitatively more venous endothelial cells from which L-Tip cells can be selected. Accordingly, loss of *dll4* augments tertiary sprout propensity. However, since arterial ISVs are almost lacking upon loss of *dll4*, the ectopic anastomotic segments observed upon gain of Vegfa, primarily connect two lateral vISVs instead of making a vISV-aISV connection. *Apln* is predominantly expressed in arterial endothelial and neural cells, and endothelial cell derived apelin has previously been shown to cell-autonomously regulate primary artery sprouting[42]. Genetic ablation of neural *apln* reduces tertiary sprouting without affecting primary sprouting suggesting that neural derived Apelin is physiologically required for mediating venous sprouting upon Vegfa. A potential physiological contribution of cell autonomous acting venous endothelial *apln* specifically in L-Tip cells, remains to be determined. L-Tip cells of tertiary sprouts are larger in size when compared to other endothelial cell populations, and more glycolysis dependent when compared to primary artery sprouts. Tertiary sprouting requires hemodynamic forces most likely via regulating endothelial cell migration events in venous ISVs and maintaining *aplnra* expression in the venous domain. In contrast, Vegf-Dll4-Notch driven primary artery sprouting, or Vegf-Dll4-Notch mediated sprouting in the neonatal mouse retina angiogenic front, do not require flow[52]. In the mouse retina arterial domain, shear stress exerted by flowing blood has been suggested to repress sprouting to allow flow induced outward remodeling[53]. How such fundamental arterial-venous differences in hemodynamic requirements arise, and how this may affect pharmacological targeting of Vegfa-driven sprouting for pro- and anti-angiogenic strategies, remains to be elucidated.

In WT animals, spinal cord vascularization starts in the period day 12–14 of development and involves angiogenic sprouting from ISVs and the Vertebral Artery (VTA). In WT, spinal cord vascularization is believed to be induced by a simultaneous down-regulation of *sflt1* and upregulation of neural *vegfaa* expression at the neuro-vascular interface collectively creating a local Vegfa gain of function scenario[29,30], a condition that would favor inducing tertiary sprouting. Indeed, tertiary

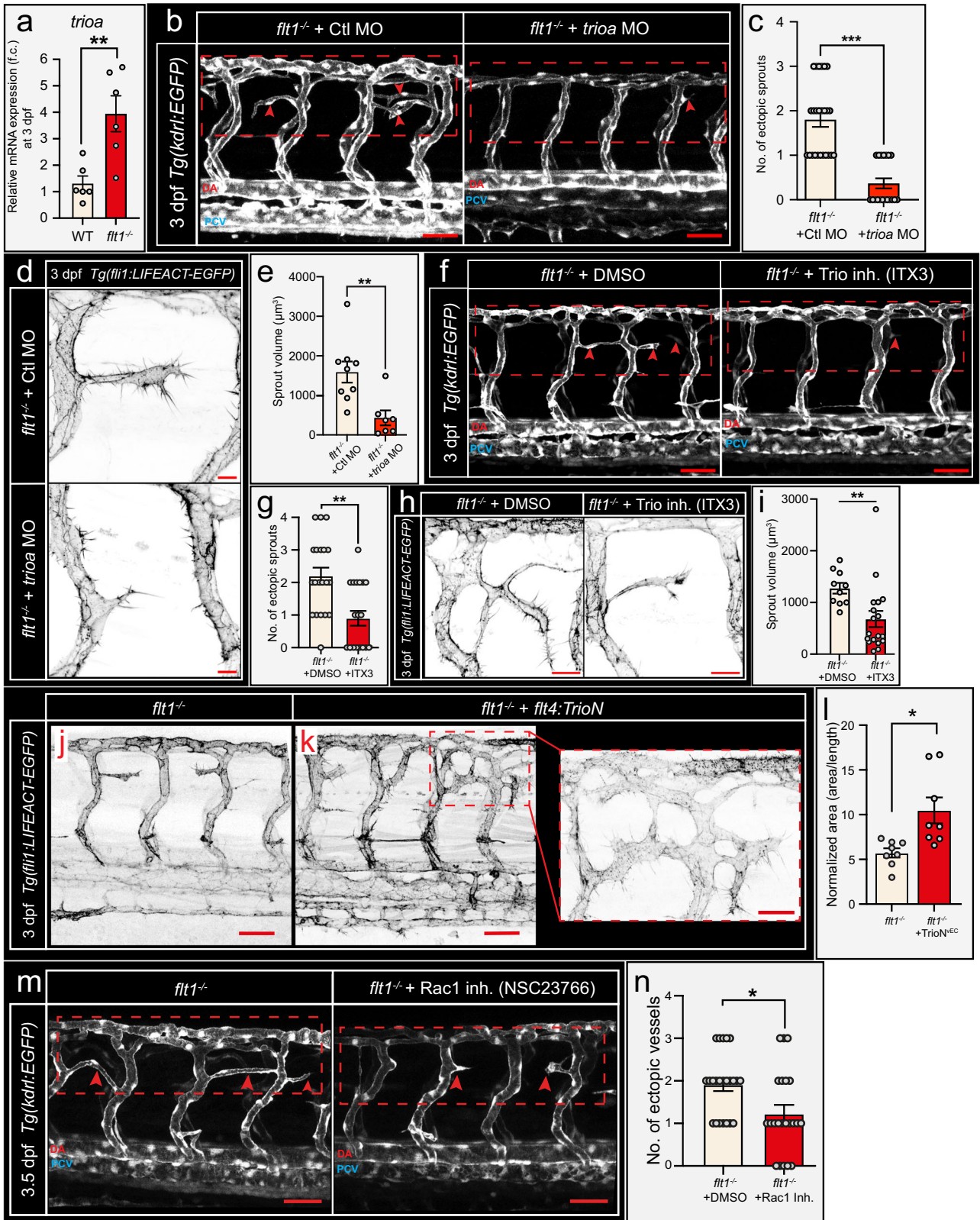

sprouting and spinal cord vascularization in WT show similarities. In both scenarios, *aplnra* and *esm1* act as positive regulators and *flt1* as a negative regulator of sprouting propensity determining the extent of spinal cord vascularization. In WT, spinal cord vascularization involves a significant contribution of lumenized sprouts, to a large extent emanating from venous ISVs. However, besides venous sprouting, the WT spinal cord is also vascularized from the arterial domain, most

prominently the VTA, an arterial vessel that is lacking at day 3-4, the period during which tertiary sprouts arise from veins upon Vegfa. Sprouts emanating from the VTA showed fewer lumenization events when compared to sprouts emanating from ISVs. We conclude that elements of tertiary sprouting are conserved during WT spinal cord vascularization but in the WT, day 12–14 scenario, there is an additional component involving a contribution from the arterial domain.

**Fig. 6 | Trio regulates endothelial morphogenesis in tertiary sprouts. a** Relative mRNA expression of *trioa* for indicated scenario at 3 dpf. Mean±s.e.m.; two-sided Unpaired *t* test with Welch's correction; *n* = 6 samples of 20 pooled embryos; *p* = 0.0095. **b** Confocal images showing trunk vasculature in *flt1*[−/−] embryos injected with control (left) or *trioa* targeting morpholino (right) at 3 dpf. Dotted box indicates ROI. Arrowheads indicate tertiary sprouts. **c** Quantification of ectopic sprouting for indicated scenario. Mean±s.e.m.; two-sided Mann–Whitney *U* test, control MO n = 25, *trioa* MO *n* = 19; *p* < 0.001. **d** Confocal images of *Tg(fli1:LIFEACT-EGFP)* reporter showing tertiary sprouts in *flt1*[−/−] embryos injected with control (top) or *trioa* targeting morpholino (bottom) at 3 dpf. **e** Quantification of sprout volume for indicated scenario. Mean±s.e.m.; two-sided Mann–Whitney *U* test; control MO *n* = 9, *trioa* MO *n* = 7; *p* = 0.0021. **f** Confocal images showing trunk vasculature in *flt1*[−/−] embryos treated with DMSO (left) or ITX3 (right) at 3 dpf. Dotted box indicates ROI. **g** Quantification of tertiary sprout numbers for indicated scenario. Mean ±s.e.m.; two-sided Mann–Whitney *U* test; *n* = 20 embryos per condition; *p* = 0.0011. **h** Confocal images of *Tg(fli1:LIFEACT-EGFP)* reporter showing ectopic sprouts in *flt1*[−/−] mutants treated with DMSO (left) or ITX3 (right) at 3 dpf. **i** Quantification of sprout volume for indicated scenario. Mean±s.e.m.; two-sided Mann–Whitney *U* test; DMSO: *n* = 10 sprouts/embryos, ITX3: *n* = 18 sprouts from 17 embryos; *p* = 0.0021. **j**, **k** Confocal images of *Tg(fli1:LIFEACT-EGFP)* reporter in *flt1*[−/−] embryo (**j**) and upon venous specific overexpression of *TrioN* (*TrioN*[vEC]) in *flt1*[−/−] mutant (**k**) at 3 dpf; Boxed region at higher magnification in the right panel. **l** Normalized ectopic network size (area/length) for indicated scenario. Mean±s.e.m.; two-sided Unpaired t-test with Welch's correction; *n* = 8 ectopic networks/per condition; *p* = 0.0133. **m** Confocal images showing trunk vasculature in vehicle treated (left) and Rac1 inhibitor (right) treated *flt1*[−/−] embryos at 3.5 dpf. Dotted boxes indicate ROI; arrowheads indicate ectopic vessels. **n** Quantification of ectopic vessels for indicated scenario. Mean±s.e.m.; two-sided Mann–Whitney *U* test; DMSO, *n* = 23, Rac1inh, *n* = 23; *p* = 0.0108. Scale bars indicate 50 μm in **b**, **f**, **j**, **k**, **m**; 20 μm in **d**, **h**; 25 μm in **k** crop. MO morpholino, Mdk medaka, inh inhibitor, vEC venous endothelial cell, DA dorsal aorta, PCV posterior cardinal vein. Source data are provided as a Source Data file.

Tertiary sprout formation requires a specific set of parenchymal derived cues that collectively augment Kdrl signaling strength above the threshold necessary for L-Tip cell selection in the venous domain. A potent Vegfa-Kdrl signal may confer robustness on L-Tip cell selection and escape from the initial flow-induced polarization[48] toward a tissue derived growth factor defined orientation, and allowing active angiogenic remodeling. The requirement of two ligands and two receptors allows for a more precise and stringent spatio-temporal control of sprout initiation by the parenchyma when compared to a scenario involving just one ligand/receptor pair. A stringent high threshold may prevent from unproductive venous sprouting in response to any small change in Vegfa expression. Instead, a two-tiered regulation of both Vegfa and Apelin and their respective receptors, may ensure that a sprout is induced only at the exact desired anatomical position in the tissue, the domain where both Vegfa and Apelin are co-expressed. The concept emerging from these observations is that organo-typical features in vascular branching may evolve from the selective coupling of endothelial growth factor receptors and the heterogeneity in the repertoire of parenchymal derived signals that can be used to activate such receptor pairs. We observed coupling of Kdrl and Apelin-receptor-a, but in other scenarios different genetic interactions may exist for example between Kdr/Kdrl/Aplnr with Flt4[54], Neuropilin (Nrp) and Plexin receptors (Plxn)[55–58], allowing myriad combinations with ligands of the Semaphorin and Vegf family. In a reductionist view, as opposed to the Dll4/Notch tango of the reference model, the choreographed dances in the revised organo-typical scenario may rely on up to four ligand/receptors pairs.

Therapeutic revascularization strategies employing Vegfa have traditionally focused on engineering branching and lumen remodeling of the arterial system[59]. Emerging single cell sequencing analyses however point toward a crucial role for endothelial cells derived from the venous domain in shaping the angio-architecture, in particular in the brain[25,27]. Yet these systems biology findings await genetic and functional confirmation. Alterations in venous endothelial behavior have been shown to underlie arterial-venous malformations and contribute to the vascular lesions observed during retinopathies[60,61]. Understanding the molecular framework how parenchymal cues influence Vegfa responsiveness and venous remodeling may help to develop novel therapeutic tools to combat vasculopathies or promote revascularization in ischemic cardiovascular diseases.

## Methods
### Ethics statement
Zebrafish husbandry and experimental procedures were performed in accordance with local and national German animal welfare standards[62] and were approved by the government of Baden-Württemberg,

Regierungspräsidium Karlsruhe, Germany (Akz.: 35-9185.81/G-11/19, 35-9185.81/G-29/22; 35-9185.64/BH KIT).

### Zebrafish maintenance and strains
Embryos were incubated at 28.5 °C and staged by hours or days post fertilization (hpf or dpf, respectively). We used two *flt1* mutant alleles: the *flt1*[ka604], *flt1*[ka602]. The *flt1*[ka604] (exon 3, −14 nt allele) and the *flt1*[ka602] (exon 3, −5 nt allele) have a premature termination codon (PTC) resulting in a truncated protein devoid of a functional extracellular VEGF binding domain; these mutants lack both functional *sflt1* and *mflt1* as previously published[29]. The following zebrafish lines were used as previously described[29,30,49,63–68]: *Tg(kdrl:Hsa.HRAS-mCherry)*[s916], *Tg(fli1:nEGFP)*[y7], *Tg(fli1:EGFP)*[y1], *Tg(Xla.tubb:dsRed)*[zf148], *Tg(kdrl:EGFP)*[s843], *Tg(−0.8flt1:RFP)*[hu5333], *Tg(flt4:mCitrine)*[hu7135], *vhl*[hu2117] *TgBAC(pdgfrb:Gal4FF)*[ncv24], *Tg(5xUAS:EGFP)*[nkuasgfp1a], *Tg(kdrl:TagBFP)*[mu293], *Tg(kdrl:NLS-EGFP)*[ubs1], *TgBAC(apln:GFP)*[bns157], *Tg(fli1:LIFEACT-EGFP)*[mu240], *aplnra*[mu296], *aplnrb*[mu270], *Tg(fli1:PECAM1a-EGFP)*[ncv27], *Tg(CoupTFII-96S:GFP)*[lcr5].

### Generation of mutant and transgenic lines
Zebrafish embryos were injected at the one-cell stage with a glass microneedle and a microinjector (World Precision Instruments). For transgenesis *Tol2* mRNA, transcribed from pCS2FA (a kind gift from Koichi Kawakami) with the mMESSAGE mMACHINE SP6 Transcription Kit was injected at a concentration of 25 ng μl⁻¹ together with Tol2 destination vectors[69]. For mutagenesis, 1 nl of a mixture of Cas9 protein (0.5 μg/uL, IDT) and sgRNA (3 μM, IDT) were co-injected into one-cell stage embryos. The sgRNAs that have been used to create the *esm1* mutant (formally named *esm1*[ka616]) are indicated in Suppl. Table 1. The primers used to genotype the *esm1* mutant are indicated in Suppl.Table 2.

### Generation of p5E entry vectors for mdk.flt4 promotor
For generating the *p5E-mdk.flt4* entry vector, the promotor sequence was amplified from the plasmid *pMiniTol2-mdk-3.8-flt4* (kindly provided by Prof. Stefan Schulte-Merker) and inserted by restriction cloning into an empty p5E vector (Suppl. Table 3).

### Generation of middle entry vectors
The *apln, esm1 and apela* coding sequences were amplified from zebrafish cDNA using the primers indicated in Suppl. Table 4 and cloned into a *pME-EGFP-p2A* plasmid by restriction cloning. The resulting plasmids were named *pME-EGFP-p2A-esm1, pME-EGFP-p2A-apln, pME-EGFP-p2A-apela*, and contained the coding sequences in frame with the EGFP-p2A sequence.

For the PTXS1 experiment, a synthetic DNA plasmid containing a coding sequence for the S1 subunit of the PTX toxin (PTXS1) was ordered to Eurofins Genomics. The coding sequence for the PTXS1 subunit was

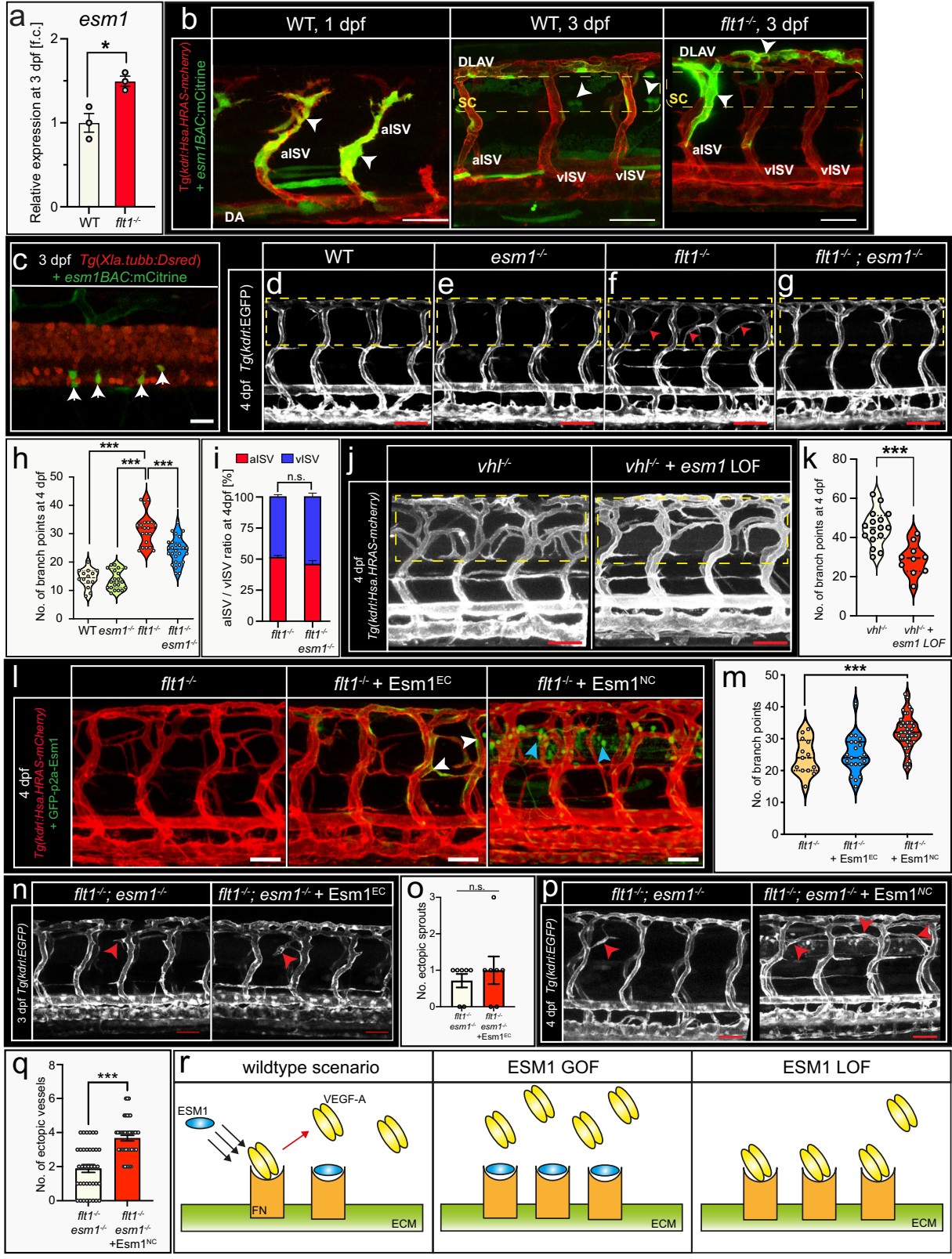

then cloned into a *pME-EGFP-p2A* by restriction cloning (SmaI/XbaI). The resulting plasmid was named *pME-EGFP-p2A-PTXS1*.

## Generation of overexpression constructs

To generate neuronal overexpression constructs for *apln*, *apela*, *esm1* and *gal4vp16*, Gateway reactions were performed using the following plasmids: p5E-Xla.tubb, a pME (*pME-EGFP-p2A-esm1*, *pME-EGFP-p2A-*

*apln*, *pME-EGFP-p2A-apela* or *pME-gal4-vp16*), a *p3E-polyA* and a pDestTol2-Cry-GFP or pDestTol2-CG2-EGFP. The resulting plasmids were named *pDestTol2-CryGFP-Xla.tubb-EGFP-P2A-apln-polyA*, *pDestTol2-CryEGFP-Xla.tubb-EGFP-p2A-esm1-polyA*, *pDestTol2-CryEGFP-Xla.tubb-EGFP-p2A-apela-polyA* and *pDestTol2-CG2-EGFP-Xla.tubb-Gal4vp16-polyA*. Neural specific *apln* and *esm1* gain of function are indicated as *apln*[NC] and *esm1*[NC] respectively.

**Fig. 7 | Tertiary sprouting requires Esm1. a** *Esm1* expression for indicated scenario. Mean±s.e.m.; two-sided unpaired *t* test; *n* = 3 samples of 20 pooled embryos; *p* = 0.0176. **b** Images of WT or *flt1⁻/⁻* embryos injected with *esm1-Bac:mCitrine* construct. Box indicates the spinal cord. Arrowheads indicate *esm1* expression. **c** Image of *esm1* expression (arrowheads) in spinal cord at 3 dpf. **d**–**g** Trunk vasculature in WT (**d**), *esm1⁻/⁻* (**e**), *flt1⁻/⁻* (**f**), and *esm1⁻/⁻;flt1⁻/⁻* embryos (**g**) at 4 dpf. Box indicates ROI. Arrowheads indicate ectopic vessels. **h** Branching complexity for indicated scenario. Violin plot indicates median and interquartiles; two-sided Unpaired *t* test with Welsch's correction; WT: *n* = 14, *flt1⁻/⁻*: *n* = 20, *esm1⁻/⁻*: *n* = 20, *flt1⁻/⁻; esm1⁻/⁻*: *n* = 26. *p* < 0.0001 for all indicated comparisons. **i** Arterial to venous ratio for indicated scenario. Mean±s.e.m.; two-sided Fischer's exact test; *flt1⁻/⁻*: *n* = 20, *flt1⁻/⁻;esm1⁻/⁻*: *n* = 26. **j** Trunk vasculature in *vhl⁻/⁻* and *vhl⁻/⁻* upon *esm1* LOF embryos at 4 dpf. Dotted box indicates ROI. **k** Branching complexity for indicated scenario. Mean±s.e.m.; two-sided unpaired *t* test; *vhl⁻/⁻*: *n* = 16; *vhl⁻/⁻*+*esm1* LOF: *n* = 10. *P* = 0.0001. **l** Trunk vasculature in *flt1⁻/⁻*, *flt1⁻/⁻* upon vascular *esm1* GOF (*esm1^EC*) and *flt1⁻/⁻* upon neuronal

*esm1* GOF (*esm1^NC*) embryos at 4 dpf. Arrowheads indicate *esm1* expression. **m** Branching complexity for indicated scenario. Violin plot showing median and interquartiles; two-sided Unpaired t-test with Welch's correction; *flt1⁻/⁻*: *n* = 13; *flt1⁻/⁻* + *Esm1^EC*: *n* = 21; *flt1⁻/⁻* + *esm1^NC*: *n* = 40. p(*flt1⁻/⁻* vs.*flt1⁻/⁻*+esm1^NC)<0.0001. **n** Trunk vasculature in *flt1⁻/⁻;esm1⁻/⁻* and *flt1⁻/⁻;esm1⁻/⁻* upon vascular *esm1* GOF (*esm1^EC*) embryos at 3 dpf. Arrowheads indicate ectopic sprouts. **o** Ectopic sprout number for indicated scenario. Mean±s.e.m.; two-sided Mann–Whitney *U* test; *n* = 7 per condition. **p** Trunk vasculature in *flt1⁻/⁻;esm1⁻/⁻* and *flt1⁻/⁻;esm1⁻/⁻* upon neuronal *esm1* GOF (*esm1^NC*) embryos at 4 dpf. Arrowheads indicate ectopic vessels. **q** Ectopic vessel number for indicated scenario. Mean±s.e.m.; two-sided Mann–Whitney *U* test, *flt1⁻/⁻;esm1⁻/⁻*: *n* = 37; *flt1⁻/⁻;esm1⁻/⁻* + *esm1^NC*: *n* = 43. *p* < 0.0001. **r** Schematic illustration of Esm1 function. Scale bars indicate 20 μm in **b**; 10 μm in **c**; 50 μm in **d**–**g**, **k**, **l**, **n**, **p**. LOF loss of function, EC, endothelial cell, NC neuronal cell, SC spinal cord, aISV arterial intersegmental vessel, vISV venous intersegmental vessel, DA dorsal aorta, FN Fibronectin, ECM extracellular matrix, GOF gain of function. Source data are provided as a Source Data file.

---

For vascular specific overexpression of *esm1*, a Gateway reaction was performed using the following plasmids: *p5E-fli1, pME-EGFP-p2A-esm1, p3E-polyA* and *pDestTol2-CG2-EGFP*. The resulting plasmid was named *pDestTol2-CG2-EGFP-fli1-EGFP-p2A-esm1-polyA*. For venous overexpression of PTXS1 and hTrioN, Gateway reactions were performed using the following plasmids: *p5E-mdk.flt4, pME-EGFP-p2A-hTrioN*[28] or *pME-EGFP-p2A-PTXS1*, p3E-polyA and pDestTol2-Cry-GFP. The resulting plasmids were named *pDestTol2-CryEGFP-mdk.fl4-EGFP-p2A-hTrioN-polyA* and *pDestTol2-CryEGFP-mdk.fl4-EGFP-p2A-PTXS1*. For somites specific overexpression of apelin, Gateway reactions were performed using the following plasmids: *p5E-unc505, pME-EGFP-p2A-apln, p3E-polyA* and *pDestTol2-Cry-GFP*. The resulting plasmid was named *pDestTol2-unc503-EGFP-p2A-apln-polyA*.

TrioN construction contains the N-terminal part of human TRIO (amino acids 1-1685) including the Sec14 domain, spectrin repeats, the Rac1/RhoG GEF1 domain and SH3 domain, but excluding the RhoA-specific GEF2 domain and kinase domain[28].

### Neuronal specific knock out of *apln*

A plasmid encoding for *Gal4* under a neuronal promoter *(pDest-Tol2CG2-EGFP-Xla.tubb-gal4vp16-polyA)* and a plasmid encoding for *Cas9* under a UAS promoter and two *apln* sgRNAs (*pDestTol2-UAS-Cas9-t2a-EGFP-U6-aplnsgRNA1-U6-aplnsgRNA2*, gift from Christian Helker) were co-injected (25 ng/μl each) with *Tol2* mRNA (25 ng/μl) into the one cell stage zebrafish embryos. Neural specific *apln* loss of function is indicated as *apln^ΔNC*.

### Generation of the *esm1* BAC

The BAC clone *esm1:mCitrine* was generated as previously described[70]. Briefly, the BAC clone CH211-66D12 containing the *esm1* gene was ordered from BPAC Resource Center. The *mCitrine* coding sequence was inserted directly after the ATG site of the *esm1* gene by BAC recombineering. The primers used for the BAC modification and recombination can be found in Suppl. Table 5.

### Morpholino knockdown

The following morpholino antisense oligomers (MOs; Gene Tools) were injected into the yolk of one-cell stage embryos: see Suppl. Table 6 for morpholino sequences and concentrations.

### Inhibitor treatments

Embryos were dechorionated prior to inhibitor treatment. The inhibitors were added at 2.5 dpf followed by incubation at 28.5 °C until the embryos were analyzed between 3 and 4 dpf. For inhibitors dissolved in DMSO, control embryos were mock treated with DMSO (Sigma). Inhibitor concentrations are listed in Suppl.

Table 7. Embryos were randomly assigned to experimental groups.

### Gene expression analysis by real-time qPCR

Total RNA of zebrafish embryo trunks (20 embryos per sample) was isolated with TRIzol and purified with Relias Tissue RNA extraction Kit (Promega) with included DNase step according to the manufacturer's instructions. The quality control was done using electrophoresis and Nanodrop. The RNA concentration was measured by Nanodrop. cDNAs were obtained using Random Primers and MMLV-Reverse Transcriptase (Promega). For qPCR SYBR-Green PCR Mastermix (BioRad) was used and the qPCR was run on a Biorad CFX connect machine using the Biorad CFX Maestro 1.0 software. Gene expression data were normalized to beta-actin and the data are presented as fold change. The primer sequences can be found in Suppl. Table 8.

### Confocal microscopy

Zebrafish larvae were embedded in 0.5% (w/v) low-melting agarose (NuSieve GTG Agarose, Lonza) in 35 mm glass bottom microscopy dishes (MatTek). The agarose was covered with E3 medium and 0.003% (w/v) PTU (Sigma). For the time-lapse, the E3 medium was also supplemented with 0.112 mg ml⁻¹ Tricaine. Confocal t- (for time-lapse imaging) and z-stacks were acquired using a Leica SP8 confocal microscope with ×20 or ×40 water immersion objectives, HyD or PMT detectors and LAS X v3.5.7 software. For the time-lapse, the resonant scanner was also used. Images are displayed as maximum intensity projections of the z-stacks. High resolution microscopy was performed on a Zeiss LSM900 with an Airyscan 2 module, a x40 water immersion objective, PMT detectors and Zeiss Zen v2.3 Software.

### Analysis of volume and surface area

To analyze the EC volume and surface area during primary and tertiary sprouting, images of *Tg(kdrl:Has.HRAS-mcherry; fli1:PECAM1a-EGFP)* embryos (1dpf WT and 3dpf *flt1⁻/⁻*) were acquired. Tip or L-tip cells were manually segmented based on mCherry and EGFP signals using the 'Segmentation Editor' plugin in Image J.

To analyze the tertiary sprout volume and surface area, control or treated 3dpf *flt1⁻/⁻ Tg(fli1:LIFEACT-EGFP)* were imaged. Tertiary sprouts were manually segmented based on EGFP signal using the 'Segmentation Editor' plugin in Image J. Volume and surface area were measured on the segmented object using the '3D Objects counter' plugin in Image J/Fiji 2.3.0.

To analyze the TrioN gain of function experiment, the ectopic vessels were manually segmented on Z-projection. The area of the segmented networks was normalized by the length of the ectopic vessel to obtain the normalized area.

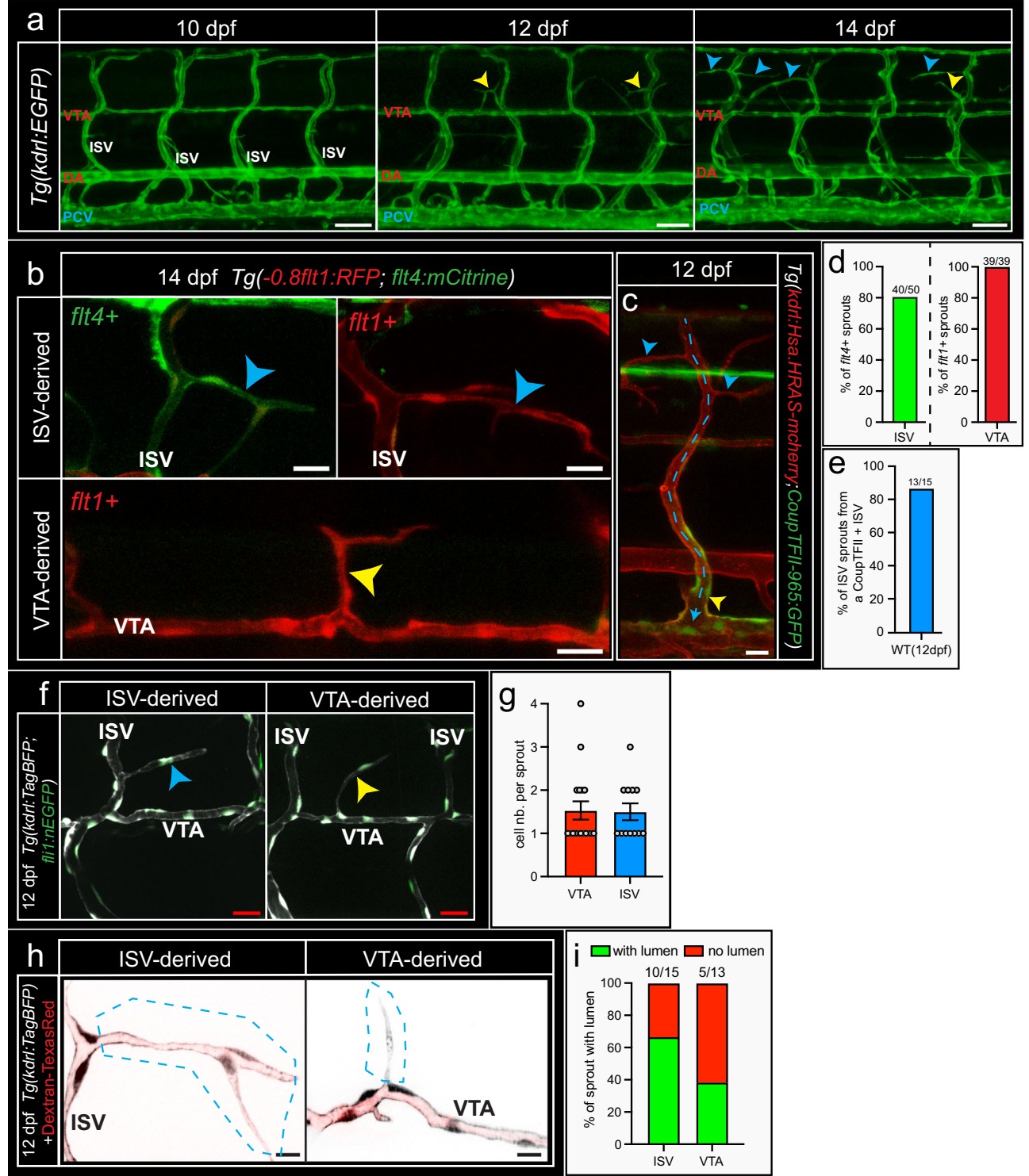

## Vascular network analysis

For 3 dpf embryos, sprout numbers in the trunk were counted manually on 4 ISVs. For 4 dpf embryos, the vascular network complexity in the trunk was assessed as previously described[29]. Briefly, a skeleton representation of the vasculature was generated using the 'skeletonize' plugin in Image J/Fiji 2.3.0. Segment number, branch point number and total branch length were calculated using the 'analyze skeleton' plugin. For 14 dpf embryos, sprout numbers in the trunk were counted manually on 6 ISVs. For the analysis of the hindbrain vasculature, the vasculature was manually segmented and then analyzed using the 'analyze skeleton' plugin.

## Vessel and sprout diameter analysis

Vessel diameters were quantified using the 'VasoMetrics' plugin in ImageJ/Fiji 2.3.0. Plugin was used on Z-stack projection and the diameter means were used for the analysis.

For sprout diameter analysis, lateral lines were drawn at five (14 dpf larvae) or three (3 dpf embryos) evenly distributed positions. The median of these lines was calculated and used for the analysis.

## FACS sorting

Per sample, 100 embryos from *Tg(kdrl:EGFP)* line were collected and dissociated into a cell suspension as described[71]. Briefly, the embryos

**Fig. 8 | Spinal cord vascularization in day 12 WT larvae shows morphological similarities to tertiary sprouting. a** Confocal images showing trunk vasculature at 10, 12 and 14 dpf in WT. Sprouts emanating from ISVs (blue arrowhead) and VTAs (yellow arrowhead). Representative images of 10 embryos per time point. **b** Confocal images of *Tg(−0,8flt1:RFP; flt4:mCitrine)* showing the arterial (*flt1*+, red)− venous (*flt4*+, green) identity of ISV and VTA derived sprouts at 14 dpf. **c** Confocal image showing a venous ISV expressing the venous marker *CoupTFII* in the ventral domain (yellow arrowhead), and two sprouts emanating from the dorsal aspect of this vISV (blue arrowheads) at 12 dpf. Blood flow indicated by dotted arrow. **d** Expression of arterial-venous identity markers *flt1* and *flt4* in ISV (left panel) and VTA (right panel) derived sprouts. Note: the majority of the ISV derived sprouts express the venous marker *flt4,* whereas all VTA derived sprouts express the artery marker *flt1*. Percentage; *n* = 50 sprouts for ISV derived and 39 sprouts for VTA derived from 19 animals. **e** Quantification of the percentage of ISV derived sprouts that emerged from a *CoupTFII* expressing vein. Percentage; *n* = 15 sprouts from 7 animals. **f** Confocal images of *Tg(kdrl:TagBFP;fli1:nEGFP)* showing EC nuclei in ISV (blue arrowhead) and VTA (yellow arrowhead) derived sprouts at 12 dpf. **g** Quantification of nuclei number in ISV and VTA derived sprouts. Mean±s.e.m.; *n* = 17 ISV derived sprouts and 12 VTA derived sprouts from 12 animals. **h** Injection of TexasRed-Dextran into *Tg(kdrl:TagBFP)* at 12 dpf to image blood plasma distribution, shows blood plasma in the ISV derived sprout consistent with the sprout being lumenized. In most VTA derived sprouts, no blood plasma was detected, consistent with absence of lumenization during the early stage of sprout remodeling. **i** Quantification of lumenization in ISV and VTA derived sprouts based on images in **h**. *n* = 15 ISV derived sprouts and 13 VTA derived sprouts from 11 animals. Scale bars indicate 50 μm in **a**; 20 μm in **b**, **c**, **f** and 10 μm in **h**. DA dorsal aorta, PCV posterior cardinal vein, ISV intersegmental vessel, VTA vertebral artery. Source data are provided as a Source Data file.

were dissociated with a mix of 0.25% Trypsin and collagenase (4 mg/mL), centrifuged to collect the cells, resuspended into DMEM + 10% Fetal Bovine Serum and filtered to remove the debris. Cells were then FACS sorted for GFP+ cells and keep on ice until downstream applications.

## scRNAseq

For the analysis of the Kdrl+ cells, following cell dissociation and FACS sorting (described above), samples were processed on the 10X Chromium platform using 10X Single Cell 3' v3 chemistry following the manufacturer's guidelines (10X Genomics, Pleasanton, USA). For each genotype (WT and *flt1*−/−) two samples were prepared and processed for GEM generation, reverse transcription, cDNA amplification and library construction following manufacturer's instructions. The libraries were sequenced on a NextSeq 2000 sequencer. The data were first processed using the 10X Cellranger v6.1.1 pipeline (mkfast and counts) and mapped to the zebrafish genome GRCZ11 v107. The above procedures were carried out as a service by the EMBL GeneCore facility (Heidelberg, Germany).

The resulting matrices were subjected to quality control, normalization and analysis in Seurat[72] version 4.2 package for R, version 4.2.2. Cells were selected to continue with down-stream analysis based on feature count and mitochondrial gene percentage. Only cells that had more than 200 features and fewer than 4500 features expressed were retained. Additionally, only cells that had less than 5 % of their reads from mitochondrial genes were kept. Data were then normalized using 'NormalizeData' function with scale factor 10000, then the top 2000 highly variable genes were selected and scaled to unit variance and zero mean with 'Scale Data'. Principal Component Analysis was then performed on these highly variable genes. RunUMAP and FindNeighbours were run based on the first 20 principal components and unsupervised clustering was performed with FindClusters (resolution = 0.5). Cluster annotation was based on enriched genes in each cluster identified by 'FindMarkers' function. Endothelial cell subclustering was performed as follows: A dataset containing only the EC, LEC and EdC cells was created using the subset function. Data were normalized and scaled as described above. RunPCA, RunUMAP and FindNeighbours were run based on the first 20 principal components and unsupervised clustering was performed with FindClusters (resolution = 1). Cluster annotation was based on enriched genes in each cluster identified by 'FindMarkers' function.

For the analysis of zebrafish trunk, the heads of the embryos were removed prior cell dissociation. Cell dissociation was performed similarly as described above for FACS sorting. Cell suspensions were then diluted at a concentration of 1000 cells per uL and processed on the 10X Chromium platform using 10X Single Cell 3' v3 chemistry following the manufacturer's guidelines (10X Genomics, Pleasanton, USA). For each genotype (WT and *flt1*−/−), 2 samples were prepared and processed for GEM generation, reverse transcription, cDNA amplification and library construction following manufacturer's instructions.

The libraries were sequenced on a NextSeq 2000 sequencer. The data were first processed using the 10X Cellranger pipeline (mkfast and counts) and mapped to the zebrafish genome GRCZ11 v107. The above procedures were carried out as a service by the EMBL Genomics Core facility (Heidelberg, Germany). The resulting matrices were subjected to quality control, normalization and analysis in Seurat version 4.2 package for R, version 4.2.2. Cells were selected to continue with down-stream analysis based on feature count and mitochondrial gene percentage. Only cells that had more than 200 features and fewer than 4000 features expressed were retained. Only genes expressed in at least 5 cells were retained. Additionally, only cells that had less than 5% of their reads from mitochondrial genes were kept. Data were then normalized using 'SCTransformv2' function and then integrated together. Principal Component Analysis was then performed on highly variable genes. RunUMAP and FindNeighbours were run based on the first 50 principal components and unsupervised clustering was performed with FindClusters (resolution = 0.3). Cluster annotation was based on enriched genes in each cluster identified by 'FindConservedMarkers' function. Neural cell subclustering was performed as follows: A dataset containing only the ERG, neuron and oligodendrocyte cells was created using the subset function. Data were then normalized using 'SCTransformv2' function and then integrated together. Principal Component Analysis was then performed on highly variable genes. RunUMAP and FindNeighbours were run based on the first 50 principal components, and unsupervised clustering was performed with FindClusters (resolution = 0.7). Cluster annotation was based on enriched genes in each cluster identified by 'FindConservedMarkers' function.

## CellChat analysis

EC cluster was extracted from the trunk dataset and merge with the neural subclusters. To predict the cell-cell communication between EC and neural cells, we used the R toolkit Cellchat[41]. First, we analyzed both WT and *flt1*−/− merged datasets to predict the number of interactions between EC and neural cells and the main pathway targeting the endothelial cell. The dataset was analyzed using the previously published pipeline by Jin et al., 2021 with minor changes: for the computation of the communication probability, we selected the method type "truncatedMean" with "trim = 0.1" option. Secondly, we made a differential analysis of WT and *flt1*−/− datasets to predict the ligand-receptor interactions in WT versus flt1−/−. CellChat was run separately on each dataset that was then merged as described[73] and using the following parameters: for the computation of the over-expressed genes, we used the "identifyOverExpressedGenes" function with "thresh.*p* = 0.1" option; for the computation of the communication probability, we selected the method type "truncatedMean" with "trim = 0.1"; for filtering the number of expressing cells, we used "filterCommunication" with "min.cells=2" option. The ligand-receptor interactions were analyzed only from neural cells to EC and only for the pathway previously identified to target the EC.

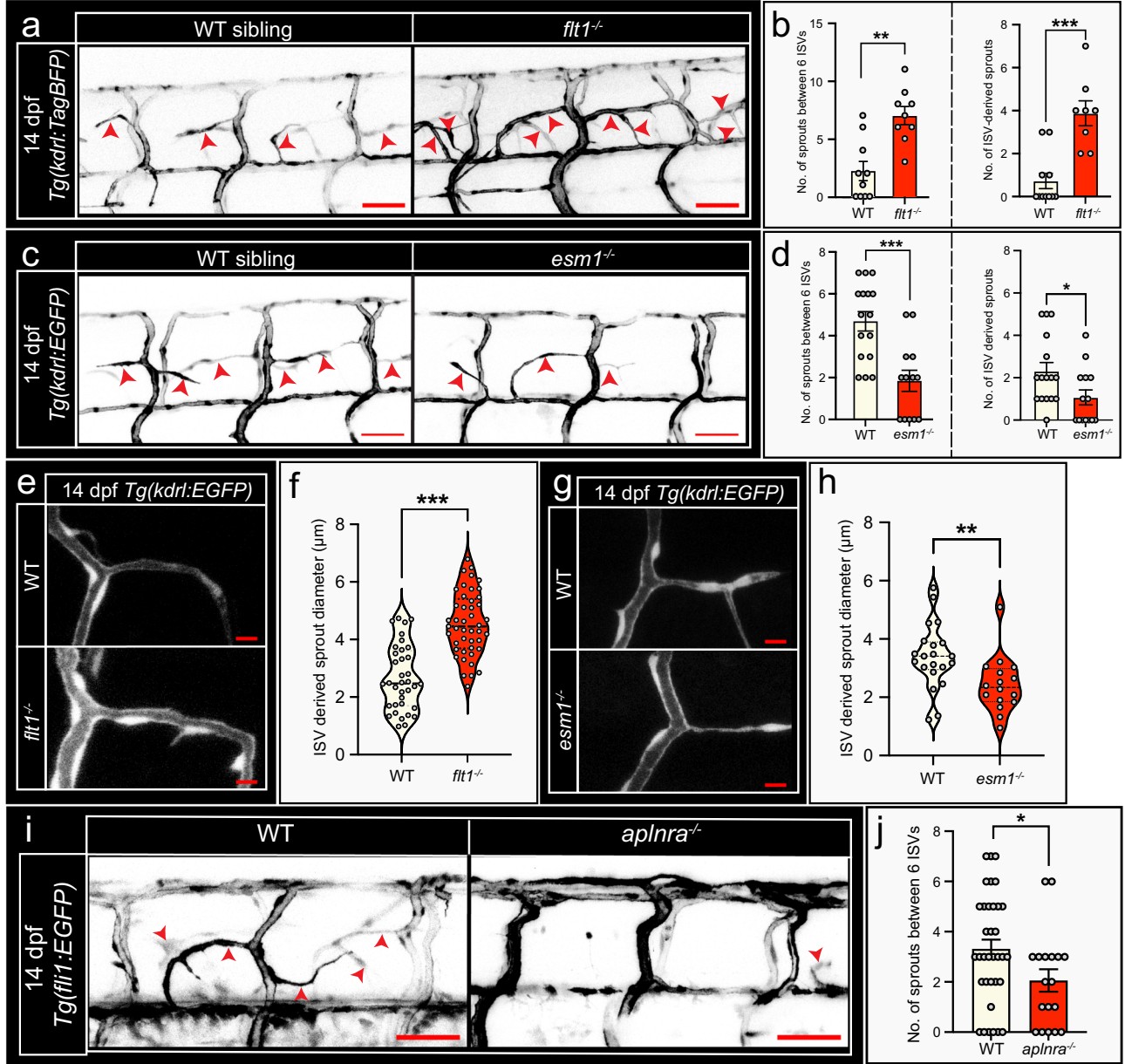

**Fig. 9 | Spinal cord vascularization in WT larvae is regulated by *aplnra*, *esm1* and *flt1*. a** Confocal images showing spinal cord vascular network in the trunk of day 14 WT siblings (left) and *flt1⁻/⁻* mutant (right). Arrowheads indicate sprouts. **b** Quantification of spinal cord vessel sprouting events (left) and ISV derived sprout number (right) in WT siblings and *flt1⁻/⁻* mutant based on images in **a**. Mean ± s.e.m.; two-sided Mann–Whitney *U* test; WT: *n* = 10; *flt1⁻/⁻*: *n* = 9; *p*(left) = 0.008, *p*(right) = 0.0004. **c** Confocal images showing spinal cord vascular network in WT siblings (left) and *esm1⁻/⁻* mutant (right) at 14 dpf. Arrowheads indicate sprouts. **d** Quantification of spinal cord sprouting events (left) and ISV derived sprout number (right) in WT siblings and *esm1⁻/⁻* mutant. Mean ± s.e.m.; two-sided Mann–Whitney *U* test; WT: *n* = 16; *esm1⁻/⁻*: *n* = 13; *p*(left) = 0.0006, *p*(right) = 0.0243. **e** Confocal images showing ISV derived sprout morphology in WT (top) and *flt1⁻/⁻* mutant (bottom) at 14 dpf. **f** Sprout diameter for indicated scenario. Violin plot showing the diameter distribution, median (dashed lines), interquartiles (dotted lines); two-sided Unpaired *t* test with Welch's correction; WT: n = 36 ISV-derived sprouts from 10 larvae, *flt1⁻/⁻*: *n* = 45 ISV-derived sprouts from 8 larvae; *p* < 0.0001. **g** Confocal images showing ISV derived sprout morphology in WT (top) and *esm1⁻/⁻* mutant (bottom) at 14 dpf. **h** Sprout diameter for indicated scenario. Violin plot showing the diameter distribution, median (dashed lines), interquartiles (dotted lines); two-sided Unpaired *t* test with Welch's correction; WT: *n* = 21 ISV-derived sprouts from 10 larvae, *esm1⁻/⁻*: *n* = 16 ISV-derived sprouts from 8 larvae; *p* = 0.0074. **i** Confocal images showing spinal cord vascular network in WT (left) and *aplnra⁻/⁻* mutant (right) at 14 dpf. Arrowheads indicate sprouts. **j** Quantification of spinal cord vessel sprouting events in *aplnra⁻/⁻* mutant and WT. Mean±s.e.m., two-sided Mann–Whitney *U* test, WT: *n* = 35 larvae; *aplnra⁻/⁻*: *n* = 18 larvae; *p* = 0.046. Scale bars indicate 50 μm in **a**, **c**, **e** and 10 μm in **g**, **i**. ISV, intersegmental vessel. Source data are provided as a Source Data file.

**Statistical analysis**

Statistical analysis was performed using GraphPad Prism v9. Each dataset was tested for normal distribution (Shapiro-Wilk test). Parametric method (unpaired Students *t*-test with or without Welch's correction) was only applied if the data were normally distributed. For non-normal distributed data sets, a non-parametric test (Mann–Whitney *U* test) was applied. For distribution analysis, the Fisher's exact test (2 categories) or the Chi-square test (>2 categories) was used. Data are represented as indicated in the legends. *, *p* < 0.05, **, *p* < 0.01 and ***, *p* < 0.001. For every treatment, treated and control embryos were derived from the same egg lay; embryos were selected on the following pre-established criteria: normal morphology, a

beating heart and circulating red blood cells. All images shown in the figures are representative examples of the respective phenotypes and expression patterns.

## Reporting summary

Further information on research design is available in the Nature Portfolio Reporting Summary linked to this article.

## Data availability

The raw data of single-cell RNA-seq generated in this study have been deposited in the Gene Expression Omnibus database under accession code GSE227806 and GSE227696. The zebrafish genome GRCz11 used in this study is available in the NCBI database under accession code GCF_000002035.6. The remaining data are available within the Supplementary Information or Source Data file. Source data are provided with this paper.

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

## Acknowledgements

We thank the colleagues of the KIT-European Zebrafish Resource Center (EZRC) for the handling and maintenance of the zebrafish lines. We thank Christian Helker for providing mutant zebrafish and tools. This work was supported by the Deutsche Forschungsgemeinschaft (DFG)—FOR2325 "Interactions at the NeuroVascular interface" to F.l.N.; and ERC-advanced grant Cor-Edit-P (ERC 101021043) to C.K. We thank the colleagues of the Flow cytometry and Genomics core facilities at EMBL in Heidelberg for the FACS sorting and single-cell experiments.

## Author contributions

L.P.: single-cell sequencing analysis, computational analysis and interpretation, zebrafish experiments and interpretation of data. A.L., A.M., N.O., M.M., M.W., H.P., D.G.: designed and performed experiments and interpreted experimental data. C.K.: critical conceptual input. F.l.N: conceptualization and design of study, data interpretation, funds raising. F.l.N. and L.P. wrote the manuscript with input from all authors.

## Funding

## Competing interests

The authors declare no competing interests.
