## [Peer Review File · Nature Communications]

Parenchymal cues define Vegfa driven venous angiogenesis by activating a sprouting competent venous endothelial subtypeREVIEWER COMMENTS

Reviewer #1 (Remarks to the Author):

The manuscript by Preau et al describes identification of pioneer cell, which governs the sprouting angiogenesis in venous vascular beds in zebrafish, and subsequent analyses on underlying molecular and cellular analyses. The authors find that synergistic interaction between Vegfa and Aplnr signaling regulates activation of Trio. In addition, they delve into the modulation of Vegfa ligand expression neuronal Esm1, linking the interplay between neuronal and vascular cells in coordinating sprouting angiogenesis. The premise of the manuscript is interesting and the quality of the images and other data are excellent. However, the manuscript appears to be extension of previous research published by the authors' group and others. Moreover, definition and characteristics of the 'pioneer cell', which appears to be poorly established and need to be further developed.

Major Comments:

1. In Figure 1d and e, the authors show that venous 'pioneer cell' emerges from the venous ISVs and connects to the neighboring pioneer cell and undergo anastomosis. Based on the observation that these cells are lumenized and decorated with mural cells, the authors suggest the new term, pioneer cell, to describe them. Considering the environments that these venous sprouts are exposed to compared to the arterial sprouts (i.e. presence of hemodynamic pressure and differentiated mural cells), it is reasonable to assume that these 'pioneer cells' are likely to be tip cells. The authors either need to clearly suggest the definition of 'pioneer cells' or better use conventionally used terminology.
2. In Figure 2, the authors rely on cellchat to identify signaling pathways mediating the interaction between neurons and endothelial cells. The main problem with this approach is that the number of ECs. Moreover, in the endothelial population the authors used for their analyses include diverse ECs residing in the trunk area. Therefore, the number of ECs adjacent to the neural tube will be further decreased. Therefore, it is unclear how reliable cellchat analyses would be in this experimental setting. In addition, in Figure 2a, it would be more convincing if the authors include the expression profiles of genes pertinent to a specific signaling pathways as a FeaturePlot.
3. The authors pinpoint Apln signaling from their cellchat analyses. Given the nature of the ligand as a secreted small polypeptide, Apln signaling activity is relying on the protein processing as well as exocytosis. Since scRNA-seq and other analyses in the manuscript detects mRNA, but not the mature ligand, it would be much more convincing if the authors directly show the deposition of Apln polypeptide. Of course, this may be technically challenging, but will definitely increase the confidence of the data shown in Figure 2.

Minor Comments:

1. In Figure 3, the authors claim that c-Able enables Vegfr2 signaling by phosphorylate its Tyr residues. The research paper cited here is under S1P/S1PR activation. Will this finding be applicable to the

regulation of Kdr1 activity in ECs stimulated with Apln/APJ?

2. Some of the captions in the figure needs to be italicized (for instance, Figure 3a and 3d).
3. Have the authors examine the effects of Apela manipulation, which could activate Aplnr?

Reviewer #2 (Remarks to the Author):

This manuscript titled “Neuronal cues define an organo-typical angiogenic venous endothelial subtype determining spinal cord vascularization” by Preau et al. describes ectopic angiogenic phenotypes to form vascular anastomosis between vISV and aISV in fish models *flt1*^{-/-} and *vhl*^{-/-}. The authors used scRNAseq to discover *aplna/b* pathways and found that loss of *aplnra* rescue the ectopic angiogenesis phenotype in *flt1* or *vhl* null fish. The authors found this rescue effect is correlated with EC migration and polarization in the vISV. *Apln*, the ligand of *aplnra*, is expressed in the neural and ECs. *Aplnra* KO or silencing *apln* can rescue the ectopic angiogenic phenotype in *flt1* null. In their attempt to search for mechanism, they found that silencing Trio (Fig.6) or *esm1* ko (Fig.7) also have the rescue effect.

Overall, this study provides high resolution data in multiple fish models to demonstrate that *aplnra*, trio and *esm1* pathways can rescue the ectopic angiogenic phenotypes in *flt1* null fish. However, there are several major issues:

1. the novelty of this study is questionable. *Flt1* null is known to increase angiogenesis. Also, since *apelin* and *esm1* pathways are known as pro-angiogenic signaling, it is not surprising that these signaling molecules are upregulated in *flt1* null (scRNAseq) and that loss of these signaling pathways can reduce the ectopic angiogenic phenotype in *flt1* null.
2. Although it is nice that *aplnra*, trio and *esm1* are found to be responsible for the phenotype, there is a lack of linkage of these three signalings (plus *flt1*) as one signaling axis. Instead, the authors attempted to explain their rescue phenotype separately by providing hypothetical explanations towards different possibilities.
3. Most of the data are descriptive and conclusions are suggestive. There seems to be an overuse of terminology and complex phrasing in the text, making it confusing and difficult to read.

Other issues:

Line48-50 is the major question this paper asks, which, however, does not seem to be a critical knowledge gap in vascular biology. Veins and arteries are fate-specified endothelial cells. Sprouting ECs derived from veins or arteries, as driven by environmental stimuli like VEGF, have lost their identity as venous or arterial. Therefore, the Notch activity in arterial or venous cells, which are driven by different environmental cues (e.g. shear stress), is not relevant to the VEGF-driven Notch signaling in sprouting ECs.

It is nice that Fig.1e shows blood flow in the lumenized tertiary sprouting EC (pioneer cells). But it is not clear why this is important to the questions asked with apelin signaling and *esm1*. This also raises more questions as hemodynamic force is involved.

Line 160: *dll4* MO does not inhibit tertiary “sprout remodeling” is a very confusing description, as *dll4* MO in supp Fig.4M seems to show an absence of the aISVs, thereby no tertiary anastomosis between vISV and aISV. It is not clear what “sprout remodeling” mean here. However, this is important since the authors seem to use this data to rule out the VEGF-*dll4*-Notch signaling.

Fig.4 g: This data showing “EC migration” is based on vascular labeling with A/V markers. Instead of saying there is “less migration against the flow from vISV to DLAV”, is it possible that the vISV is arterIALIZED in *flt1*^{-/-}; *aplnra* MO?

Lin 280: The whole paragraph addressing the hypothesis about *esm1* displacement of VEGF in binding to ECM is not supported by any data evidence. This part may move to discussion.

Fig.8 provides data on adult fish spinal cord vascularization on wt, *flt1* (increased) and *esm1* (reduced) fishes. These results are not surprising and not relevant to the major subject of this paper.

Reviewer #3 (Remarks to the Author):

This is a follow up study of the authors’ previous finding that in *Vegf* gain-of-function mutants such as *flt1*^{-/-} and *Vhl*^{-/-} mutants, ectopic sprouts emerge from the dorsal part of venous intersegmental vessel (vISV) (Wild et al. Nat Commn 2017). With a series of elegant genetic and pharmacologic approaches, the authors here characterize the cellular properties of such ectopic sprouts, called “tertiary sprouts”, in which a single lumenized endothelial cell (EC) emerges from vISV and forms an anastomotic connection with a lateral arterial intersegmental vessel (aISV).

The work in this study is nicely done and the quality of the experimental analysis is high. However, the authors need to clarify conceptual advances beyond the previous work describing that neuronal soluble *flt1* can limit ISV over-sprouting around the developing spinal cord. Does the single lumenized EC, termed “pioneer cell”, emerge only in the tertiary sprouts in the developing vISVs of *Vegf* gain-of-function mutants such as *flt1*^{-/-} and *Vhl*^{-/-} mutants? Though this is a novel finding, it is not clear whether we can define the pioneer cell-mediated tertiary sprouts as a process of spinal cord vascularization. Do the authors envision that the sprouts from vISVs in wild-types at 14 dpf (shown in Fig. 8A and C) develop as seen in *flt1*^{-/-} and *Vhl*^{-/-} mutants at 4dpf, given that neuronal *Vegf* expression is increased during the process of spinal cord vascularization at 12-14 dpf? If so, the authors need to examine the cellular properties of the sprouts from vISVs in wild-types at 12-14 dpf.

The authors provide mechanistic evidence underlying the pioneer cell-mediated tertiary sprouts in the

developing vISVs of Vegf gain-of-function mutants. Indeed, given the up-regulation of Apelin (apln)-receptor-a (aplnra) signaling, the guanine exchange factor Trio, and endothelial cell specific molecule-1 (esm1) in flt1^{-/-} mutants, the authors clearly demonstrate that the lack of these molecules attenuates the pioneer cell-mediated tertiary sprouts in the developing vISVs of Vegf gain-of-function mutants. It is not clear whether these signaling molecules are involved only in the tertiary sprouts in the developing vISVs of Vegf gain-of-function mutants? It is not clear which ones they think are the most significant. As mentioned above, are these signaling molecules also involved in the sprouts from vISVs during the process of spinal cord vascularization at 12-14 dpf? Given that Vegfa-Kdr signaling is essential for the primary sprouts from the dorsal aorta, the authors need to clarify whether these signaling molecules are involved in the primary sprout formation.

- In INTRODUCTION, the authors need to describe the previous work by their group (Wild et al. Nat Commn 2017) and the other group (Matsuoka et al. eLife 2016) and explain what questions remains unanswered.
- The authors need to clarify whether the flt1^{-/-} mutants (flt1-ka604 and flt1-ka602) lack both soluble and membrane flt1.
- The authors need to define the number of tertiary sprouts in the mutants they analyze. Is there one tertiary sprout from one vISV in flt1^{-/-} and Vhl^{-/-} mutants? There are multiple EC nuclei in tertiary sprouts in the Vhl^{-/-} mutants (Fig. 3d). The authors claim that both flt1^{-/-} and Vhl^{-/-} mutants are used as Vegf gain-of-function mutants, but it would be better to examine whether Vegfa-overexpression directs the formation of tertiary sprouts from the developing vISVs.
- Since the previous studies demonstrate the involvement of blood flow in the junctional fingers and lumen formation (Kotini et al. Cell Rep 2022), the authors need to examine whether tertiary sprouts are dependent on hemodynamic forces. What happens to tertiary sprouts in flt1^{-/-} and Vhl^{-/-} mutants with tricaine treatment (inhibition of blood flow and blood pressure) or norepinephrine treatment (increase in hemodynamic forces)?
- According to scRNAseq shown in Fig. 2, the authors need to indicate which cells express Vegfa. In Fig. 2f, the authors need to explain why most ligand-receptor interactions including vegfaa-kdr, cxcl12-cxcr4a and apln-aplnr are not enhanced in flt1^{-/-} mutants compared to wild-types.
- In Fig. 3, it is not clear whether aplnr is also up-regulated in flt1^{-/-} mutants. The authors also need to examine whether the lack of aplnr (aplnr-mu270) results in the reduction of tertiary sprouts in flt1^{-/-} and Vhl^{-/-} mutants. Although Fig.S5 indicates "Vascular phenotypes of aplnra^{-/-} and aplnr^{-/-} mutants," the authors do not show any data about aplnr^{-/-} mutants.
- In Fig. 4, it is not clear whether there is any difference in EC polarization and migration in the vISV between flt1^{-/-};aplnra^{-/-} double mutants and wild-types. The authors need to compare between wild-types, flt1^{-/-}, and flt1^{-/-};aplnra^{-/-} double mutants.
- In Fig. 5, the authors need to examine the effect of EC-derived Apelin (apln) in tertiary sprouts in flt1^{-/-} mutants. It is not clear whether neuron-specific loss of apln reduces tertiary sprouts in flt1^{-/-} mutants at 3 dpf. The authors should examine the vascular phenotype at 4 dpf.
- The authors claim that an increase in the tertiary sprout volume is due to Trio and its downstream effector Rac1 acting on EC actin remodeling (Fig. 6). It is not clear whether the Trio-Rac1 axis is involved only in tertiary sprouts in flt1^{-/-} mutants. Is there any difference in the size of ISVs between flt1^{-/-};Trio morpholino mutants or flt1^{-/-};Rac1 inhibitor mutants and wild-types. Does venous specific overexpression of TrioN alone induce tertiary sprouts in the developing vISVs?

- Although EC-specific overexpression of *esm1* does not induce additional venous sprouts in *flt1*^{-/-} mutants (Fig. 7), it would be better to show whether EC-specific overexpression of *esm1* can restore tertiary sprouts in *flt1*^{-/-};*esm1*^{-/-} double mutants.
- The capillary and neuronal phenotypes in the transverse section (Fig. 8e) are difficult to interpret because of the poor image qualities. The capillary images need to be improved due to lack of cellular morphology (especially in *esm1*^{-/-} mutants). The authors claim that change in capillaries affects neural differentiation in the spinal cord of *flt1*^{-/-} and *esm1*^{-/-} mutants, but no further investigation including proliferation and cell death of ventricular zone neural progenitors and mature neurons are provided in support.

revision number 1 – NCOMMS-23-15010A– Preau et al**General response to the reviewers' comments:**

We would like to thank the reviewers for their constructive feedback and suggestions to improve quality of the manuscript. Their remarks including “the work in this study is nicely done and the quality of the experimental analysis is high” – “the premise of the manuscript is interesting” and “the quality of the images and other data are excellent” are well taken.

To address the reviewers' concerns we performed a series of additional experiments: we show that tertiary sprouting involves hemodynamic forces, we analyzed the contribution of AV identity, and substantiated the role of *apl^{nr}b* and *apela*. We furthermore compared the cellular and molecular regulation of tertiary sprouting in mutants with a Vegfa gain of function scenario with vascularization of the WT spinal cord in more detail and outline the similarities between both processes.

Response to Reviewer 1:

Reviewer #1 (Remarks to the Author):

The manuscript by Preau et al describes identification of pioneer cell, which governs the sprouting angiogenesis in venous vascular beds in zebrafish, and subsequent analyses on underlying molecular and cellular analyses. The authors find that synergistic interaction between Vegfa and Aplnr signaling regulates activation of Trio. In addition, they delve into the modulation of Vegfa ligand expression neuronal Esm1, linking the interplay between neuronal and vascular cells in coordinating sprouting angiogenesis. **The premise of the manuscript is interesting and the quality of the images and other data are excellent.** However, the manuscript appears to be extension of previous research published by the authors' group and others. Moreover, definition and characteristics of the 'pioneer cell', which appears to be poorly established and need to be further developed.

Major Comments:

Question 1: Considering the environments that these venous sprouts are exposed to compared to the arterial sprouts (i.e. presence of hemodynamic pressure and differentiated mural cells), it is reasonable to assume that these 'pioneer cells' are likely to be tip cells. The authors either need to clearly suggest the definition of 'pioneer cells' or better use conventionally used terminology.

Answer 1: The leading cell in the tertiary sprout both guides the angiogenic sprout and forms a lumen. In line with the reviewers' suggestion and in accordance with the conventional terminology, the **Pioneer cell** may thus be termed **Lumenized Tip Cell (L-Tip Cell)**. We adapted the MS accordingly. Of note: in the arterial sprouting model, lumenization is a feature of the stalk cell, the tip cell is considered non-lumenized (Phng&Gerhardt, Dev Cell, 2009). As suggested by the reviewer we substantiated the differences between Vegf driven tertiary sprouting in the venous domain, and Vegf driven primary artery sprouting.

Additional differences described in the revised MS: **Tertiary sprouting requires hemodynamic forces – primary artery sprouting does not.** Lowering embryonic heart rate and reducing trunk perfusion impaired tertiary sprouting (**Supplemental Fig. 2a-c**). In contrast, it is established that primary sprouting does not require hemodynamics and can occur in the absence of blood flow (Isogai&Weinstein, Development, 2003). Furthermore: tertiary sprouts are more dependent on **glycolytic flux** than primary sprouts (**Supplemental Fig. 2j-m**). **Third:** primary artery sprouting is depending on **Ca²⁺ signals** in the differentiating (artery derived) tip cells (Yokota&Mochizuki, eLife, 2015), but such Ca²⁺ signals are lacking in the venous domain (**Supplemental Fig. 2n-q**).

For clarity, we added graphical illustrations showing the differences between Vegf driven sprouting in the venous and arterial domain (**Supplemental Figure 2r,s & Supplemental Figure 13**)

Question 2: It is unclear how reliable cellchat analyses would be in this experimental setting. In addition, in Figure 2a, it would be more convincing if the authors include the expression profiles of genes pertinent to a specific signaling pathways as a FeaturePlot.

Answer 2: In line with the suggestions made by the reviewer we performed signaling pathway analyses and present FeaturePlots (Supplemental Figure 5). See also reviewer 3, answer 13.

We agree with the reviewer that CellChat is a prediction tool (it is not perfect), and the reliability of the predictions made by CellChat has to be evaluated *in vivo*. For that reason, we validated several candidates identified by this method *in vivo* using a genetic approach. CellChat predicted a role for neural Apelin to EC signaling. To substantiate this prediction, we generated neural specific *apelin* loss of function embryos and found that neural specific ablation of *apelin* significantly reduced tertiary sprouting (Figure 5l,m & Supplemental Fig. 10b,c). Conversely, neural specific *apelin* gain of function promoted tertiary sprouting (Figure 5j,k). Genetically ablating *apelin-receptor-a* (*aplnra*) specifically reduced tertiary sprouting (Figure 3a-e & Supplemental Fig. 7k,l) without affecting primary and secondary sprouting and aISV/vISV remodeling in the trunk (Supplemental Figure 7d,e,j,m). We conclude that in line with the prediction made by CellChat, the Apelin - Apelin receptor-a signaling pathway plays a functional role in tertiary sprouting.

Although CellChat predicted a role for neural Cxcl12 to vascular Cxcr4a signaling, we could not confirm a specific role for this pathway in tertiary sprouting in the trunk-spinal cord region. Tertiary sprouting around the spinal cord was not affected in *flt1*^{-/-} mutants in which *cxcr4a* (Supplemental Figure 6d-h) or *cxcl12* (Supplemental Figure 6o-f) was silenced, suggesting that these molecules are not required for tertiary sprouting. In contrast, morpholino-mediated loss of *cxcr4a* (Supplemental Fig. 6a-c), or loss of *cxcl12* (Supplemental Figure 6m,n) reduced sprouting angiogenesis in the brain, consistent with the mutant phenotypes (Bussmann et al, Development, 2011), and in line with tissue dependent heterogeneity in sprout regulation. CellChat also predicted a role for the Vegfaa-Kdr1 axis. We previously validated the contribution of neural Vegfaa and vascular Kdr1, and showed that reducing neural *vegfaa* or blocking endothelial Kdr1 tyrosine kinase signaling inhibits tertiary sprouting (Wild et al, Nat Commun, 2017). Conversely, neural *vegfaa* gain of function promoted tertiary sprouting. Based on the results of these validation experiments, we decided to focus on the Apelin receptor signaling pathway, and the genetic interaction with Vegfaa-Kdr1 signaling in mediating venous sprouting.

Question 3: The authors pinpoint Apln signaling from their cellchat analyses. Given the nature of the ligand as a secreted small polypeptide, Apln signaling activity is relying on the protein processing as well as exocytosis. Since scRNA-seq and other analyses in the manuscript detects mRNA, but not the mature ligand, it would be much more convincing if the authors directly show the deposition of Apln polypeptide. Of course, this may be technically challenging, but will definitely increase the confidence of the data shown in Figure 2.

Answer 3: We agree with the reviewer that demonstrating Apelin protein expression by neural cells would strengthen our conclusions. Unfortunately, the commercial antibodies did not work in zebrafish, and generating a HA-tag knock-in into the endogenous *apln* gene locus to allow visualization of Apelin protein distribution using anti-HA antibodies, too laborious for this revision (due to the low integration rates in zebrafish this will take at least >1 year to generate the transgenic). To address the reviewers' comment more directly we therefore decided for a genetic approach. To provide direct genetic evidence demonstrating a functional role for neural-derived Apelin in tertiary sprouting, we generated a neural-specific *apln* loss of function mutant (Figure 5l,m & Supplemental Figure 10b,c) using a CrisprCas9 approach. We found that neural specific ablation of *apln* reduced tertiary sprouting in line with the prediction of our parenchyma-vessel cross-talk model. Vice versa, neural-specific *apln* gain of function promoted tertiary sprouting (Figure 5j,k).

Minor Comments:

Question 4: In Figure 3, the authors claim that c-Abl enables Vegfr2 signaling by phosphorylate its Tyr residues. The research paper cited here is under S1P/S1PR activation. Will this finding be applicable to the regulation of Kdr1 activity in ECs stimulated with Apln/APJ?

Answer 4: In support of coupling between *Aplnra* and Kdr1 via c-Abl we find that inhibiting G α -i with PTX or c-Abl with Dasatinib or Olverembatinib reduced tertiary sprouting in mutants with a VEGF

gain of function scenario (Figure 3l-p & Supplemental Fig. 8b-e). The reviewer is correct, in human cells, in the case of S1P/S1PR the coupling is achieved via differential phosphorylation of the Tyr residues of the KDR receptor. Such receptor phosphorylation events are difficult to detect in zebrafish embryos and we therefore turned to an *in vitro* approach using human umbilical vein endothelial cells (HUVEC) (see figure below – at the discretion of the reviewer – not for publication). HUVEC cells express *KDR* and interestingly, *in vitro*, maintaining *APLNR* expression was dependent on blood flow, and *APLNR* levels were low in the absence of a shear stress stimulus. These observations support our new *in vivo* findings showing that tertiary sprouting requires blood flow (Supplemental Figure 2a-c). Furthermore, to measure KDR phosphorylation involving interaction with APLNR cells, HUVEC thus have to be exposed to shear. To maintain endothelial *APLNR* expression we seeded HUVEC in Ibidi flow chambers and exposed them to flow. Using RNAseq we noticed that siRNA mediated silencing of *APLNR* in flow exposed HUVEC reduced the expression of *KDR* (see figure below – at the discretion of the reviewer – not for publication). This suggests that *in vitro* *APLNR* may act as a positive regulator of *KDR* expression under flow conditions. Thus, besides the proposed interaction between APLNR acting on KDR phosphorylation, an alternative scenario may involve a direct effect of *APLNR* on *KDR* expression.

Figure 1 – for reviewing purpose only. Left panel: overview flow experiment with HUVEC. Middle panel: *APLNR* expression in static and flow conditions. Note: *APLNR* expression requires flow, n=3, expression in FPKM. Right panel: *KDR* expression in HUVEC upon siRNA mediated knockdown of *APLNR*, in flow and in static conditions, n=3, expression in FPKM. Note that knockdown of *APLNR* reduced *KDR* expression under flow conditions.

Unfortunately, in the flow chambers one can only seed a limited number of endothelial cells, and the quantity was not sufficient to perform a Western blot, and measure KDR phosphorylation levels at the different Tyr residues. To tackle this problem, we are exploring stable transfection of human endothelial cells with *APLNR* and upscaling of the flow experiments to a scenario compatible with performing a Western blot (but this will take some time).

Question 5: Some of the captions in the figure needs to be italicized (for instance, Figure 3a and 3d).

Answer 5: we apologize for these omissions we adapted the captions in the figures accordingly.

Question 6: Have the authors examine the effects of *Apela* manipulation, which could activate *Apnr*?

Answer 6: to address this question we cloned zebrafish *apela* and expressed it under control of the neural specific promoter *Xla.tubb*. Neuronal specific *apela* gain of function in *flt1*^{-/-} had no measurable impact on tertiary sprouting propensity in the venous domain. However, a limited number of investigated embryos displayed small ectopic sprout like projections emanating from arterial ISVs; these sprouts lacked a lumen (Supplemental Fig. 10g-j). We observed no obvious changes in vessel branching upon neural specific overexpression of *apela* in WT (Supplemental Fig. 10k-m).

--- continued on the next page ---

Response to Reviewer 2:

Reviewer #2 (Remarks to the Author):

This manuscript titled “Neuronal cues define an organotypical angiogenic venous endothelial subtype determining spinal cord vascularization” by Preau et al. describes ectopic angiogenic phenotypes to form vascular anastomosis between vISV and aISV in fish models *flt1*^{-/-} and *vhl*^{-/-}. The authors used scRNAseq to discover *aplna/b* pathways and found that loss of *aplnra* rescue the ectopic angiogenesis phenotype in *flt1* or *vhl* null fish. The authors found this rescue effect is correlated with EC migration and polarization in the vISV. *Apln*, the ligand of *aplnra*, is expressed in the neural and ECs. *Aplnra* KO or silencing *apln* can rescue the ectopic angiogenic phenotype in *flt1* null. In their attempt to search for mechanism, they found that silencing *Trio* (Fig.6) or *esm1* ko (Fig.7) also have the rescue effect.

Overall, this study provides high resolution data in multiple fish models to demonstrate that *aplnra*, *trio* and *esm1* pathways can rescue the ectopic angiogenic phenotypes in *flt1* null fish.

Response: As suggested by the reviewer we clarified the linkage of *sflt*, *aplnra*, *trio* and *esm1* with respect to the regulation of venous sprouting propensity, pioneer/L-Tip cell lumenization and anastomosis formation. In addition, we now show that hemodynamic forces contribute to tertiary sprouting, and we addressed the consequences of loss of *dll4* in more detail.

Major Comments:

Question 1: the novelty of this study is questionable. *Flt1* null is known to increase angiogenesis. Also, since *apelin* and *esm1* pathways are known as pro-angiogenic signaling, it is not surprising that these signaling molecules are upregulated in *flt1* null (scRNAseq) and that loss of these signaling pathways can reduce the ectopic angiogenic phenotype in *flt1* null.

Answer 1: We thank the reviewer for this challenging remark, prompting us to clarify the news-value of our paper, as we see it.

As stated above the main thrust of the MS is the description of a novel form of sprouting, which features single lumenized endothelial cells. The differences to classical sprouting forms with regard to morphologic, molecular and morphologic criteria are clearly laid out (response to Rev#1, answer 1). They comprise the essential signaling of *neural apelin* and *neural esm1* as part of a parenchyma specific regulation of vascular development, which may represent just an example of how parenchymal cells pattern their own vasculature according to their organ-specific perfusion needs. However, since we focused on neuronal tissue only, we refrained at this point from further generalization.

In more detail *Apelin-receptor-a (aplnra)* is a G-protein coupled receptor expressed by endothelial cells and has to the best of our knowledge not been assigned a function in the vascular system as of yet. We show that *aplnra* is expressed by a subpopulation of venous endothelial cells, and this subpopulation is physiologically required for venous sprouting in mutants with a *Vegfa* gain of function scenario. We are also not aware of any other zebrafish or mouse study demonstrating the physiological relevance of genetic interaction between *Aplnra* and *Kdr1* to augment *Vegfa* receptor signaling output and promote venous sprouting. Conceptually, the notion that venous sprouting angiogenesis at the neuro-vascular interface requires two neural derived ligands, *Vegf* and *Apelin*, interacting with two genetically coupled receptors, *Kdr1* and *Aplnra* on competent venous endothelial cells, allows for very precise spatial control of sprouting at high resolution in a tissue specific manner – we believe that this is an aspect thus far not considered in explaining the spatial positioning of sprouting in a tissue dependent manner.

Thirdly, *Esm1* is a secreted molecule that is produced by, among others, endothelial cells, and neural cells during development. We are not aware of studies addressing the impact of neural/non-vascular *Esm1* on angiogenesis, or the consequences of loss/gain of neural *Esm1* on vascular development in the developing mouse spinal cord or zebrafish spinal cord. In the mouse community, *Esm1* is used as an endothelial tip cell marker. Consistent with this we find that *esm1* marks the classical tip cell population as observed in developing *Vegf-Dll4-Notch* driven primary artery sprouts (Figure 7b, left panel). However, sprouts in the venous domain lack consistent *esm1* expression (Figure 7b, right panel); our genetic approaches and rescue experiments indicate that

endothelial *esm1* cannot provide an explanation for venous sprouting. Instead, we identified a functional role for **neural *esm1*** (Figure 7l-f). In line with this we show that **neural specific expression of *esm1*** in tertiary sprout lacking *flt1*^{-/-};*esm1*^{-/-} double mutants **rescues** the tertiary sprouting phenotype (Figure 7p,q). In contrast, endothelial specific expression of *esm1* in *flt1*^{-/-};*esm1*^{-/-} double mutants does not provide a rescue (Figure 7n,q). These genetic data indicate that neural *esm1*, not vascular *esm1* is the physiologically relevant contributor for tertiary sprouting.

We show that both *esm1* and *aplnra* are specifically required for tertiary sprouting in the venous domain but dispensable for Vegf-Dll4-Notch driven sprouting in the arterial domain or aISV/vISV remodeling, a process dependent on adequate secondary sprouting. This indicates that *esm1* and *aplnra* are not just “general markers of any sprout” and consequently “upregulated” during “any sprouting event” but employed in a highly specific context dependent manner consistent with tissue dependent heterogeneity in sprouting forms.

Fourth, the reviewer is correct that Flt1 plays a role in spinal cord vascularization in zebrafish, chicken and mouse models via regulation of Vegf bio-availability. However, it is still debated which cell-type is responsible for this effect, and how Vegf bio-availability levels are titrated in a spatial-temporal manner at the neuro-vascular interface (Hogan&Schulte Merker, Dev Cell, 2017). We defined a new, hierarchically overarching determinant of Vegfa availability involving neural *esm1*. *In vitro* data suggest that Esm1 can augment Vegfa availability by displacing Vegfa from ECM binding moieties (Rocha&Adams, Circ. Res. 2014). This predicts that expressing *esm1* increases Vegf availability, whereas loss of *esm1* allows for more Vegf to bind to the ECM, hence, decrease Vegf availability. In line with the mechanism proposed *in vitro*, we demonstrate that neural *esm1* determines the tertiary sprouting propensity upon loss of *flt1* or *vhl*. Loss of *esm1* annihilates the pro-angiogenic responses observed in *flt1*^{-/-} and *vhl*^{-/-} mutants. Thus, for Vegfa driven tertiary sprouting to unfold, *esm1* must be expressed in neurons (Figure 7d-k). This shows that ultimately, not the increase in Vegfa availability resulting from loss of *flt1* or *vhl*, but *neural esm1* expression levels are the critical overarching determinant of Vegfa driven venous sprouting propensity (see also supplemental Figure 14). To our knowledge such non-cell autonomous role for neural *esm1* in mediating the vascular phenotype in *flt1* mutants has not been demonstrated in fish or mouse *in vivo*.

Taken together: we identified several key molecular and cellular differences between Vegf driven venous spouting and Vegf-Dll4/Notch driven sprouting in the arterial domain. We have added a graphical illustration showing the characteristics of tertiary sprouting back-to-back with the Dll4/Notch dependent sprouting model (Supplemental Fig. 13 & 14).

Question 2. Although it is nice that *aplnra*, *trio* and *esm1* are found to be responsible for the phenotype, there is a lack of linkage of these three signalings (plus *flt1*) as one signaling axis.

Answer 2: We are not exactly sure what kind of linkage of the named signaling molecules the reviewer envisions or whether this is a necessary element of the description of a novel phenomenon. What we can provide, though, is a list of indispensable elements of this phenomenon, which we now - advised by reviewer #1 termed L-tip cell sprouting. We show that in venous endothelial cells, Trio mediated endothelial cell enlargement downstream of Vegf-Kdr1 is pivotal for lumenization of L-Tip cells and tertiary sprout expansion along the spinal cord in mutants with a Vegf gain of function scenario (Figure 6a-l). As explained above, Vegf signaling strength depends on vascular *Aplnra*-Kdr1, neural Vegf and *Apelin*, with Vegf availability further fine-tuned by neural sFlt1 and neural Esm1. Vegf expression and sFlt1 splicing are regulated by hypoxia and couple to the differentiation status of the spinal cord neurons (Wild, Nat Commun, 2017). We recently showed how Vegf signaling activates the GEF Trio, and how Trio subsequently activates the RhoGTP-ase Rac1 specifically/only in endothelial cell junctions, to promote endothelial cell size increase and lumen diameter (Klems et al, Nat Commun, 202).

Tertiary sprouting requires Vegfa and our data suggest that Vegf availability at the neuro-vascular interface is determined by three factors: 1) the neuronal production of Vegfa, 2) the neuronal production of soluble Flt1 acting as a negative regulator of Vegfa bio-availability, and 3) the neuronal production of Esm1 acting as a positive regulator of Vegfa bio-availability. Collectively the balance between the expression levels of these factors will determine how much Vegfa will ultimately be presented to the Kdr1 expressing venous EC. In this MS we show that loss of neural *esm1* annihilates

the venous sprouting response in mutants with a Vegf gain of function scenario including *flt1*^{-/-} and *vhl*^{-/-} (Figure 7 & Supplemental Figure 12). Conversely, for venous sprouting angiogenesis to unfold in *flt1*^{-/-} and *vhl*^{-/-} mutants we find that *esm1* has to be expressed in neural cells (Figure 7).

Next, we provide evidence showing that Vegf driven venous sprouting requires genetic interaction between *Kdr1* and Apelin-receptor-a. Genetic interaction between Apelin-receptor-a and *Kdr1* may act to augment *Kdr1* signaling output – meaning that at an equal dosage of Vegf, Vegf signaling output and sprouting capacity will be higher if Apelin-receptor-a signaling couples to *Kdr1*. In support of this we show that inhibiting the coupling (by *Gαi* and c-Abl inhibition) or reducing Apelin-receptor-a signaling either by ablating *aplnra* or neural apelin reduces venous sprouting capacity in mutants with a Vegf gain of function scenario (Figure 3 & Supplemental Fig. 7k-m & Supplemental Fig. 8b-e). To our knowledge genetic interaction between Apelin-receptor-a and *Kdr1* to regulate Vegf signaling strength has not been demonstrated before (neither in zebrafish nor in mouse, or human endothelial cells). Selection of the sprouting endothelial cell in the venous domain thus requires: a competent *aplnra* and *kdr1* expressing venous EC, interacting with neural Apelin and Vegf, inducing venous sprouting exactly at the neuro-vascular interface. We postulate that this process allows sprout positioning at a much higher spatial resolution when compared to a model with only a single parenchymal derived ligand.

In conclusion: tertiary sprouting is determined by two-tiered regulation of Vegf signaling strength: 1) at the parenchymal level: regulation of Vegf availability through the opposing actions of sFlt1 and Esm1, 2) at the intracellular level: via control of signaling strength through genetic interaction of apelin-receptor-a with *Kdr1*. We have included a graphical abstract showing the connection between the mentioned molecules (Supplement Figure 11i; 14). We adapted the text of the discussion to make these points more clear.

Other issues:

Question 3: Line48-50 is the major question this paper asks, which, however, does not seem to be a critical knowledge gap in vascular biology. Veins and arteries are fate-specified endothelial cells. Sprouting ECs derived from veins or arteries, as driven by environmental stimuli like VEGF, have lost their identity as venous or arterial. Therefore, the Notch activity in arterial or venous cells, which are driven by different environmental cues (e.g. shear stress), is not relevant to the VEGF-driven Notch signaling in sprouting ECs.

Answer 3: The reviewer is correct, the recent single cell data sets from mouse retina (Eichmann lab, Dev. Cell, 2022) and liver (Benedito lab, Nature Cardiovascular Research, 2023) indeed show significant differences in the expression of AV identity genes between arterial, venous and endothelial tip cell populations. Except for *Unc5b* (besides tip cell marker also considered to mark arterial EC), there was no clear arterial-venous signature in sprouting vessels for the tissues investigated. In line with the reviewers' suggestion, we removed this text from the introduction.

Spurred by the question raised by reviewer we investigated the role of venous EC identity in venous sprouting in more detail in the zebrafish trunk vasculature. Consistent with tertiary sprouts emanating from veins, we found that tertiary sprouts expressed the venous marker *flt4*, and continued to do so until they made an anastomotic connection with the lateral aISV (Supplemental Figure 3a,b). To substantiate that tertiary sprouting requires endothelial cells with a venous identity, we created a trunk vasculature in which almost all trunk ISVs remained arterial by suppressing secondary sprouting through inhibiting *flt4* (for details on the experimental procedure see Hogan et al, Development, 2009). When the trunk vasculature consists of arterial ISVs only, we find that tertiary venous sprouting is absent (Supplemental Figure 3c-e).

If tertiary sprouting indeed requires endothelium with a more venous signature, we next reasoned that promoting venous ISV formation in *flt1*^{-/-} mutants should augment branching. In zebrafish, it is established that loss of the Notch ligand *dll4* promotes venous cell fate and *dll4* loss-of-function embryos display a trunk vasculature consisting almost exclusively of venous ISVs, at the expense of aISVs (Supplemental Figure 3i-k). For details on the experimental procedure see Leslie et al, Development, 2007). Accordingly, loss of *dll4* in *flt1*^{-/-} mutants, and generating a trunk

consisting of veins only, significantly augmented ectopic branching when compared with WT control, or *flt1*^{-/-} mutant (Supplemental Figure 3i-k).

We conclude that in the zebrafish embryo trunk vasculature, tertiary sprouting is specific for the venous domain and requires EC with a venous identity. To what extent these observations are species or organ dependent remains to be determined.

Question 4: It is nice that Fig.1e shows blood flow in the lumenized tertiary sprouting EC (pioneer cells). But it is not clear why this is important to the questions asked with apelin signaling and *esm1*. This also raises more questions as hemodynamic force is involved.

Answer 4: One obvious morphological difference we observed during confocal imaging is the lumenization of the tip cell, a feature that in the reference model is assigned to the stalk cell, not the tip cell (Phng&Gerhardt, Dev Cell, 2009). As a consequence of the lumenization, the sprout is immediately filled with blood plasma and exposed to hemodynamic forces. We believe that the latter has functional consequences: we observed pericyte recruitment around the tertiary sprout and these pericytes may counteract the forces exerted by pressure and stabilize the growing sprout. In contrast, pericytes are not associated with primary artery sprouting in the zebrafish embryo trunk (Wild et al, Nat Commun, 2017).

Since the tertiary sprout is already lumenized, and functionally connected to the vISV, it is exposed to hemodynamic forces - as evidenced by the finger-like junctions (Figure 1f,g). The reviewer asks whether hemodynamics influence the process of tertiary sprouting and we find that this is indeed the case. Lowering heart rate and trunk perfusion, significantly reduced tertiary sprouting in *flt1*^{-/-} mutants *in vivo* (Supplemental Figure 2a-c). To address the contribution of hemodynamics in more detail, we analyzed human venous endothelial cells exposed to blood flow using Ibidi chambers *in vitro* (see also #Rev1, answer 4). Our preliminary *in vitro* data show that hemodynamic factors are required for maintaining *APLNR* expression in venous EC, and loss of *APLNR* reduces *KDR* expression under flow conditions (figure of pilot data in our response to #Rev1, answer to question 4).

We next addressed the molecular mechanism accounting for the lumenization of the sprout and given the observed EC enlargement we focused on the GEF Trio, as Trio gain of function has been shown to promote EC enlargement and lumen formation in zebrafish and human model systems (Klems et al, Nat Commun, 2020). Trio acts downstream of Vegf-Kdrl to promote endothelial cell enlargement by regulating acto-myosin tension specifically at the cell junction (Klems et al, Nat Commun, 2020). We show that reducing *Trio* expression, or pharmacological inhibition of Trio activity in *flt1*^{-/-}, reduces EC size & sprout volume indicating that activation of Trio plays a functional role in lumenization of the venous tip cell (Figure 6d,e,h,i).

Taken together: in tertiary sprouts, Vegfa dependent Trio activation in venous EC, culminate into EC enlargement and lumenization. As a consequence of the lumenization, the sprout is filled with blood plasma, exposed to hemodynamic forces, which triggers the recruitment of pericytes to stabilize the sprout. Reducing hemodynamic forces abrogates tertiary sprouting.

Question 5: Line 160: *dll4* MO does not inhibit tertiary “sprout remodeling” is a very confusing description, as *dll4* MO in supp Fig.4M seems to show an absence of the aISVs, thereby no tertiary anastomosis between vISV and aISV. It is not clear what “sprout remodeling” mean here. However, this is important since the authors seem to use this data to rule out the VEGF-*dll4*-Notch signaling.

Answer 5: we agree that this maybe confusing. We wanted to highlight the heterogeneity in sprouting remodeling phenotypes observed upon loss of *dll4* between different organs. We therefore compared the *dll4* loss of function phenotype in the trunk-spinal cord region with the phenotype observed in the central nervous system. It is established that in the brain, loss of *dll4* completely inhibits sprout remodeling (Bussmann&Siekman, Development, 2011). In contrast, in our spinal cord scenario, sprouting and anastomosis remodeling are not reduced upon loss of *dll4* (Supplemental Figure 3i,m). This is in line with organ dependent heterogeneity in *Dll4*-Notch mediated sprouting responses as previously demonstrated also for bone tissue (Ramasamy&Adams, Nature 2014).

Loss of *dll4* promotes venous cell fate, hence, upon loss of *dll4* in the trunk there are more venous ISVs that can generate tertiary sprouts. However, the spatial positioning of the venous sprout, sprout lumenization, sprout progression, and anastomosis formation are *not* impaired (Supplemental Fig. 3i-m) as the parenchymal cues are still in place, and the venous competent cells are readily available. The reviewer is correct, since trunk aISVs are almost absent upon loss of *dll4* (as they are all transformed into vISVs), anastomotic connections emerge between two lateral venous ISVs instead of between a venous and arterial ISV. The latter is the consequence of the artery to venous cell fate switch upon loss of *dll4*. Thus, the sprouting process per se is unaltered, but the anastomotic vessel segment now connects vISVs-vISV instead of vISV-aISV. To clarify this point, we have rephrased the text of the results and discussion.

Question 6: Fig.4 g: This data showing “EC migration” is based on vascular labeling with A/V markers. Instead of saying there is “less migration against the flow from vISV to DLAV”, is it possible that the vISV is arterIALIZED in *flt1*^{-/-}; *aplra* MO?

Answer 6: In line with the reviewers’ suggestion, we checked if venous EC changed their identity and became arterIALIZED but found no evidence supporting such scenario (Supplemental movie 4 and 5 & Supplement Figure 9n,o). We found no evidence that loss of *aplra* affected arterial-venous identity in trunk vessels (Supplemental Fig. 7b-e).

Question 7: Lin 280: The whole paragraph addressing the hypothesis about *esm1* displacement of VEGF in binding to ECM is not supported by any data evidence. This part may move to discussion.

Answer 7: in line with the suggestion made by the reviewer, we moved this paragraph to the discussion. The *in vitro* data supporting the proposed competition model between ESM1 and VEGF for ECM binding can be found in the paper by the group of Ralf Adams (Rocha&Adams; Circ. Res. 2014).

Question 8: Fig.8 provides data on adult fish spinal cord vascularization on wt, *flt1* (increased) and *esm1* (reduced) fishes. These results are not surprising and not relevant to the major subject of this paper.

Answer 8: we removed the data addressing the adult zebrafish spinal cord from the MS. We replaced this figure with a more detailed cellular analysis of the sprouting events in the day 12 -14 WT spinal cord (new Figure 8 & 9).

--- continued on the next page ---

Reviewer #3 (Remarks to the Author):

This is a follow up study of the authors' previous finding that in Vegf gain-of-function mutants such as *flt1*^{-/-} and *Vhl*^{-/-} mutants, ectopic sprouts emerge from the dorsal part of venous inter-segmental vessel (vISV) (Wild et al. Nat Commn 2017). With a series of elegant genetic and pharmacologic approaches, the authors here characterize the cellular properties of such ectopic sprouts, called "tertiary sprouts", in which a single lumenized endothelial cell (EC) emerges from vISV and forms an anastomotic connection with a lateral arterial intersegmental vessel (aISV).

The work in this study is nicely done and the quality of the experimental analysis is high.

General Response: We thank reviewer 3 for the kind remarks and the helpful suggestions to strengthen our manuscript. We have performed the requested experiments, and added new data comparing similarities of tertiary sprouting around the spinal cord in mutants with a Vegf gain of function scenario with vascularization of the WT spinal cord around day 12-14. We furthermore addressed *apelin-receptor-b*, and the impact of hemodynamics. Before addressing the questions point-to-point, we like to take the liberty to explain the rationale behind our experimental approach as we believe that this may clarify several of the reviewers' questions regarding tertiary sprouting versus spinal cord vascularization in WT, as well as the conceptual novelties, before going in depth into molecular and cellular controls mechanisms. In the revised version we adapted the text to make similarities and differences between tertiary sprouting and spinal cord vascularization in WT more clear.

We consider *Vegfa* as an important target for revascularization strategies. In this context we are interested how different vascular beds respond to *Vegfa* gain of function, as this would mimic the therapeutic pro-angiogenic approach in patients with ischemic cardiovascular disease. *Vegfa* is also the main driver of vascularization in most organs involving a local tissue specific *Vegf* gain of function scenario. We postulate that the organ specific variability in *Vegfa* driven sprouting processes involves specific parenchymal cues. Understanding this cross-talk, may help us in developing more specific pro-angiogenic strategies. For us, the trunk spinal cord – vascular interface is a model to learn how parenchymal cues can influence *Vegf* dependent sprouting processes.

To address this, we evaluated variability in sprout remodeling in a series of mutants and transgenics with a *Vegf* gain of function scenario including *flt1* and *vhl* mutants. In the trunk of mutants with a *Vegfa* gain of function scenario, we observed ectopic venous sprouting, at 3dpf, in a part of the venous domain that is in close contact with the spinal cord. We found such venous sprouting events intriguing as sprouting in the venous domain has thus far been associated with *Vegfc*-*Flt4* (or sometimes *Bmp* signaling) whereas *Vegfa* is considered to be the main driver of sprouting of arterial intersegmental vessels involving cross-talk with *Dll4*-*Notch* (generally regarded as the reference sprouting model). Taken together these observations raised two questions: what determines venous endothelial responsiveness to *Vegfa*, and why do the ectopic venous sprouts preferentially develop in close proximity to the spinal cord – in other words, is this process the result of specific parenchymal cues (as you would also expect in organo-typical control).

To address these questions, we first compared the cellular and molecular events controlling *Vegfa* induced venous sprouting with *Vegfa* induced arterial sprouting (the reference model), and found that these two processes are fundamentally different. In the revised MS we now describe 8 cellular and molecular differences in great detail. Most notably, venous sprouting does not require *Dll4*-*Notch*. Instead, we find that venous sprouting requires genetic interaction between *Aplnra* and *Kdr1* in a subset venous endothelial cells, activation of the GEF Trio downstream of venous *Kdr1* signaling, neural *apelin*, *vegfa* and neural *esm1*, coordinated via intricate neuro-vascular cross-talk. In the period day 1-3, loss of *aplra* or *esm1* do not affect primary artery or secondary venous sprouting or other aspects of aISV / vISV remodeling in the trunk suggesting that, during this developmental window, *aplra* and *esm1* are specific for tertiary sprouting.

The next question is, is *Vegfa* induced ectopic venous sprouting around the spinal cord at day 3 (which we termed tertiary sprouting), comparable to the sprouting process that governs spinal cord vascularization in the period day 12-14 in WT.

Indeed, there are morphological and molecular similarities. In both scenarios, *aplra* and *esm1* act as positive regulators of spinal cord vascularization by regulating sprouting propensity (Figure 9c,d,i,j). Conversely, *flt1* acts as a negative regulator of sprouting in both scenarios (Figure

9a,b). In the WT scenario, the majority of sprouts innervating the spinal cord emanate from ISVs expressing the venous markers *flt4* or *CouprTFII*, in line with venous sprouting events contributing to spinal cord vascularization (similar to the day3-Vegf GOF scenario) (Figure 8a-e). We furthermore observed lumenized sprouts emanating from IVSs in WT. In the WT scenario the percentage of ISV derived sprouts with a blood plasma filled lumen was 67% as verified by labelling blood plasma with FITC-dextran (Figure 8h,i). In the period day 12-14, *flt1*^{-/-} mutants displayed a larger sprout lumen diameter when compared to WT, whereas *esm1*^{-/-} mutants showed a significantly reduced diameter in line with sprout diameter being a function of Vegf signaling strength (Figure 9e-h). Besides venous sprouting, the WT spinal cord is also vascularized from the arterial domain, most prominently the VTA, an arterial vessel that is lacking at day 3. Sprouts emanating from the VTA showed fewer lumenization events when compared to sprouts emanating from ISVs. ISV and VTA derived sprouts showed equal numbers of nuclei (Figure 8f,g). We conclude that the tertiary sprouting and day 12-14 spinal cord vascularization processes show similarities including the presence of lumenized sprouts, and the functional involvement of *aplnra* and *esm1*. However, the spinal cord vascularization processes are not completely identical in particular when considering the contribution of arterial sprouting in WT day 12-14. The data obtained suggest that elements of tertiary sprouting are recapitulated during WT spinal cord vascularization. To make the similarities and differences more clear we adapted the text of the manuscript accordingly, and we changed the title of the manuscript. The trunk spinal cord model showed us how parenchymal cues can induce a molecular distinct Vegf driven sprouting process. We postulate that knowledge on how parenchymal cues can control a variety of molecular distinct Vegfa driven sprouting processes, may help us understand how different organs shape their own “organo-typical” vascular network.

Question 1: The authors need to clarify conceptual advances beyond the previous work describing that neuronal soluble *flt1* can limit ISV over-sprouting around the developing spinal cord.

Answer 1: We added a new paragraph to the introduction and remodeled the discussion and addressed the outstanding questions and the conceptual advances regarding the novel Vegfa driven venous sprouting form. New text: “*In vivo* imaging of a series of zebrafish embryos with a Vegfa gain of function scenario (including *flt1* and *vhl* mutants) shows that upon Vegfa, sprouts preferentially emanate from trunk intersegmental veins, not from intersegmental arteries. These findings are intriguing as loss of function studies have previously shown that Vegfa is essential for primary artery sprouting, and Vegfc-Flt4 signaling for venous sprouting events (Hogan&Schulte Merker, Dev Cell, 2017). These observations raise several questions as what determines Vegfa sprouting responsiveness in venous endothelial cells, and are there differences between the molecular and cellular processes that govern Vegfa-driven sprouting in the arterial versus the venous domain. Moreover, venous sprouting upon Vegfa is spatially restricted to the dorsal aspect of intersegmental veins, a domain that is in close contact with the developing spinal cord neural system (Matsuoka, 2016; Wild, 2017). Ablation of spinal cord radial glia cells (Matsuoka, eLife, 2016) or neural cell specific inactivation of *flt1* (Wild, Nat Commun, 2017) phenocopies venous hypersprouting as observed upon Vegfa suggesting that neural cells may provide local instructive cues determining the spatial positioning of Vegf driven venous sprouting events. Still unresolved is how such neural cues contribute to regulating Vegf bio-availability and venous Kdr1 signaling strength to promote venous sprouting specifically at the neuro-vascular interface”.

We find that Vegfa induced venous sprouting requires genetic interaction between Apelin-receptor-a (*aplnra*) and Kdr1 to promote Vegfa signaling strength in a specific subset of competent venous endothelial cells. These venous cells migrate, in a blood flow dependent manner from the ventral to the dorsal aspect of venous ISVs, where they become juxtaposed to Vegfa and Apelin producing spinal cord neural cells and subsequently start to form a lumenized sprout. Loss of neural *apln* abrogates this process. We show that the venous endothelial cell selected for sprouting, here termed lumenized tip cell (based on the suggestion by #Rev1 now termed L-Tip cell), increases in size involving activation of the GEF Trio, and remodels into a unicellular, lumenized and pericyte covered, long-range anastomotic connection with an arterial ISV. After connecting with the arterial ISV, a perfused anastomotic vessel network forms at the level of the spinal cord. Vegfa induced sprout selection in the venous domain occurs independent of Dll4-Notch, requires hemodynamic forces, and is more glycolysis dependent when compared to Vegf-Dll4-Notch driven arterial sprouting – the reference sprouting model. We furthermore show that Vegf induced venous sprouting

requires the upregulation of endothelial cell specific molecule-1 (*esm1*) by spinal cord neural cells. *Esm1* is regarded as a competitive antagonist for Vegf binding to the extracellular matrix (ECM) (Rocha&Adams Circ Res, 2014). Consequently, increasing *esm1* levels and thereby masking potential Vegf binding sites in the spinal cord ECM, effectively increases Vegf availability at the neuro-vascular interface. In line with this we show that loss of (neural) *esm1* specifically inhibits ectopic venous sprouting in embryos with a Vegf gain of function scenario including *flt1*^{-/-} and *vhl*^{-/-} mutants, whereas neural specific overexpression of *esm1* augments the extent of venous sprouting and anastomosis formation. Genetic ablation or overexpression of *aplnra* or *esm1* has no impact on primary artery or secondary sprouting processes in the trunk. This suggests a unique contribution of vascular *aplnra* and neural *esm1* specifically in tertiary sprouting during this developmental stage. We also demonstrate that *aplnra* and *esm1* contribute to spinal cord vascularization in day 12-14 WT.

Conceptually, our observations may provide a frame-work to help understand and explain how tissues pattern their own vasculature. We propose a concept that integrates the local parenchymal dependent regulation of Vegfa responsiveness, with the selection of a special sprouting competent venous endothelial cell, to explain the spatial positioning of sprouting angiogenesis in a tissue context dependent manner. The requirement for two parenchymal derived cues, limits sprout activation specifically to a subset of endothelial cells that express the reciprocal receptor pair. Such specific interaction allows for a very stringent control of the spatial positioning of the sprouting process. We have added a new figure to the manuscript summarizing our venous sprouting model, back-to-back with the reference Dll4-Notch sprouting model (Supplemental Figure 13 & 14).

In the trunk, *flt1* is expressed in arteries and in neural cells. Regarding the role of arterial *flt1*, we would like to refer to a recent paper by Klems & le Noble, Nat Commun, 2020. Klems shows that arterial derived sFlt1 regulates Kdr1 signaling strength in arterial EC to control EC size and arterial lumen diameter involving the guanine exchange factor Trio. Arterial sprouting is not affected upon loss of *flt1*. Furthermore, arterial derived sFlt1 protein remains in close proximity of the arterial wall, and does not diffuse far enough to reach the venous domain hence making arterial derived sFlt1 an unlikely regulator of venous sprouting events. In the current MS we demonstrate that parenchymal derived cues determine Vegfa driven sprouting in the venous domain.

Question 2: Does the single lumenized EC, termed “pioneer cell”, emerge only in the tertiary sprouts in the developing vISVs of Vegf gain-of-function mutants such as *flt1*^{-/-} and *Vhl*^{-/-} mutants? **Though this is a novel finding**, it is not clear whether we can define the pioneer cell-mediated tertiary sprouts as a process of spinal cord vascularization.

Answer 2: for a detailed response - please see the **General Response** to reviewer 3. Indeed, we would like to make a distinction between the two processes: on the one hand, we describe a novel, Vegf driven, venous sprouting process that involves parenchymal cues, driving molecular and cellular events distinct from the reference Vegf-Dll4-Notch model. On the other hand, we examined if molecules and cellular events involved in tertiary sprouting, may contribute to vascularization of the spinal cord as occurs in day 12-14 WT. We concluded that the day 3 and day 12-14 spinal cord vascularization processes show similarities including the presence of single lumenized EC sprouts, and the functional requirement of *aplnra* and *esm1*. However, the spinal cord vascularization processes are not completely identical in particular when considering the contribution of arterial sprouting in WT day 12-14. To make this point more clear, we adapted the text of the manuscript accordingly, and we also adapted the title of the manuscript. The trunk spinal cord model showed us how parenchymal cues may contribute to heterogeneity in Vegf driven sprouting processes, and overall we believe that such knowledge may contribute to understanding how organs may shape their own vascular network. We are however very cautious, and for above mentioned reasons, do not want to claim that tertiary sprouting mediated sprouting around the spinal cord is exactly the same as the sprouting process observed at day12-14 – there is overlap, and there are differences (see Figure 8 and 9). The main thrust of the MS is the description of a novel Vegfa driven form of venous sprouting, which features lumenized endothelial cells, and is driven by parenchymal cues. To make this more clear we adapted the text of the discussion and title.

Question 3: Do the authors envision that the sprouts from vISVs in wild-types at 14 dpf (shown in Fig. 8A and C) develop as seen in *flt1*^{-/-} and *Vhl*^{-/-} mutants at 4dpf, given that neuronal Vegf expression is increased during the process of spinal cord vascularization at 12-14 dpf? If so, the authors need to examine the cellular properties of the sprouts from vISVs in wild-types at 12-14 dpf.

Answer 3: In line with the reviewers' suggestion, we examined the morphology of day 12-14 WT ISV sprouts and present new data in Figure 8 & 9. Please also see our **General Response** to reviewer 3, and response to question 2. In the day 12-14 WT scenario, the majority of ISV derived sprouts (80-85%) that innervate the spinal cord, emanate from ISVs that express the venous marker *flt4* or *CouptFII*, in line with a significant contribution of venous sprouting events in spinal cord vascularization (similar to the day 3-Vegf GOF scenario) (Figure 8a-e). We furthermore observed lumenized ISV sprouts: in the WT scenario the percentage of ISV derived sprouts with a blood plasma filled lumen was 67% as verified by labelling blood plasma with FITC-dextran (Figure 8h,i). At day 12, we observed single lumenized EC sprouts as well as lumenized sprout with more than one nuclei per sprout (Figure 8f,g). Interestingly, sprout diameter showed an association with Vegfa signaling strength (Figure 9e-h). In the period day 12-14, *flt1*^{-/-} mutants displayed a larger sprout lumen diameter when compared to WT, whereas *esm1*^{-/-} mutants showed a significantly reduced diameter in line with sprout diameter being a function of Vegfa-Trio signaling strength (Figure 9e-h).

We had WT available in *flt1;flt4* double reporter or *CouptFII* reporter background, allowing clear identification of arterial-venous ISV identity. Unfortunately, the *aplnra*^{-/-} mutants were in *Tg(fli1:eGFP)* background (which also labels neural crest derived cells and lymphatics which sometimes obscures imaging of spinal cord vessels) which made a 100% positive identification of venous ISVs, at this stage of development, difficult. At day 14, in *flt1*^{-/-}, *esm1*^{-/-} and *aplnra*^{-/-} mutants, we therefore first quantified the total amount of sprouting events we observed at the level of the spinal cord (Figure 9b,d,i,j). Next, where possible, we quantified the ISV derived sprouting events. We found that loss of *esm1* and *aplnra* reduced ISV sprouting propensity (Figure 9). Since the analyses in day 14 WT *flt1;flt4* or *CouptFII* show that the majority of ISV derived sprouts are of venous origin, the reduced sprouting propensity as observed in *aplnra*^{-/-} and *esm1*^{-/-} mutants most likely reflects a reduction in venous ISV sprouting. A more precise answer on the differential contribution of venous and arterial ISV sprouting, local hemodynamic factors, pericytes, notch activity, glycolytic status and the cross-talk with spinal cord cells herein will require backcrossing to several relevant reporter lines substantiated by detailed single cell analysis at day 12-14 and determining *aplnra* and *esm1* at single cell resolution, but we believe this is better suited for another manuscript. Collectively our data suggest that major components of the tertiary sprouting process like the formation of single lumenized EC sprouts are indeed recapitulated during WT spinal cord vascularization but that the processes are not 100% identical.

The authors provide mechanistic evidence underlying the pioneer cell-mediated tertiary sprouts in the developing vISVs of Vegf gain-of-function mutants. Indeed, given the up-regulation of Apelin (*apln*)-receptor-a (*aplnra*) signaling, the guanine exchange factor Trio, and endothelial cell specific molecule-1 (*esm1*) in *flt1*^{-/-} mutants, the authors clearly demonstrate that the lack of these molecules attenuates the pioneer cell-mediated tertiary sprouts in the developing vISVs of Vegf gain-of-function mutants.

Question 4: It is not clear whether these signaling molecules are involved only in the tertiary sprouts in the developing vISVs of Vegf gain-of-function mutants? It is not clear which ones they think are the most significant.

Answer 4: - please also see the response to reviewer 3 - questions 3 and 5. During early developmental stages, *aplnra* and *esm1* appear very specific for mediating tertiary sprouting; neither *aplnra* (Supplemental Fig. 7b-j,m) nor *esm1* (Supplemental Fig. 12e-h,k,l) are required for primary artery sprouting or secondary venous sprouting and subsequent AV remodeling of trunk ISVs; neither in WT nor in *flt1*^{-/-} or *vhl*^{-/-} mutant. At later stages, in the period day 12-14, we find that *aplnra* and *esm1* regulate sprouting propensity at the level of the spinal cord (Figure 9c,d,i,j). Previous studies have shown that WT spinal cord vascularization requires Vegf (Matsuoka, eLife, 2016), and our data suggest that – similar to the *flt1*^{-/-} day 3 scenario – *aplnra* and *esm1* may contribute to Vegf responsiveness between day 12-14.

Trio regulates endothelial cell size downstream of Kdr1 signaling, and EC size enlargement is one element contributing to the morphogenesis of the tertiary sprout in *flt1*^{-/-}. We previously showed how Trio is regulated downstream of Vegf signaling, and how Trio regulates EC size and arterial lumen diameter in zebrafish and human cell systems via promoting acto-myosin tension at endothelial cell junctions (Klems et al, Nat Commun, 2020). In the current MS, we focused on Trio in the venous domain and we demonstrate that overexpression of Trio in WT veins *in vivo* caused endothelial cell enlargement without inducing ectopic sprouting (Supplement Fig. 11g,h). Trio is highly upregulated in *flt1*^{-/-} and inhibiting Trio in *flt1*^{-/-} mutants reduced EC enlargement and lumenization of tertiary sprouts. Based on the data presented in the current MS and our previous work (Klems et al, Nat Commun, 2020), we conclude that Trio regulates endothelial cell size and shape. In mutants with a Vegf gain of function scenario, Trio mediated endothelial cell enlargement is required for achieving lumenization of the L-Tip cell and its expansion. In zebrafish, all trunk EC require a basal degree of Trio expression to maintain their size and loss of Trio causes severe vascular remodeling defects in all trunk vessels (for details see Klems et al, Nat Commun, 2020). Taken together: in mutants with a Vegf gain of function scenario, *aplnra*, *esm1* and *Trio* regulate distinct steps of the tertiary sprouting process: sprout cell selection in the venous domain, Vegf availability at the neuro-vascular interface, and EC size respectively (Supplemental Fig. 11i & Supplemental Fig. 14). We are currently exploring the contribution of mentioned molecules in the context of tissue repair.

Question 5: As mentioned above, **5a)** are these signaling molecules also involved in the sprouts from vISVs during the process of spinal cord vascularization at 12-14 dpf? Given that Vegfa-Kdr signaling is essential for the primary sprouts from the dorsal aorta, **5b)** the authors need to clarify whether these signaling molecules are involved in the primary sprout formation.

Answer 5: *Aplnra*^{-/-} mutants showed normal primary artery sprouting, and unaltered arterial to venous ISV ratios indicating that *aplnra* is not required for early trunk arterial/venous ISV remodeling (Supplemental Fig. 7b-j,m). *Esm1*^{-/-} mutants showed normal primary artery sprouting, and unaltered aISV/vISV ratios indicating that *esm1* is not required for early trunk arterial/venous ISV remodeling (Supplemental Fig. 12e-h,k,l). We conclude that both *esm1* and *aplnra* are not required for primary sprouting and ISV arterial-venous remodeling. In the period day 12 – 14 we find that *aplnra* and *esm1* regulate sprouting propensity during spinal cord vascularization (Figure 9c,d,i,j). For more details also see response to reviewer 3 - question 2, and our general response to reviewer 3.

Question 6: In introduction, the authors need to describe the previous work by their group (Wild et al. Nat Commn 2017) and the other group (Matsuoka et al. eLife 2016) and explain what questions remains unanswered.

Answer 6: In line with the reviewers' suggestion, we have adapted the text of the Introduction, mention both studies and address the open questions. The new text reads:

"*In vivo* imaging of zebrafish embryos with a Vegfa gain of function scenario shows that upon Vegfa, sprouts preferentially emanate from trunk intersegmental veins, not from intersegmental arteries (Matsuoka, eLife 2016; Wild, Nat Commun, 2017). These observations are intriguing as loss of function studies have previously shown that Vegfa is essential for primary artery sprouting, and Vegfc-Flt4 signaling for venous sprouting events (Hogan&Schulte Merker, Dev. Cell 2017). These observations raise several questions as what determines Vegfa sprouting responsiveness in venous endothelial cells, and are there differences between the molecular and cellular processes that govern Vegfa-driven sprouting in the arterial versus the venous domain. Moreover, venous sprouting upon Vegfa is spatially restricted to the dorsal aspect of intersegmental veins, a domain that is in close contact with the developing spinal cord neural system (Matsuoka, eLife 2016; Wild, Nat Commun, 2017). Ablation of spinal cord radial glia cells (Matsuoka, eLife 2016) or neural cell specific inactivation of *flt1* (Wild, Nat Commun, 2017) phenocopies venous hypersprouting as observed upon Vegfa suggesting that neural cells may provide local instructive cues determining the spatial positioning of Vegf driven venous sprouting events. Still unresolved is how such neural cues determine local Vegf bio-availability and venous Kdr1 signaling strength to promote venous sprouting

specifically at the neuro-vascular interface (Hogan&Schulte Merker, Dev. Cell 2017; Matsuoka, eLife 2016; Wild, Nat Communs, 2017)".

Question 7: The authors need to clarify whether the *flt1*^{-/-} mutants (*flt1*-ka604 and *flt1*-ka602) lack both soluble and membrane *flt1*.

Answer 7: the *flt1* ka604 (exon 3, -14 nt allele) and the *flt1* ka602 (exon 3, -5 nt) have a premature termination codon (PTC) resulting in a truncated protein devoid of a functional extracellular VEGF binding domain; these mutants lack both functional *sflt1* and *mflt1* as previously published (see Wild, Nat Communs, 2017, M&M part). The membrane bound Flt1 specific mutant (*mflt1* ka605) displays no vascular phenotype (Wild, Nat Communs, 2017), suggesting that sFlt1 and not mFlt1 is the relevant mediator of the venous sprouting phenotype. We thus far found no functional role for *mflt1* in trunk vascular remodeling (see also Klems, Nat Communs 2020). The *flt1* ka605 mutant was not used in the current study.

Of note: since the reviewer seems familiar with the different *flt1* mutant alleles of the different labs, we would like to point out that the *flt1* mutant used by the Matsuoka lab is also lacking both *mflt1* and *sflt1* (similar to ka602 and ka604). In their paper "Radial glia regulate vascular patterning around the developing spinal cord", Matsuoka et al., analyzed two *flt1* mutants. The *flt1*^{fh390} allele is an ENU mutant (Pan et al., 2015; Rossi et al., 2016), affecting exon 11, creating a loss of both the transmembrane and tyrosine kinase domains of mFlt1 without affecting the soluble form. Similar to our *mflt1* mutant (*mflt1* ka605), the *flt1*^{fh390} mutants have no vascular phenotype in the trunk. The *flt1*^{bns29} allele was created by using the CRISPR/Cas9 technique with a sgRNA targeting exon 3 of the *flt1* transcript (sgRNA sequence: 5' - GTATTGCAGCCGGCTGACAC - 3') and resulting in a premature STOP codon leading to a truncated 85aa. This mutant is a full *flt1* mutant affecting both sFlt1 and mFlt1 variants and is similar to our *flt1*^{-/-} mutants (*flt1*^{ka602} and *flt1*^{ka604}) also created by targeting exon 3 (sgRNA sequence: 5' - GGGACGGTGGGAGCTCCAGT - 3'). All these *flt1* alleles display the hyper-sprouting phenotype in the trunk.

sFlt1 and mFlt1 arise from alternative splicing of *flt1* mRNA at exon13-exon14. Creating a *sflt1* specific mutant without affecting the *mflt1* isoform is, in our opinion at present technically not feasible (simply because the first 13 exons are shared by both isoforms and deleting them thus affects both *sflt1* and *mflt1*). Both the Wild and the Matsuoka paper suggest that *sflt1* is the physiologically relevant regulator of sprouting propensity at the level of the spinal cord; hence given that *mflt1* is dispensable for sprouting in this setting, loss of *flt1* (as in the *flt1*^{-/-} mutant) is functionally equivalent to loss of *sflt1* but genetically not identical to a pure *sflt1* isoform specific mutant.

Question 8: The authors need to define the number of tertiary sprouts in the mutants they analyze. Is there one tertiary sprout from one vISV in *flt1*^{-/-} and *Vhl*^{-/-} mutants?

Answer 8: We quantified these parameters and found that on average there is one sprout per vISV but occasionally we observed two (Supplement Fig. 1d,e).

Question 9: There are multiple EC nuclei in tertiary sprouts in the *Vhl*^{-/-} mutants (Fig. 3d)

Answer 9: The reviewer is correct but the image the reviewer is referring to, is an image taken at day 4, after the formation of the venous to artery anastomosis. After anastomosis formation we see EC migration events and an increase in the number of ECs in the anastomosis. To clarify this issue, we measured EC number before (3dpf) and after (5dpf) anastomosis formation (Supplement Fig. 1f,g); before anastomosis it is 1, after the anastomosis it increases on average to 2.

Question 10: The authors claim that both *flt1*^{-/-} and *Vhl*^{-/-} mutants are used as Vegf gain-of-function mutants, but it would be better to examine whether Vegfa-overexpression directs the formation of tertiary sprouts from the developing vISVs.

Answer 10: We previously showed that ectopic venous sprouting was conserved between five different *vegfa* gain of function scenarios including **inducible neural specific Vegfaa165** overexpression (for clarity we inserted the relevant Fig.5, panel e; from Wild et al, Nat Communs,

2017). We measured venous sprouting in *flt1*^{-/-} mutants, *vhl*^{-/-} mutants, *ptena*^{-/-};*ptenb*^{-/-} double mutants, *flt1*^{-/-};*vhl*^{-/-} double mutants, and upon endoxifen inducible neuron specific *vegfaa165* overexpression. Venous sprouting was observed in all five VEGF gain of function scenarios (from Wild, Nat Commun, 2017, Figure 5, panel (e): Quantification of ectopic sprouting in indicated mutants and inducible neuronal-specific *vegfaa165* gain-of-function. In all models ectopic sprouting upon Vegf preferentially occurs in veins, mean±s.e.m., n=10-13/per group, t-test.

With respect to VEGF-overexpression, we tried three approaches: constitutive overexpression, tissue specific overexpression, and inducible tissue specific Vegf overexpression. The first approach results in supra-physiological amounts of VEGF and severe trunk vascular remodeling defects that cannot be interpreted (we describe such remodeling defects in detail in Klems, Nat Commun, 2020). We also examined (non-inducible) neural specific overexpression of *vegfaa165* and found

that it induced ectopic sprouting at the level of the spinal cord (Supplement Fig. 1p,q). However, the overall network showed signs of vessel overgrowth, suggestive of a Vegf overdose. So yes, it is possible to use a *vegfaa*-overexpression approach to induce ectopic venous sprouting but with the current expression vectors, it is very difficult to precisely titrate the physiologically relevant Vegf dosage, within the required time-window, while avoiding adverse effects like vessel overgrowth and leakage. We therefore prefer using the *flt1*^{-/-} and *vhl*^{-/-} mutant.

Flt1 and Vhl regulate Vegf at a different level, via Vegf scavenging, and Vegf transcription respectively. To obtain supra-physiological Vegf levels, we generated *flt1*^{-/-};*vhl*^{-/-} double mutants. The *flt1*;*vhl* double mutants, besides prominent venous sprouting, developed a small number of ectopic arterial sprouts (see Wild, 2017; Fig. 5e); a similar observations was made in inducible neuron specific *vegfaa165* gain of function embryos sprouts (Wild, 2017; Fig. 5e,h). The low sprouting frequency in the arterial domain is due to high Notch activity repressing ectopic sprouting from arteries. Beyond the focus of the current MS: we demonstrated that inhibiting Notch specifically in aISVs of *flt1*^{-/-} mutants, allows ectopic sprouting from arteries - however these sprouts failed to form a lumen (we observed a similar behavior in VTA derived sprouts). We concluded that in intersegmental arteries, Notch restricts ectopic sprouting in response to Vegf. Increasing Vegf to supra-physiological levels (as occurs in the *flt1*^{-/-};*vhl*^{-/-} double mutant or upon inducible neural specific *vegfa* overexpression) can overcome the repression exerted by active Notch in arteries (for more details on how Notch signaling was manipulated in arteries we refer to Wild et al, Nat Commun, 2017).

Question 11: Since the previous studies demonstrate the involvement of blood flow in the junctional fingers and lumen formation (Kotini et al. Cell Rep 2022), the authors need to examine whether tertiary sprouts are dependent on hemodynamic forces. What happens to tertiary sprouts in *flt1*^{-/-} and *Vhl*^{-/-} mutants with tricaine treatment (inhibition of blood flow and blood pressure) or norepinephrine treatment (increase in hemodynamic forces)?

Answer 11: In line with suggestion made by reviewer we examined tertiary sprouting in *flt1*^{-/-} mutants upon Tricaine and Epinephrine treatment (Supplemental Fig. 2a-f).

Tricaine treatment significantly reduced heart rate by about 50%, and significantly inhibited tertiary sprouting (Supplemental Fig. 2a-c). The same was observed with BDM (data not shown). Increasing heart rate in *flt1*^{-/-} mutants using epinephrine did not accelerate tertiary sprouting (Supplemental Fig. 2d-f). We conclude that tertiary sprouting requires hemodynamic forces. Of note: Vegf-Dll4/Notch driven primary sprouting and aISV formation does not depend on hemodynamic forces, and can occur in the absence of blood flow (Isogai&Weinstein, Development, 2003); in Vegf Dll4-Notch driven arterial sprouts, lumenization is a feature of the stalk cell, not the tip cell. To substantiate the hemodynamic component we started *in vitro* experiments (see also response to

#Rev1, Answer 4) and found that in human venous endothelial cells, flow is required to maintain endothelial *APLNR* expression; and loss of *APLNR* under flow conditions reduces *KDR* expression in venous endothelium. These preliminary *in vitro* data support the concept that hemodynamic forces contribute to tertiary sprouting in part by maintaining *Aplnr* expression.

Question 12: According to scRNAseq shown in Fig. 2, the authors need to indicate which cells express Vegfa.

Answer 12: we added new plots and indicate the *vegfa* expressing cells (Supplemental Fig. 5b,c). Previous studies using FACS sorted cells derived from *Tg(Xla.Tubb:DsRed)* (pan-neural marker) and *Tg(mnx1:GFP)* (motor neuron marker) transgenics further support neural expression of *vegfa* ligands (see in Wild et al, Nat Commun, 2017 the Figures 6a,b & Supplementary Figure 1a-k).

Question 13: In Fig. 2f, the authors need to explain why most ligand-receptor interactions including *vegfaa-kdr*, *cxcl12-cxcr4a* and *apln-aplnr* are not enhanced in *flt1*^{-/-} mutants compared to wild-types.

Answer 13: See also #Rev1, Answer 2. In Fig. 2f, we used CellChat analysis to predict the interaction probability of ligand-receptor pairs between endothelial cells and neural cells in both WT and *flt1*^{-/-} mutants. Based on single cell data, CellChat identifies differentially overexpressed ligands and receptors for each cell group (not between experimental conditions). Then CellChat computes the "law of mass action value" based on the average expression values of a ligand by one cell group and that of a receptor by another cell group, as well as their cofactors. This value is then used for the statistical analysis and to predict the communication between two cell groups mediated by these signaling genes. This method could indeed allow you to detect an increased probability of interaction if you have an increased number of cells expressing a certain gene or a higher level of expression for a gene, also if this increase is cell type specific. However, in our single cell data, our genes of interest are generally expressed by few cells and at a low level, so it is difficult to obtain a significant increase in expression even if we can see a tendency. Basically, every component identified by single cell RNA-seq should be confirmed by qPCR, and all CellChat predicted *in silico* interactions should be validated by *in vivo* experiments – and that is what we did.

qPCR and bulk RNA, established that *apln* and *aplnra* expression are significantly upregulated in *flt1*^{-/-} mutants (Supplement Fig. 7a and Fig. 5a of this MS & in Wild et al, Nat Commun, 2017). Subsequently, genetically ablating *aplnra* or neural *apln* confirmed that these genes have a physiologically relevant contribution in tertiary sprouting. Taken together these data support the prediction made by CellChat that Aplin-Aplnra signaling might be relevant for tertiary sprouting. We selected *Cxcr4-Cxcl12* because CellChat predicted a lot of EC-neural interactions for this pathway (for example it is known that *Cxcr4-Cxcl12* is important in regulating vascular development in certain parts of the nervous system, so it is not surprising it emerges in this analysis). We therefore validated this pathway *in vivo* and found no functional role for *Cxcl12-Cxcr4a* in mediating tertiary sprouting (Supplement Fig. 6d-k,o-i), and therefore did not examine this pathway in more detail. In *flt1*^{-/-}, the loss of the Vegf scavenging receptor soluble Flt1 and the resulting increase in Vegf bio-availability increase Kdr signaling strength; consequently the *vegfaa* and *kdr* mRNA levels may thus remain unaltered in *flt1*^{-/-}. The essential contribution of neural derived *vegfaa* and vascular *kdr* in venous sprouting was established previously (Wild et al, Nat Commun, 2017). Based on these *in vivo* results we decided to focus on interaction of Apelin - Apelin-receptor-α signaling with the Vegfa-Kdr signaling cascade.

Question 14: In Fig. 3, it is not clear whether *aplnrb* is also up-regulated in *flt1*^{-/-} mutants.

Answer 14: We find that *aplnrb* is significantly upregulated in *flt1*^{-/-} (Supplement Fig. 7n).

Question 15: The authors also need to examine whether the lack of *aplnrb* (*aplnrb*-mu270) results in the reduction of tertiary sprouts in *flt1*^{-/-} and *Vhl*^{-/-} mutants.

Answer 15: *Aplnrp* was significantly upregulated in *flt1*^{-/-} (Supplement Fig. 7n), and preferentially expressed in the arterial domain, most notably arterial tip cells (Figure 3k). *Aplnrp*^{-/-} mutants and *aplnrp*^{-/-};*flt1*^{-/-} double mutants showed severe trunk aISV and DLAV remodeling defects (Supplemental Fig. 7o-w), which consequently resulted in trunk vISV perfusion deficits, hence rendering analysis of hemodynamics driven tertiary sprouting in the venous domain impossible (because there were no functional mature perfused vISVs). As *aplnrp* was predominantly expressed in arterial tip cells and since *aplnrp* acted as regulator of aISV remodeling, we ruled *aplnrp* out as specific regulator of tertiary sprouting and therefore focused on Apelin-receptor-a.

Question 16: Although Fig.S5 indicates "Vascular phenotypes of *aplnrp*^{-/-} and *aplnrp*^{-/-} mutants," the authors do not show any data about *aplnrp*^{-/-} mutants.

Answer 16: we apologize for this omission. We now include the images of the *aplnrp*^{-/-} mutants in Supplement Figure 7o-w.

Question 17: In Fig. 4, it is not clear whether there is any difference in EC polarization and migration in the vISV between *flt1*^{-/-};*aplnrp*^{-/-} double mutants and wild-types. The authors need to compare between wild-types, *flt1*^{-/-}, and *flt1*^{-/-};*aplnrp*^{-/-} double mutants.

Question 17: in line with the suggestion made by the reviewer we performed the analysis: WT vs *flt1*^{-/-} migration data are in Supplement Fig. 9a-e; the polarization data in Supplement Fig. 9k, all other data in Figure 4.

Question 18: In Fig. 5, the authors need to examine the effect of EC-derived Apelin (*apln*) in tertiary sprouts in *flt1*^{-/-} mutants. It is not clear whether neuron-specific loss of *apln* reduces tertiary sprouts in *flt1*^{-/-} mutants at 3 dpf. The authors should examine the vascular phenotype at 4 dpf.

Answer 18: To substantiate the physiological role of neural *apln* in tertiary sprouting we generated a CrisprCas9 based neuron specific *apln* loss of function scenario in *flt1*^{-/-} embryos. We found that genetically ablating *apln* from neural cells reduced tertiary sprouting at the level of the spinal cord in *flt1*^{-/-} mutants at day 3 and 4 (Fig. 5l,m & Supplemental Fig. 10b,c). These data suggest that spinal cord neural cells produce a physiologically relevant amount of Apelin necessary for tertiary sprouting to unfold.

A publication by Helker (eLife, 2020) has previously suggested that in developing arteries endothelial *apln* acts cell-autonomously to promote aISV sprouting, and loss of *apln* impairs aISV development and perturbs trunk perfusion. This study indicates that endothelial Apelin is acting in the arterial domain. This is supported by our single cell data indicating high levels of *apln* expression in the tip cells of the arterial domain and arterial Cap, whereas *apln* expression in venous endothelial cells is very low when compared to arteries (see single cell seq graph below, Figure 5b).

We are particularly interested in factors specific for venous remodeling, and since loss of *apln* impairs growth of arteries it is not specific for venous development. We are also not convinced that an endothelial cell autonomous acting factor can determine the spatial positioning of a sprout – in our view this requires a parenchymal signal. It could however be argued that L-Tip cells produce apelin in a cell autonomous manner, and this L-Tip cell derived apelin together with neural derived Apelin,

contributes to tertiary sprouting. Technically, to demonstrate a contribution of L-Tip cell derived apelin in a genetic approach would require generating a L-Tip cell specific loss of function mutant. We are technically not able to produce a L-Tip cell specific *apln* loss of function mutant, as we are not aware of a promoter specific and strong enough for driving sufficient Cas9 specifically and with high efficiency in a single venous endothelial cell in the dorsal domain of vISVs, without affecting other vascular domains. Although most functional and sequencing data indicate that endothelial apelin acts in the arterial domain, for reasons explained above, we can formally not rule out a contribution in the venous domain. To acknowledge this, we have adapted the text in our MS accordingly.

Question 19: The authors claim that an increase in the tertiary sprout volume is due to Trio and its downstream effector Rac1 acting on EC actin remodeling (Fig. 6). It is not clear whether the Trio-Rac1 axis is involved only in tertiary sprouts in *flt1*^{-/-} mutants. Is there any difference in the size of ISVs between *flt1*^{-/-};Trio morpholino mutants or *flt1*^{-/-};Rac1 inhibitor mutants and wild-types.

Answer 19: we previously reported that Trio can regulate EC size, in zebrafish trunk vessels and in human arterial and venous endothelial cells *in vitro* (Klems et al, Nat Commun, 2020). We described in great detail on how VEGF-Kdr signaling activates Trio, and how Trio subsequently activates Rac1 specifically in junctional regions of endothelial cells to promote actin remodeling, acto-myosin tension at the cell junction and EC size *in vitro* and *in vivo* (Klems et al, Nat Commun, 2020). In short: using deletion constructs we identified the GEF1 domain of Trio (TrioN) to be responsible for activating Rac1 in (venous and arterial) endothelial cells *in vitro*. Using photo-activatable Rac1 constructs and Rac1 sensor assays, we next showed that Trio activates Rac1 specifically at the endothelial cell junction. There Rac1 locally promotes actin remodeling and acto-myosin tension culminating into endothelial cell enlargement. We furthermore demonstrated that aISV specific TrioN gain of function *in vivo* in the zebrafish trunk, promotes EC size and outward arterial lumen diameter remodeling – without inducing ectopic sprouting. For a more detailed description of the Trio loss of function, and Rac1 inhibitor - diameter phenotypes we would like to refer to Klems et al, Nat Commun, 2020. Furthermore, our collaborators recently showed that serine (S1785/1786) phosphorylation of Trio potentiates the localization of Trio to junctional regions, resulting in locally promoting the exchange for Rac1 specifically at junctional regions (Daniel & van Buul, Small GTPases, 2023). Hence, this explains why Trio specifically activates Rac1 only in EC junctional regions, and this junctional region-specific activity of Rac1 induces the acto-myosin tension required for the cell size enlargement. Thus Trio is a special GEF as it seems to activate Rac1 only in the junctional regions, the region controlling the EC size, and not in other areas of the cell. In the current MS we describe how venous specific overexpression of TrioN in the WT trunk vasculature regulates venous ISV lumen caliber (see #Rev 3, answer 20), without inducing sprouting. We conclude that in the venous domain, specific elements of tertiary sprout morphogenesis upon Vegf – namely EC size increase and lumenization – require Trio. Gain of Trio in the absence of a Vegf gain of function scenario, is not sufficient to induce a venous sprout.

Question 20: Does venous specific overexpression of TrioN alone induce tertiary sprouts in the developing vISVs?

Answer 20: No, venous specific overexpression of TrioN in the trunk augmented venous ISV lumen diameter (consistent with Trio mediating EC size), without inducing ectopic sprouting (Supplement Figure 11g,h). See also Rev#3, answer 19.

Question 21: Although EC-specific overexpression of *esm1* does not induce additional venous sprouts in *flt1*^{-/-} mutants (Fig. 7), it would be better to show whether EC-specific overexpression of *esm1* can restore tertiary sprouts in *flt1*^{-/-}; *esm1*^{-/-} double mutants.

Answer 21: In line with the suggestion made by the reviewer we overexpressed *esm1* under control of the *fli1* promoter in *flt1*^{-/-}; *esm1*^{-/-} double mutants but found no significant rescue of sprouting events (Figure 7n,d).

Question 22: The capillary and neuronal phenotypes in the transverse section (Fig. 8e) are difficult to interpret because of the poor image qualities. The capillary images need to be improved due to lack of cellular morphology (especially in *esm1*^{-/-} mutants). The authors claim that change in capillaries affects neural differentiation in the spinal cord of *flt1*^{-/-} and *esm1*^{-/-} mutants, but no further investigation including proliferation and cell death of ventricular zone neural progenitors and mature neurons are provided in support.

Answer 22: we thank the reviewer for the feedback on this figure, and we agree that additional analyses can strengthen this part. Raising additional adult mutant fish and cross them with suitable neural stem cell, neural progenitor and cell-cycle reporter lines will require substantial amounts of time (at least 8-10 months to get to adult homozygous mutants in the appropriate backgrounds), which is more than allocated for a revision, and we therefore omitted this figure from the current MS. We replaced this figure with a more detailed cellular analysis of sprouting events in day 12 – 14 WT in particular the occurrence of lumenized sprouts, the contribution of venous ISV derived sprouts in spinal cord vascularization and the molecular control of sprouting propensity by *esm1* and *aplnra*.

REVIEWER COMMENTS

Reviewer #1 (Remarks to the Author):

The revised manuscript by Preau et al effectively addressed most of my concerns. In particular, I appreciate their effort to clearly define the "pioneer cell", which was changed to a more generalized 'lumenized tip cell'. Moreover, including hemodynamic forces in their analyses provides more physiological relevance to their findings. It would have been more convincing had the authors been able to demonstrate the relevance of Apln polypeptide. However, the reviewer also acknowledges technical difficulties associated with the proposed experiments and appreciate the efforts the authors had put into this matter.

Reviewer #2 (Remarks to the Author):

I think the authors for a comprehensive reply.

Reviewer #3 (Remarks to the Author):

The authors have provided additional data aiming to enhance the characterization of tertiary sprouts, termed lumenized tip cell (L-Tip cell), originating from the dorsal part of venous inter-segmental vessel (vISV) in a Vegf gain of function scenario including flt1 and vhl mutants at 3-4 dpf. While the authors have indeed demonstrated comparable L-Tip cell-led lumenized sprouts invading the spinal cord parenchyma in a wildtype scenario at 12-14 dpf, this reviewer still posits potential limitations in the conceptual advancements concerning the understanding of the mechanism driving Vegfa-mediated venous endothelial sprouting.

The emergence of L-Tip cell-led tertiary sprouts is exclusively observed in the dorsal part of vISV in flt1 and vhl mutants. While the authors have indicated the emergence of ectopic sprouts in mutants carrying neural-specific Vegfaa165 overexpression, the presence of a combined phenotype featuring ectopic sprouts and vessel overgrowth has prevented further examination to ascertain whether these ectopic sprouts are indeed L-Tip cell-led. This suggests that L-Tip cell-led sprouts might arise in response to a restricted range of Vegfa concentrations, or specifically in flt1 and vhl mutants. Considering that secondary venous sprouts from the posterior cardinal vein (PCV) is not L-Tip cell-led, the characterization of L-Tip cell-led tertiary sprouts seems to leave uncertainty regarding how it contributes to understanding the mechanism that underlies venous endothelial sprouting.

The detailed analysis of spinal cord vascularization is a valuable addition in this revised manuscript.

However, the authors need to differentiate between ISV-derived sprouts and VTA-derived arterial sprouts, and assess the effects of *flt1*, *aplnra*, *esm1*, and *trioa* deletion specifically in the ISV-derived sprouts. While *esm1* knockout specifically reduces the number of L-Tip cell-led tertiary sprouts and *trio* knockdown reduces the volume of L-Tip cell-led tertiary sprout in *flt1* mutants at 3-4 dpf, *esm1* knockout results in reductions in both L-Tip cell-led sprout number and volume during spinal cord vascularization at 12-14 dpf. The authors need to examine what happens to the volume of L-Tip cell-led sprout in *trio* knockdown in the spinal cord. The authors also need to explain whether VTA-derived arterial sprouts are L-Tip cell-led. Do the authors envision the emergence of L-Tip cell-led sprouts in both arterial and venous sprouting events during spinal cord vascularization at 12-14 dpf? Does this represent a distinct feature from L-Tip cell-led tertiary sprouts at 3-4 dpf?

- The authors should include the descriptions of the *flt1* mutants (response to reviewer 3's question 7) in the text.
- (Comments about the additional data): The description of the data regarding metabolic sensitivity in primary arterial sprouts versus L-Tip cell-led tertiary sprouts through the treatment of AZ67 (Supplemental Fig. 2J-M) in the text needs improvement. What is AZ67? The difference in tertiary sprouts between *flt1* mutants treated with DMSO versus those treated with AZ67 does not seem significant in the images (Supplemental Fig. 2I).
- (Comments about the additional data): In Supplemental Fig. 2N, it remains unclear whether GCaMP5G signals are detected in primary arterial sprouts: the GCaMP5G signals outlined by dotted lines do not appear to overlap with *Kdr1*;TagBFP signals, indicative of arterial sprouts.
- (Comments about the additional data): In Supplemental Fig. 3F-H, the data indicate that *Dll4* knockdown in a wild-type scenario results in the induction of tertiary sprouts as observed in *flt1* mutants. The authors need to clarify why an increase in the number of venous ISVs can induce tertiary sprouts without relying on a *Vegf* gain of function scenario.

Reviewer #3 (Remarks to the Author):

The authors have provided additional data aiming to enhance the characterization of tertiary sprouts, termed lumenized tip cell (L-Tip cell), originating from the dorsal part of venous inter-segmental vessel (vISV) in a Vegf gain of function scenario including *flt1* and *vhl* mutants at 3-4 dpf. While the authors have indeed demonstrated comparable L-Tip cell-led lumenized sprouts invading the spinal cord parenchyma in a wildtype scenario at 12-14 dpf, this reviewer still posits potential limitations in the conceptual advancements concerning the understanding of the mechanism driving Vegfa-mediated venous endothelial sprouting.

Question 1: The emergence of L-Tip cell-led tertiary sprouts is exclusively observed in the dorsal part of vISV in *flt1* and *vhl* mutants. While the authors have indicated the emergence of ectopic sprouts in mutants carrying neural-specific *Vegfaa165* overexpression, the presence of a combined phenotype featuring ectopic sprouts and vessel overgrowth has prevented further examination to ascertain whether these ectopic sprouts are indeed L-Tip cell-led. This suggests that L-Tip cell-led sprouts might arise in response to a restricted range of Vegfa concentrations, or specifically in *flt1* and *vhl* mutants. Considering that secondary venous sprouts from the posterior cardinal vein (PCV) is not L-Tip cell-led, the characterization of L-Tip cell-led tertiary sprouts seems to leave uncertainty regarding how it contributes to understanding the mechanism that underlies venous endothelial sprouting.

Answer 1: To determine the range of Vegf levels capable of inducing tertiary sprouting in the day 3-4 zebrafish trunk, we examined 6 different Vegf gain of function scenarios: *flt1*^{-/-}, *vhl*^{-/-}, *ptena*^{-/-};*ptenb*^{-/-} double mutants, *flt1*^{-/-};*vhl*^{-/-} double mutants, neuron specific and inducible *vegfaa165* gain of function (GOF) and somite specific constitutive *vegfaa165* gain of function (technical details on how these scenarios were induced are described in detail in Wild *et al.*, Nat Commun 2017; Klems *et al.*, Nat Commun 2020). Lumenized sprouts and perfused anastomotic connections could be observed in all these scenarios except for the constitutive *vegfaa165* gain of function scenario; this scenario consistently induced EC overgrowth. To clarify for this reviewer the range of Vegf levels that are compatible with the formation of lumenized sprouts and anastomotic connections we provide here an explanatory **Figure 1** (see below) showing on the one hand Vegf signalling output, and on the other hand the vascular phenotypes we observed in the trunk-(the displayed images of the trunk vasculature from the different Vegf GOF scenarios are derived from Wild *et al.*, Nat Commun 2017; Klems *et al.*, Nat Commun 2020, the current manuscript, and previously unpublished data-sets).

==== continued on the next page =====

Figure 1: Graphical illustration showing Vegf signalling strength and corresponding vascular phenotype for the indicated mutants and transgenic scenarios. *Fit1^{-/-};vhl^{-/-}*, endoxifen inducible neural specific *vegfaa165* gain of function scenario (adapted from Wild *et al.*, Nat Communs, 2017), somite specific constitutive *vegfaa165* (from Klems *et al.*, Nat Communs, 2020).

Formation of lumenized and perfused anastomotic vessel segments are not only observed in *fit1^{-/-}* and *vhl^{-/-}* mutants, but also in *fit1/vhl* double mutants (these double mutants have higher Vegf signalling strength than the single mutants, Wild *et al.*, Nat. Communs, 2017) and upon endoxifen inducible neural specific *vegfaa165* gain of function. In the inducible neuron specific *vegfaa* gain of function scenario we observed ectopic venous sprouts at the level of the spinal cord (**Figure 2** - below) similar to that observed in *fit1^{-/-}* or *vhl^{-/-}* mutants (quantification in Wild *et al.*, Nat Communs, 2017).

Figure 2. Confocal imaging of the trunk vasculature at 3dpf showing formation of tertiary sprouts upon endoxifen inducible neuronal specific *vegfaa165* gain of function. aSV arterial, vSV venous intersegmental vessel.

In addition, at later stages, we observed lumenized anastomotic connections at the level of the neural tube, and dilated trunk ISVs (**Figure 1**, inducible *vegfa165* GOF scenario) in line with Vegf gain of function promoting EC and lumen size (Klems *et al.*, Nat. Commun, 2020; and this manuscript).

To summarize our response to the reviewers' question: yes, it is possible to uncouple sprouting and EC overgrowth when expressing *vegfaa165*, in an inducible and neuron specific manner, and tightly titrate *vegfa* levels in a spatio-temporal manner. In our setup, the productive angiogenesis phenotype in the inducible neural specific *vegfaa165* gain of function scenario (**Figure 2** – see page 2) was obtained when endoxifen was administered from 2.5 dpf to 3 dpf at a concentration of 1 μ M. This inducible tissue specific transgenic approach is technically feasible, however, much more time consuming than examining *flt1*^{-/-} and *vhl*^{-/-} mutants and for practical reasons we therefore prefer studying the mutants. **We conclude:** formation of L-tip cell-led sprouts can occur in a wide range of Vegf levels, including those levels achieved by activating the HiF1a-(Vhl)-Vegf pathway or titrating soluble Flt1, both of which are highly relevant regulators of Vegf availability during (patho)physiological conditions, as well as with transgenic expression of *vegfaa165* (when done in an inducible and neuron specific manner).

Of note: secondary venous sprouting, as mentioned by the reviewer is a Vegfc – Vegfr3 driven process, and detailed analysis of sprouting morphology in Vegfc gain of function scenarios is beyond the scope of this manuscript.

Question 2: The detailed analysis of spinal cord vascularization is a valuable addition in this revised manuscript. However, the authors need to differentiate between ISV-derived sprouts and VTA-derived arterial sprouts, and assess the effects of *flt1*, *aplnra*, *esm1*, and *trioa* deletion specifically in the ISV-derived sprouts.

Answer 2: in line with the suggestions made by the reviewer **we performed additional analyses** and differentiated between ISV-derived sprouts and VTA-derived arterial sprouts. The **analyses** for the *flt1*^{-/-} mutant and *esm1*^{-/-} mutant are presented in **Figure 9 B,D and in new Suppl. Figure 13 F,G**. Loss of *flt1* increased, whereas loss of *esm1* significantly reduced ISV derived sprouting (**Figure 9A-D**). For the *aplnra* mutant, we were unable to precisely determine the origin of all the sprouts in larvae due to the transgenic background used for labelling vessels (we used *Tg(fli1:EGFP)*; *fli1* besides arteries and veins also labels the lymphatics and the lymphatics derived fluorescence caused a lot of background, hampering a clear view on vessel identity in this setting). To overcome these problems, we quantified the total sprout number – a parameter that also includes ISV derived sprouts - and observed a significant reduction upon loss *aplnra* (**Fig. 9i and 9J**). *Trioa* mutants died during embryonic stages and could therefore not be examined at larval stages. We conclude that *flt1*, *esm1* and *aplnra* contribute to sprouting events at the level of the spinal cord during larval stages, consistent with their actions observed at day 3-4.

Question 3: While *esm1* knockout specifically reduces the number of L-Tip cell-led tertiary sprouts and *trio* knockdown reduces the volume of L-Tip cell-led tertiary sprout in *flt1* mutants at 3-4 dpf, *esm1*

knockout results in reductions in both L-Tip cell-led sprout number and volume during spinal cord vascularization at 12-14 dpf. The authors need to examine what happens to the volume of L-Tip cell-led sprout in trio knockdown in the spinal cord. (larval stage or embryo stage).

Answer 3: we thank the reviewer for suggesting this experiment. Unfortunately, *trioa* mutants died during embryonic stages and could therefore not be examined at larval stages. Morpholinos are not active beyond day 5, hence examining vascular phenotypes at larval stages from morphant embryos injected with a low dosage *trioa* targeting morpholino (to circumvent embryonic lethality) is not useful either. However, since Trio acts downstream of Vegf signalling to regulate EC and lumen size, we hypothesized that changes in *trioa* expression upon modulating Vegf availability or signaling, will be reflected in the sprout lumen dimensions. Thus, loss of *flt1* and thereby augmenting Vegf bio-availability predicts increased Vegf signalling strength, active Trio, causing larger ECs and increased diameter. This is exactly what we observed *in vivo*: sprouts from *flt1*^{-/-} mutants displayed a significantly larger diameter when compared to controls at day 14 (**Figure 9E and 9F**). Conversely, loss of *esm1* predicts reduced Vegf signalling strength, Trio activity and sprout lumen dimension. Indeed, sprout lumen diameter was significantly reduced in *esm1*^{-/-} mutant larvae, (**Figure 9G and 9H**). We also expanded the measurements of venous sprouts to analysis of *flt1*^{-/-};*esm1*^{-/-} double mutants at 3 days. We found that the *flt1*^{-/-};*esm1*^{-/-} double mutants showed a decrease in both sprout surface and volume when compared to *flt1*^{-/-} single mutant (**Supplemental Figure 12R and 12S**).

Question 4: The authors also need to explain whether VTA-derived arterial sprouts are L-Tip cell-led. Do the authors envision the emergence of L-Tip cell-led sprouts in both arterial and venous sprouting events during spinal cord vascularization at 12-14 dpf? Does this represent a distinct feature from L-Tip cell-led tertiary sprouts at 3-4 dpf?

Answer 4: we thank the reviewer for this interesting suggestion that some of the arterial sprouts may show characteristics we attribute to veins. We re-examined the VTA sprouting events and concluded that there are two possible explanations. Scenario 1: while the VTA is considered arterial, some parts of the VTA actually expressed both arterial and venous markers, most likely due to the particular flow patterns in this region (see **Figure 3 on page 5 below, and new Suppl. Figure 13 C and D in the revised MS**).

==== continued on the next page =====

Figure 3. Confocal imaging (left) and graphical illustration of VTA and ISV AV identity (middle) in 12 dpf spinal cord. Note that within the VTA territory (boxed area), some domains express both the arterial marker *flt1* (red) and the venous marker *flt4* (green) suggestive of a population of endothelial cells with a mix arterial-venous identity (boxed area in the left panel; domain indicated in orange on the middle panel). Right panel: Quantification of VTA-sprouts emanating from *flt1*;*flt4* expressing domain. (**new Supplemental Figure 13E in the revised MS**).

Thus, with respect to arterial-venous identity, VTA EC's are not all homogeneous arterial, leaving open the possibility that the VTA EC's expressing both AV markers tend to behave more like a vein. We indeed identified VTA endothelial cells that expressed both *flt1* and *flt4* and gave rise to a sprout (**new Suppl. Figure 13E**). Scenario 2: the occurrence of lumenization and the time-point of sprout imaging. Lumenization of primary arterial sprouts forming the alSVs typically starts upon lumen remodeling of the trailing stalk cell (involving processes such as inverse blebbing, fusion of vacuoles or cord hollowing). At the stage when lumenization occurs in the arterial sprout, it is considered multi-cellular and already quite expanded. In contrast, the L-tip cell-led tertiary sprout in the venous domain is lumenized from the beginning, and at this stage the sprout is uni-cellular. This leaves open the option that the lumenized VTA sprout we present in our figure was imaged at a relatively late stage of its development, after stalk cell lumen remodeling took place, with the sprout being multi-cellular. We therefore decided to re-examine the VTA-derived sprouts.

Figure 4. Confocal imaging of VTA sprouts after injection fluorescent dextran to label blood plasma (left,middle) and quantification (right) for indicated scenario. Note 2 populations: 1) the short non lumenized VTA sprouts (left) and 2) the more expanded - elongated VTA sprouts containing blood plasma, indicative of a functional lumen (middle). (**new Supplemental Figure 13A,B in the revised MS**).

In the population of VTA derived sprouts we observed two populations: (1) slim, short, predominantly non-lumenized endothelial sprout structures (**Figure 4**, left panel) (compatible with early stages of arterial sprout formation), and (2) elongated and mostly lumenized structures (**Figure 4**, middle panel), consistent with a later stage of arterial sprout development. Unfortunately, we do not have a nuclear marker in these images to establish whether the sprout is multi-cellular in this latter population. Lumenization propensity was higher in the elongated sprout population in line with the concept that arterial sprouts become lumenized at later stages once the stalk cells build a lumen (**Figure 4**, right panel and new **Suppl. Figure 13 A and B**).

To address the reviewer's question more precisely, it would be essential to redo all *in vivo* imaging at day 12-14, and include a time-lapse imaging of VTA formation in *Tg(fli1:nuclear-EGFP; kdrl:HRAS-mCherry)* to monitor cell number, monitor arterial-venous markers using *Tg(-0.8flt1:RFP;flt4:mCitrine)* double reporter and determine Notch signalling strength with Notch reporter to establish identity, image pericytes, and determine lumenization by simultaneous injection of FITC-dextran to image blood plasma in the developing sprouts at the level of the spinal cord in day 12-14 larvae. VTA formation is a very interesting element contributing to spinal cord vascularization, but investigating the impact of VTA ECs with dual identity on sprouting behaviour at day 12 is beyond the context of the current revision. We therefore refrain from claiming that, in certain conditions, arterial sprouts can be L-tip cell-led.

Question 5: The authors should include the descriptions of the *flt1* mutants (response to reviewer 3's question 7) in the text.

Answer 5: In line with the reviewers' question, we extended the description of our *flt1*^{-/-} mutants – we added a paragraph to the materials and methods. For a more comprehensive description on how these *flt1* mutant alleles were generated and validated, in particular with respect to *sflt1* and *mflt1* expression, we would like to refer to Wild *et al.* Nat Communs, 2017; Klems *et al.*, Nat Communs, 2020.

Question 6: The description of the data regarding metabolic sensitivity in primary arterial sprouts versus L-Tip cell-led tertiary sprouts through the treatment of AZ67 (Supplemental Fig. 2J-M) in the text needs improvement. What is AZ67? The difference in tertiary sprouts between *flt1* mutants treated with DMSO versus those treated with AZ67 does not seem significant in the images (Supplemental Fig. 2I).

Answer 6: we thank the reviewer for this suggestion. A series of publications by Katrin De Bock & Peter Carmeliet showed that angiogenic sprouts rely on glycolysis, a process that generates ATP without the need for oxygen. The rationale behind this is, that angiogenic sprouts tend to grow and expand into regions that are devoid or low in oxygen availability, hence, the Krebs-cycle and oxidation in the mitochondria to produce energy will not work efficiently in such environment. PFKFB3 is an essential enzyme in the glycolysis pathway (it generates an allosteric regulator of a glycolytic key enzyme), and inhibiting the glycolysis pathway by blocking PFKFB3 has been suggested to reduce angiogenesis. To compare the glycolysis sensitivity between Vegf driven arterial and venous sprouts, we incubated embryos with

AZ67, a potent inhibitor of PFKFB3 (Burmistrova *et al.*, Scientific Reports, 2019). We noted that tertiary sprouts were more sensitive to inhibiting glycolysis when compared to arterial sprouts. We thus conclude that in the trunk, tertiary sprouts rely more on glycolysis than arterial sprouts. To strengthen the results of this experiment, we imaged and quantified more embryos. We added the new images and the quantification in **Supplemental Fig. 2I**.

Question 7: In Supplemental Fig. 2N, it remains unclear whether GCaMP5G signals are detected in primary arterial sprouts: the GCaMP5G signals outlined by dotted lines do not appear to overlap with *Kdr1*;TagBFP signals, indicative of arterial sprouts.

Answer 7: We agree that the signal in the BFP reporter line is relatively weak when compared to the GFP signal of the GCaMP5G reporter leading to the impression that both signals do not overlap. To clarify this issue we improved the images at better resolution using a different endothelial reporter *Tg(kdr1:HRAS-mCherry)* – the new images are present in **Supplemental Figure 12N**. Of note: GCaMP5G is expressed under the control of an endothelial specific promoter, hence its signal is EC specific.

Question 8: In Supplemental Fig. 3F-H, the data indicate that *Dll4* knockdown in a wild-type scenario results in the induction of tertiary sprouts as observed in *flt1* mutants. The authors need to clarify why an increase in the number of venous ISVs can induce tertiary sprouts without relying on a Vegf gain of function scenario.

Answer 8: Theoretically there are 2 possibilities to explain the changes in Vegf signalling strength and sprouting in the *dll4* loss of function scenario: 1) increased sensitivity of endothelial cells for Vegf or 2) increased expression of *vegfaa*, *apelin* or downregulation of *sflt1* in spinal cord neurons. Spinal cord glia cells and neurons produce sFlt1, Apelin and Vegf. *Dll4* and *notch* are well expressed in the spinal cord neurons and since the *dll4* targeting morpholino targets *dll4* expression in both vessels and nerves, loss of *dll4* may have affected the expression of *sflt1*, *apelin*, *vegfaa* at the level of the spinal cord, creating a pro-angiogenic environment. This scenario can be tested in a follow-up study by FACS sorting of trunk spinal cord neural cell populations and measuring *sflt1*, *apelin* and *vegfaa* expression by qPCR. Alternatively, it is well established that loss of *dll4* augments the sensitivity of endothelial cells for Vegf. *Aplnra* contributes to Vegf signalling strength in veins, and we propose that the venous sprouting events upon loss of *dll4* may involve a component regulated by *aplnra*. In our follow-up study we are planning to address the interactions between notch, AV identity in EC and regulation of *aplnra* at the single cell level in more detail.

REVIEWERS' COMMENTS

Reviewer #3 (Remarks to the Author):

The authors have diligently addressed the previous comments from this reviewer, presenting additional data and explanations in response to the comments. With the administration of endoxifen at the appropriate developmental timing and concentrations, the authors have nicely demonstrated the emergence of L-Tip cell-led sprouts in the inducible neuron-specific vegfaa165 mutants, mirroring a similar phenotype observed in flt1 and vhl mutants. These data suggest that L-Tip cell-led sprouts appear to arise in response to a restricted range of Vegfa, rather than specifically in flt1 and vhl mutants.

Overall, this reviewer recommends publication.